# Validation of Sentinel-5P TROPOMI tropospheric NO₂ products by comparison with NO₂ measurements from airborne imaging, ground-based stationary, and mobile car DOAS measurements during the S5P-VAL-DE-Ruhr campaign

Kezia Lange[1], Andreas Richter[1], Anja Schönhardt[1], Andreas C. Meier[1], Tim Bösch[1], André Seyler[1], Kai Krause[1], Lisa K. Behrens[1], Folkard Wittrock[1], Alexis Merlaud[2], Frederik Tack[2], Caroline Fayt[2], Martina M. Friedrich[2], Ermioni Dimitropoulou[2], Michel Van Roozendael[2], Vinod Kumar[3], Sebastian Donner[3], Steffen Dörner[3], Bianca Lauster[3], Maria Razi[3], Christian Borger[3], Katharina Uhlmannsiek[3], Thomas Wagner[3], Thomas Ruhtz[4], Henk Eskes[5], Birger Bohn[6], Daniel Santana Diaz[7], Nader Abuhassan[8], Dirk Schüttemeyer[9], and John P. Burrows[1]

[1]Institute of Environmental Physics, University of Bremen, Bremen, Germany
[2]Royal Belgian Institute for Space Aeronomy, Brussels, Belgium
[3]Max Planck Institute for Chemistry, Mainz, Germany
[4]Institute for Space Science, FU Berlin, Berlin, Germany
[5]KNMI, Royal Netherlands Meteorological Institute, De Bilt, Netherlands
[6]Institute of Energy and Climate Research, IEK-8: Troposphere, Forschungszentrum Jülich GmbH, Jülich, Germany
[7]LuftBlick, Innsbruck, Austria
[8]Joint Center for Earth Systems Technology, University of Maryland, Baltimore County, USA
[9]European Space Agency, ESA-ESTEC, Noordwijk, Netherlands

**Correspondence:** Kezia Lange (klange@iup.physik.uni-bremen.de)

**Abstract.** Airborne imaging differential optical absorption spectroscopy (DOAS), ground-based stationary and car DOAS measurements were conducted during the S5P-VAL-DE-Ruhr campaign in September 2020. The campaign area is located in the Rhine-Ruhr region of North Rhine-Westphalia, Western Germany, which is a pollution hotspot in Europe comprising urban and large industrial sources. The DOAS measurements are used to validate space-borne NO₂ tropospheric vertical column density (VCD) data products from the Sentinel-5 Precursor (S5P) TROPOspheric Monitoring Instrument (TROPOMI). Seven flights were performed with the airborne imaging DOAS instrument for measurements of atmospheric pollution (AirMAP), providing measurements which were used to create continuous maps of NO₂ in the layer below the aircraft. These flights cover many S5P ground pixels within an area of 30 km x 35 km and were accompanied by ground-based stationary measurements and three mobile car DOAS instruments. Stationary measurements were conducted by two Pandora, two zenith-sky and two MAX-DOAS instruments. Ground-based stationary and car DOAS measurements are used to evaluate the AirMAP tropospheric NO₂ VCDs and show high Pearson correlation coefficients of 0.88 and 0.89 and slopes of $0.90 \pm 0.09$ and $0.89 \pm 0.02$ for the stationary and car DOAS, respectively.

Having a spatial resolution of about 100 m x 30 m, the AirMAP tropospheric NO₂ VCD data create a link between the ground-based and the TROPOMI measurements with a nadir resolution of 3.5 km x 5.5 km and is therefore well suited to validate the

TROPOMI tropospheric $NO_2$ VCD. The observations on the seven flight days show strong $NO_2$ variability, which is dependent on the three target areas, the weekday, and the meteorological conditions.

The AirMAP campaign data set is compared to the TROPOMI $NO_2$ operational off-line (OFFL) V01.03.02 data product, the reprocessed $NO_2$ data, using the V02.03.01 of the official level-2 processor, provided by the Product Algorithm Laboratory (PAL), and several scientific TROPOMI $NO_2$ data products. The AirMAP and TROPOMI OFFL V01.03.02 data are highly

correlated (r = 0.87) but are showing an underestimation of the TROPOMI data with a slope of $0.38 \pm 0.02$ and a median relative difference of -9 %. With the modifications in the $NO_2$ retrieval implemented in the PAL V02.03.01 product the slope and median relative difference increased to $0.83 \pm 0.06$ and +20 %. However, the modifications resulted in larger scatter and the correlation decreased significantly to r = 0.72. The results can be improved, by not applying a cloud correction for the TROPOMI data in conditions with high aerosol load and when cloud pressures are retrieved close to the surface. The influence

of spatially higher resolved a priori $NO_2$ vertical profiles and surface reflectivity are investigated using scientific TROPOMI tropospheric $NO_2$ VCD data products. The comparison of the AirMAP campaign data set to the scientific data products shows that the choice of surface reflectivity data base has a minor impact on the tropospheric $NO_2$ VCD retrieval in the campaign region and season. In comparison, the replacement of the a priori $NO_2$ profile in combination with the improvements in the retrieval of the PAL V02.03.01 product regarding cloud heights can further increase the tropospheric $NO_2$ VCDs. This study

demonstrates that the underestimation of the TROPOMI tropospheric $NO_2$ VCD product with respect to the validation data set has been and can be further significantly improved.

## 1 Introduction

The reactive nitrogen oxides, nitrogen monoxide (NO) and nitrogen dioxide ($NO_2$) collectively known as $NO_x$ ($= NO + NO_2$), are important tropospheric air pollutants and have a strong impact on the tropospheric chemistry. In addition to emissions from

soils, natural biomass burning and lightning, they are largely released into the troposphere by a variety of human activities. These include fossil fuel combustion processes of power plants, by traffic and in industrial areas, as well as man-made biomass burning. $NO_x$ is primarily emitted as NO, which is reacting with ozone ($O_3$) and is rapidly forming $NO_2$. The $NO_x$ sources are spatially and temporally highly variable, and nitrogen compounds are reactive and short lived. As a result, the spatial and temporal variability of $NO_2$ is large, especially in regions characterized by a variety of $NO_x$ emission sources. $NO_x$ in the

troposphere is toxic and impacts the chemical composition and environmental condition, e.g., through tropospheric ozone catalytic production cycles (Chameides and Walker, 1973; Fishman and Crutzen, 1978; Jacob et al., 1996) or its reaction with the hydroxyl radical, OH, the most important tropospheric daytime oxidizing agent. Accurate knowledge of the spatial and temporal distribution of $NO_2$ in the troposphere is therefore required to better understand tropospheric chemistry.

Atmospheric $NO_2$ is remotely observed on a variety of platforms, including ground-based stations, moving platforms such

as cars, ships or aircraft, and environmental satellites. Applying the DOAS (Differential Optical Absorption Spectroscopy) technique (Platt and Stutz, 2008) in the UV and visible spectral range, the absorption signature of $NO_2$ can be identified and column densities can be retrieved.

After earlier satellite missions have observed stratospheric $NO_2$, to investigate stratospheric $O_3$ chemistry (Dubé et al., 2020), $NO_2$ in the troposphere has been retrieved from space observations since the launch of GOME in 1995 (see e.g., Burrows et al.,

1999; Richter and Burrows, 2002; Beirle et al., 2010; Boersma et al., 2011; Hilboll et al., 2013a). As $NO_2$ has high spatial variability in the troposphere, the spatial resolution has been gradually improved from GOME (ground footprint 320 km x 40 km) to SCIAMACHY (60 km x 30 km), GOME-2 (80 km x 40 km), OMI (13 km x 24 km), and to the recent TROPOMI instrument (5.5 km x 3.5 km at nadir). With a focus on diurnal variations, projects with geostationary instruments are now being deployed such as the Korean instrument GEMS (Kim et al., 2020), launched in February 2020, NASA's TEMPO (Zoogman et al., 2017)

planned for launch in 2023, and ESA's Sentinel-4 (Ingmann et al., 2012) planned for launch in 2024.

To ensure the accuracy of satellite data products for use in research, policy making, or other applications, each data product from satellite sensors needs to be validated and its accuracy determined. Validation measurements are needed in polluted and clean regions by independent instruments operating on different platforms. Measurements from ground-based sites provide continuous validation data from different locations for the trace gas products, retrieved from satellite instruments (e.g.,

Verhoelst et al., 2021). Measurements from mobile ground-based platforms like cars enable the observation of the spatial variability in addition to its temporal evolution. Thus, they are used for the comparison with satellite observations (Wagner et al., 2010; Constantin et al., 2013; Wu et al., 2013) and the validation of airborne remote sensing measurements (Meier et al., 2017; Tack et al., 2017; Merlaud et al., 2018). Airborne remote sensing measurements are an additional valuable source of validation data. Airborne mapping experiments have been performed in the recent years using different aircraft imaging DOAS instru-

ments such as AMAXDOAS, APEX, AirMAP, SWING, SBI, GeoTASO or GCAS (e.g., Heue et al., 2005; Popp et al., 2012; Schönhardt et al., 2015; Meier et al., 2017; Tack et al., 2019; Judd et al., 2020). The aircraft viewing geometry is similar to that of a satellite, but airborne measurements are able to measure at higher spatial resolution than the satellite sensors. Airborne observations are only available for short periods and are concentrated on the campaign region, but compared to measurements from ground-based sites offer the advantage that larger areas and full satellite ground pixels are observed in a relatively short

period around the satellite overpass. Thus, spatiotemporal variations of trace gas data products become visible at sub satellite ground pixel resolution. The combination of airborne imaging, ground-based stationary and mobile measurements enables the validation of satellite data products over a long period and at a high spatial resolution.

Focusing on TROPOMI, Verhoelst et al. (2021) have compared TROPOMI tropospheric $NO_2$ VCDs of OFFL V01.02 - V01.03.02 to tropospheric $NO_2$ VCD data from in total 19 MAX-DOAS ground stations. Depending on the level of pollu-

tion, the TROPOMI tropospheric $NO_2$ VCD data show a negative bias compared to the ground-based observations. Recent studies by Tack et al. (2021) and Judd et al. (2020), comparing airborne tropospheric $NO_2$ VCD data products to TROPOMI tropospheric $NO_2$ VCD data of V01.02 and V01.03.01, also show a significant underestimation of TROPOMI compared to the airborne observations.

Modifications in the TROPOMI $NO_2$ retrieval led to V02.02, operational since 1 July 2021. The main changes influencing

the tropospheric $NO_2$ VCD are: (1) Cloud pressures derived from the new FRESCO-wide algorithm, leading to lower cloud pressures and thus larger tropospheric $NO_2$ VCDs over polluted scenes with small cloud fractions, and (2) over cloud-free scenes a surface albedo correction is leading to larger tropospheric $NO_2$ VCDs. On average ground-based validation shows an

improvement of the negative bias of the tropospheric $NO_2$ VCDs from -32 % to -23 % (van Geffen et al., 2022b).

Different aspects that influence the tropospheric $NO_2$ VCD determination and possible reasons for the underestimation of the TROPOMI tropospheric $NO_2$ VCD data, compared to the validation data, are discussed in several studies (e.g., Judd et al., 2020; Tack et al., 2021; Verhoelst et al., 2021; van Geffen et al., 2022b; Douros et al., 2022). The limited knowledge of the $NO_2$ profiles, and differences in the averaging kernels between instruments having different viewing geometries, are identified as significant potential sources of disagreement between satellite and validation data. Similarly, inaccuracies in the knowledge of the aerosol load and aerosol vertical profile lead to underestimations as well as overestimations of the tropospheric $NO_2$ VCD, depending also on the viewing geometry. In addition, the knowledge about the surface reflectivity and cloud conditions and their treatment in satellite retrieval algorithms needs to be taken into account.

In the present study, results from a comprehensive field study conducted in North Rhine-Westphalia in September 2020 are presented. The campaign area is located in the West of Germany and includes the highly polluted Ruhr Area, a metropolitan region with large cities, industrial facilities, power plants and arterial highways. Background areas with low pollution, as well as moderately polluted regions are also observed, which increases the dynamic range of observed values. This campaign utilized the mapping capabilities of the Airborne imaging DOAS instrument for Measurements of Atmospheric Pollution (AirMAP) and includes a ground-based component for the evaluation of the AirMAP data set, comprised of three mobile car DOAS and six stationary DOAS devices. AirMAP is used for regional mapping of areas large enough to contain several TROPOMI pixels. Possible reasons for the bias of the TROPOMI tropospheric $NO_2$ VCD product are investigated by a systematic variation of the relevant input parameters in the satellite retrieval.

The field campaign site and setup are described in Sect. 2. The instruments and data sets are explained in Sect. 3. After a thorough comparison of AirMAP to stationary DOAS (Sect. 4) and car DOAS data (Sect. 5), the campaign data set is used to evaluate TROPOMI tropospheric $NO_2$ products (Sect. 6), including the operational OFFL V01.03.02 product active during the campaign phase and the reprocessed data PAL V02.03.01. Starting from these base versions, scientific products are developed that enable a dedicated assessment of the retrieval issues described above and the assumptions used about the $NO_2$ profile, clouds, and surface reflectivity.

## 2 The S5P-VAL-DE-Ruhr campaign

The objective of the S5P-VAL-DE-Ruhr campaign, an activity within the ESA QA4EO project, was to perform comprehensive field studies optimized for TROPOMI tropospheric $NO_2$ VCD validation including airborne, ground-based stationary and mobile car DOAS measurements.

The campaign activities took place in September 2020 in North Rhine-Westphalia including the Ruhr area, a densely populated and strongly polluted urban agglomeration in the West of Germany. The Ruhr area itself has a population of 5 million. Together with the populated surroundings and metropolitan centers along the Rhine, the region is called Metropolitan area Rhine-Ruhr (MRR). It comprises a population of more than 10 million inhabitants, large power plants, energy intensive industrial facilities and several large highways. $NO_2$ pollution in the MRR is clearly visible in TROPOMI maps of Europe showing widespread

enhanced NO$_2$ amounts. Figure 1 shows the monthly average for September 2020 of the tropospheric NO$_2$ VCD using the TROPOMI PAL V02.03.01 product for central Europe (left) and a close-up of the S5P-VAL-DE-Ruhr campaign region (right).

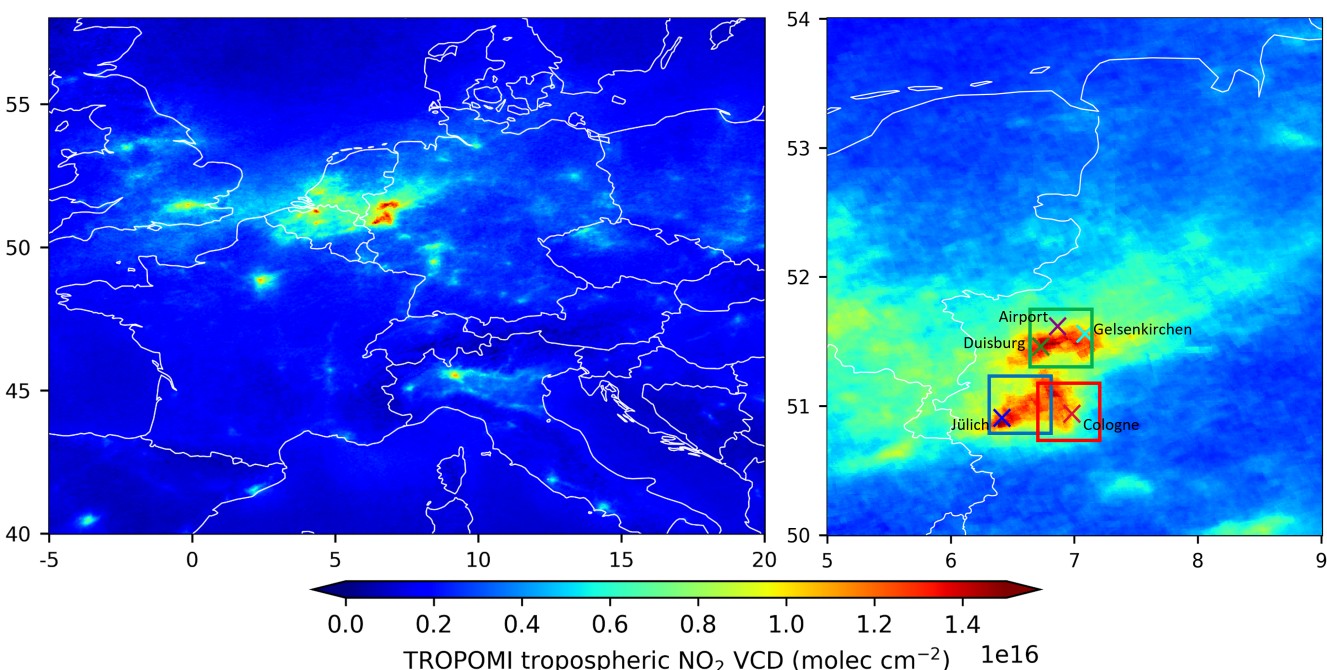

**Figure 1.** S5P TROPOMI tropospheric NO$_2$ VCD taken from the PAL V02.03.01 product for the month of September 2020, in central Europe (left) and a close-up map of the campaign target area, North Rhine-Westphalia (right). The three research flight target areas and the ground-based measurement sites are shown.

A key contribution to the campaign is the airborne AirMAP instrument explained below in Sect. 3.2. AirMAP was installed on a Cessna T207A aircraft that was based at an airport close to Dinslaken, North Rhine-Westphalia. Within the designated campaign area, three research flight areas were defined (see. Fig. 1), in which AirMAP performed in total seven flights on seven consecutive days. The aircraft observations covered a large number of neighboring TROPOMI ground pixels reasonably close in time to the TROPOMI observations.

Figure 2 shows a map of the region, in which flights were made during the campaign, including examples of the flight patterns flown in the three research flight areas within the region: around Jülich in the Southwest (blue track), around Cologne in the Southeast (red track) and around Duisburg in the North (green track). The research flight area around Jülich is expected to be dominated by the emissions of three large lignite fired power plants located in the area (see European Pollutant Release and Transfer Register, https://industry.eea.europa.eu/, last access: 18 November 2022). The research flight area around Cologne is a mixed urban and industrial area. The flight area around Duisburg has a similar character to that of the Cologne area but includes the central metropolitan Ruhr area, which has a large variety of pollution sources. The individual research flight area on each

of the campaign days was selected after assessment of the weather and atmospheric conditions, in particular wind direction and the objective of measuring all of the three research flight areas on a clear-sky day. For the flight days, the weather conditions were favorable having mostly cloud free scenes over the particular target area.

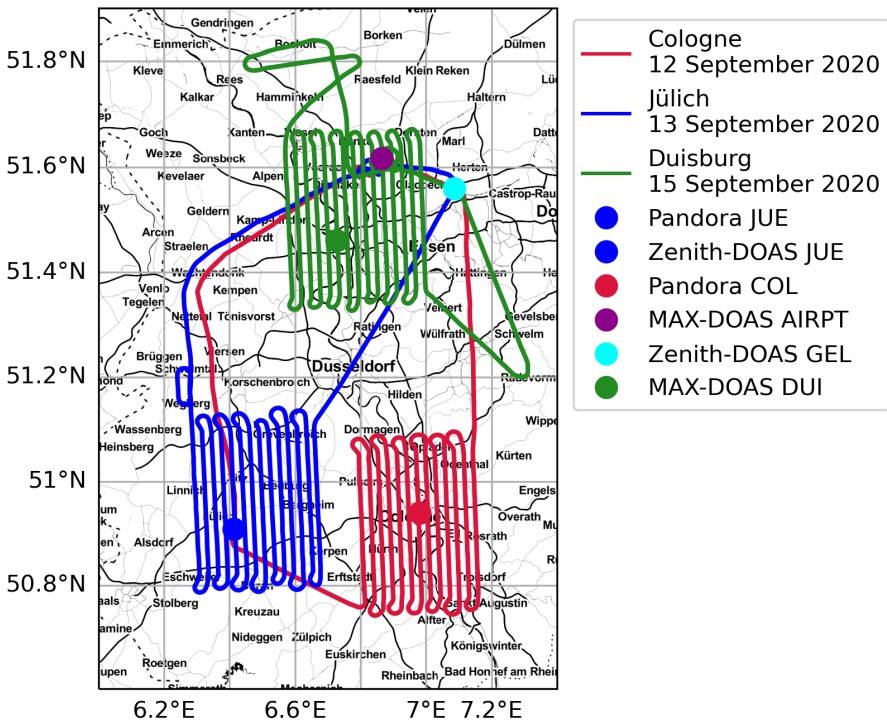

**Figure 2.** Overview of the flight area of the Ruhr campaign, including exemplary flight patterns in the three target areas and locations of the stationary instruments in Jülich (blue), Cologne (red), Dinslaken airport (purple), Gelsenkirchen (cyan) and Duisburg (green).

The selected flight area is covered with straight flight tracks in a lawn mower style. Neighboring flight legs are flown in opposite directions and have an overlap of approximately 30 % at the edges of the airborne instrument swath. For each flight, 13 to 15 flight tracks, each having a width and length of approximately 3 km and 35 km, were performed above the target area. The transfer flights between airport and target areas were used to overpass nearby stationary instruments. Flight schedules used the S5P overpass times to optimize the amount of data for validation. In general, it was planned to have the S5P overpass in the middle of the flight. On days where two overpasses per day occurred in the target area, the flight schedules were optimized towards the overpass time at the smaller viewing zenith angle (VZA) of TROPOMI. More details of the flights are given in Table 1.

The campaign delivered validation measurements by a mobile and a stationary component. In addition to the measurements made by AirMAP, the mobile component included three car DOAS devices. The stationary component comprised six ground-

**Table 1.** List of the aircraft activities including S5P overpass information. All times are UTC. On days with two S5P overpasses over the area, flights were arranged to coincide with the overpass at smaller VZA of TROPOMI.

| Date | Flight time (UTC) | Flight area | S5P overpass (UTC) with VZA | Comments |
|---|---|---|---|---|
| 12 Sep 2020 | 10:17-13:37 | Cologne | 10:51 (67.4°), 12:31 (15.9°) | |
| 13 Sep 2020 | 10:20-13:36 | Jülich | 12:12 (8.8°) | |
| 14 Sep 2020 | 10:14-13:47 | Duisburg | 11:53 (30.7°), 13:35 (64.9°) | No TROPOMI data |
| 15 Sep 2020 | 09:15-12:44 | Duisburg | 11:35 (46.7°), 13,15 (55.4°) | |
| 16 Sep 2020 | 10:37-14:05 | Duisburg | 11:16 (57.7°), 12:56 (41.9°) | Only one car DOAS |
| 17 Sep 2020 | 10:45-14:16 | Jülich | 10:57 (65.5°), 12:37 (22.6°) | |
| 18 Sep 2020 | 10:48-14:08 | Cologne | 12:18 (1.6°) | |

based remote sensing instruments of three different types, i.e., two Pandora instruments, two MAX-DOAS instruments and two
fixed zenith-sky devices. All the instruments were placed at suitable locations within the selected research flight areas shown in Fig. 2. With this combination of measurements, a comprehensive comparison of the airborne measurements with different types of ground-based instruments is made possible. This provides a basis for the evaluation of the TROPOMI tropospheric NO$_2$ VCD products with the airborne data, which cover well the satellite pixel areas. The airborne imaging data link the ground-based observations with restricted spatial but good temporal coverage to satellite observations that have large swath
widths but at a single instance in time.

## 3 Instruments and data sets

During the S5P-VAL-DE-Ruhr campaign, tropospheric VCDs of NO$_2$ were retrieved from instruments mounted on satellite, airborne, car and stationary ground-based platforms. All these instruments are passive remote sensing spectrometers using the DOAS technique (Platt and Stutz, 2008) by analyzing visible and UV spectra of scattered sun light. The instruments involved
in the S5P-VAL-DE-Ruhr campaign activities are listed in Table 2. The data analysis was done independently by the operating institutes and the DOAS fitting window was chosen based on the spectrometer's spectral wavelength range and from the groups experience with their instrument.

### 3.1 S5P TROPOMI

The Copernicus satellite S5P was launched into a Sun-synchronous orbit at 824 km in October 2017. S5P carries a single
instrument, TROPOMI, which comprises a hyperspectral spectrometer measuring radiation in the ultraviolet, visible, and near and shortwave infrared spectral regions (Veefkind et al., 2012). TROPOMI provides observations between 10:50 and 13:45 UTC over the campaign region, measuring the distribution of atmospheric columns from trace gases such as NO$_2$,

**Table 2.** List of instruments included in S5P-VAL-DE-Ruhr campaign activities with location and observation geometry. Car DOAS instruments are operated by three different institutes: Institute of Environmental Physics, University of Bremen (IUP), Max Planck Institute for Chemistry in Mainz (MPIC) and the Royal Belgian Institute for Space Aeronomy (BIRA).

| Instrument | Location/Platform | Observation geometry | Spectral range (nm) | Fitting window (nm) | VCD retrieval and AMF information (columns in $molec\, cm^{-2}$) |
|---|---|---|---|---|---|
| TROPOMI | Sentinel-5P | Push-broom, nadir | 310-500 | 405-465 | van Geffen et al. (2022a) |
| AirMAP | FU-Berlin Cessna T207A aircraft | Push-broom, nadir | 429-492 | 429-492 | $VCD_{trop, ref} = 1 \cdot 10^{15}$ Radiative transfer model SCIATRAN |
| IUP car DOAS | Mobile car | Zenith-sky | 290-550 | 425-490 | $VCD_{trop, ref} = 1 \cdot 10^{15}$ AMF (90°) = 1.3 |
| MPIC car DOAS | Mobile car | Zenith-sky and 22° | 300-460 | 400-460 | Using dSCD (22°), Wagner et al. (2010) AMF (90°) = 1.3, AMF (22°) = 3 |
| BIRA car DOAS | Mobile car | Zenith-sky and 30° | 200-750 | 450-515 | dSCD (30°) with sequential 90° reference AMF (90°) = 1.3, AMF (30°) = 2.5 |
| Zenith-DOAS JUE | Jülich (50.91° N, 6.41° E) | Zenith-sky | 290-550 | 425-490 | $SCD_{ref} = 1 \cdot 10^{16}$ AMF (90°) = 1.3 |
| Zenith-DOAS GEL | Gelsenkirchen (51.56° N, 7.09° E) | Zenith-sky | 290-550 | 425-490 | $SCD_{ref} = 1.7 \cdot 10^{16}$ AMF (90°) = 1.3 |
| MAX-DOAS DUI | Duisburg (51.46° N, 6.73° E) | Multi-axis | 282-414 | 338-370 | dSCD (30°) with sequential 90° reference AMF (90°) = 1.3, AMF (30°) = 2.5 |
| MAX-DOAS AIRPT | Airport Dinslaken (51.62° N, 6.87° E) | Multi-axis | 300-463 | 411-445 | MMF inversion algorithm Friedrich et al. (2019) |
| Pandora COL | Cologne (50.94° N, 6.98° E) | Multi-axis | 270-520 | 435-490 | Cede et al. (2021) |
| Pandora JUE | Jülich (50.91° N, 6.41° E) | Multi-axis | 281-523 | 435-490 | Cede et al. (2021) |

HCHO, CHO.CHO, BrO, $SO_2$, $O_3$, CO, $CH_4$ and aerosol and cloud properties. Thereby TROPOMI extends a long record of satellite-based observations. With its good signal-to-noise ratio and a spatial resolution at nadir of 3.5 km x 5.5 km (initially 3.5 km x 7 km, changed on 6 August 2019), which is more than 10 times better than that of its predecessor, the Ozone Monitoring Instrument (OMI, Levelt et al. (2006)), it is currently the best instrument for monitoring small-scale emission sources of $NO_x$ from space.

### 3.1.1 TROPOMI NO$_2$ operational OFFL V01.03.02 product

During the campaign activities in September 2020, the TROPOMI level-2 NO$_2$ OFFL V01.03.02 product was generated operationally. For the retrieval of NO$_2$ slant column densities (SCD) the measured spectra are analyzed using the DOAS technique in the fitting window 405 nm - 465 nm. The SCDs are separated into their stratospheric and tropospheric parts, using the TM5-MP global chemistry transport model. The tropospheric SCDs are converted into tropospheric VCDs by applying tropospheric air mass factors (AMFs), estimated using a look-up table of altitude-dependent AMFs, the OMI Lambertian equivalent reflectivity (LER) climatology (Kleipool et al., 2008), NO$_2$ vertical profiles from the TM5 model, and cloud fraction and pressure information from the FRESCO-S algorithm (van Geffen et al., 2022a).

Validation by comparison with other observations has shown that NO$_2$ data versions V01.02 - 01.03 are biased low by up to

50 % over highly polluted regions (e.g., Verhoelst et al., 2021). As discussed in several validation studies (see e.g., Judd et al., 2020; Verhoelst et al., 2021; van Geffen et al., 2022b), this underestimation could be related to biases in the cloud pressure retrieval, to a too high cloud pressure from the FRESCO-S algorithm, in particular when the cloud fractions are low and/or during periods of high aerosol loading. Other stated factors that could contribute to the underestimation are: (1) the low spatial resolution of the used a priori $NO_2$ profiles from the TM5 global chemistry transport model (e.g., Judd et al., 2020; Tack et al., 2021), (2) the use of the OMI LER climatology given on a grid of 0.5° x 0.5° for the AMF and cloud fraction retrieval in the $NO_2$ fit window, and (3) the GOME-2 LER climatology (0.25° x 0.25°) measured at mid-morning used for the NIR-FRESCO cloud retrieval (van Geffen et al., 2022b). These LER climatologies are not optimal for TROPOMI, because of TROPOMI's higher spatial resolution and the missing consideration of the viewing angle dependency in the LER products (Lorente et al., 2018; van Geffen et al., 2022b). In V02.04, operational since July 2022, a directionally dependent LER (DLER) climatology derived from TROPOMI observations given on a resolution of 0.125° x 0.125° is applied for AMF and cloud fraction retrieval in the $NO_2$ fit window and to the NIR-FRESCO cloud retrieval (Eskes and Eichmann, 2022). Since V02.04 is not yet reprocessed and thus not available for the campaign period, it is not included and discussed in this study.

### 3.1.2 Scientific TROPOMI $NO_2$ V01.03.02 CAMS product

The scientific TROPOMI $NO_2$ V01.03.02 CAMS product is based on the operational OFFL V01.03.02 product. The original 1° x 1° TM5 a priori $NO_2$ profiles are replaced by the Copernicus Atmospheric Monitoring Service (CAMS) analyses. AMFs and tropospheric $NO_2$ VCDs were recalculated using the averaging kernels and other quantities available in the level-2 $NO_2$ files, following the approach described in the TROPOMI product user manual (Eskes et al., 2022). Between the surface and 3 km the CAMS European regional analyses with an improved resolution of 0.1° x 0.1° are used. For altitudes between 3 km and the tropopause the CAMS global analyses (0.4° x 0.4°) are used. More detailed explanations can be found in Douros et al. (2022).

### 3.1.3 TROPOMI $NO_2$ PAL V02.03.01 product

Modifications in the TROPOMI $NO_2$ retrieval led to the OFFL V02.02 product, which is operationally produced, since 1 July 2021. To obtain a harmonized data set, a complete mission reprocessing was performed using the latest operational product OFFL V02.03.01, of 14 November 2021. The reprocessed data version available from 1 May 2018 to 14 November 2021 provided by the Product Algorithm Laboratory (PAL) is labeled as PAL V02.03.01. This provided the opportunity to compare the campaign data set to the OFFL V01.03.02 and the new PAL V02.03.01. The main change compared to the OFFL V01.03.02 impacting the tropospheric $NO_2$ VCD data is the use of the FRESCO-wide algorithm instead of the FRESCO-S algorithm, which was already introduced in V01.04 and was operational from 29 November 2020 to 1 July 2021. The FRESCO-wide algorithm provides lower and therefore more realistic cloud pressures (i.e. clouds are at higher altitudes), especially for scenes when cloud fractions are low. This change results in decreased tropospheric AMFs, which leads to higher tropospheric $NO_2$ VCDs (van Geffen et al., 2022b). Another update that can have a significant impact is the correction of the surface albedo over cloud free scenes by using the observed reflectance. This increases the tropospheric $NO_2$ VCDs by about 15 % over polluted

regions in case the retrieved cloud fraction is zero (van Geffen et al., 2022b). For this study the effect is negligible since only 1 out of the here analyzed 117 TROPOMI pixels is observed as cloud free. van Geffen et al. (2022b) also describes the following other modifications, which have only a small or no impact on the tropospheric $NO_2$ VCD data. Level-1b v2.0 (ir)radiance spectra are updated in the new version, and are increasing the $NO_2$ SCD of about 3 %, from which most of it ends up in a slightly increased stratospheric VCD. The improved level-1b v2.0 also leads to a small increase of completely cloud-free pixels and to slightly lower cloud pressures for pixels with a small cloud fraction, resulting in tropospheric $NO_2$ VCDs being about 5 % higher for these ground pixels. An introduced outlier removal is increasing the amount of good quality retrievals over the South Atlantic Anomaly and over bright clouds where saturation can occur. The change to new spatially higher resolved snow and ice information is increasing the amount of valid retrievals at high latitudes. On average, the new data version increased the tropospheric $NO_2$ VCDs by 10 % to 40 % compared to the V1.x data, depending on season and pollution. The largest increase is found in wintertime at mid and high latitudes. First comparisons to ground-based measurements show an improvement of the negative bias of the TROPOMI tropospheric $NO_2$ VCDs from on average -32 % to -23 % (van Geffen et al., 2022b).

### 3.1.4   Scientific TROPOMI $NO_2$ IUP V02.03.01 product

For the evaluation of the influence of auxiliary data on the TROPOMI $NO_2$ product, we developed a customized scientific product rebuilding the V02.03.01 data product, named IUP V02.03.01. The IUP V02.03.01 gives the possibility to change the a priori assumptions such as surface reflectance, which cannot be done using the averaging kernel approach used for V01.03.02 CAMS.

The a priori $NO_2$ vertical profile shapes for the operational TROPOMI $NO_2$ retrieval are taken from the TM5 model and have a resolution of 1° x 1° (~100 km x 100 km), which is much coarser than the TROPOMI data (3.5 km x 5.5 km at nadir). In highly polluted regions, such as the campaign area, high spatial variability of $NO_2$ VCDs are observed. The $NO_2$ plumes from sources, such as power plants, industrial complexes or cities, cannot be resolved in the model. To demonstrate the impact of higher resolved a priori $NO_2$ vertical profiles, we recalculated AMFs and the tropospheric $NO_2$ VCDs with a lookup-table created with the radiative transfer model SCIATRAN (Rozanov et al., 2014) using a priori tropospheric profiles from the 0.1° x 0.1° CAMS regional analyses for altitudes between the surface and 3 km. For altitudes between 3 km and the tropopause, where horizontal variability is in general small, the TM5 model analyses are used. Two maps showing the $NO_2$ distribution of the CAMS regional and the TM5 analyses for the campaign region can be found in the Appendix Fig. A1. In the following, this data version using the CAMS regional analyses is called IUP V02.03.01 REG.

The surface reflectivity information from the 5-year OMI LER climatology, used for the operational TROPOMI AMF calculations has a resolution of 0.5° x 0.5°. After more than 3 years of TROPOMI data acquisition, a TROPOMI surface reflectivity database, estimated from 36 months of TROPOMI v1.0.0 level-1b data, provides LER data, as a function of month, wavelength, latitude and longitude and at a finer spatial resolution of 0.125° x 0.125° (Tilstra, 2022). The recalculation of AMFs and tropospheric $NO_2$ VCDs using the regional CAMS $NO_2$ profiles and the TROPOMI LER results in the product named IUP V02.03.01 REG TROPOMI LER. The use of the TROPOMI LER in this data set is limited to the $NO_2$ AMFs and not extended to the cloud retrieval.

In addition to the traditional LER database, a DLER database has been generated using TROPOMI data. The DLER database is in addition a function of the TROPOMI viewing direction and provides generally higher values than the LER database, which does not take into account the directional dependence of the surface reflectance (Tilstra, 2022). Recalculating AMFs and tropospheric $NO_2$ VCDs with the regional CAMS $NO_2$ profiles and the TROPOMI DLER yields the IUP V02.03.01 REG DLER product, which again does not recalculate cloud parameters.

The different TROPOMI $NO_2$ products with their most important differences are summarized in Table 3.

**Table 3.** TROPOMI $NO_2$ product versions with the most important differences between the analyzed products.

| TROPOMI $NO_2$ product versions | $NO_2$ vertical profile | Reflectivity | Clouds | Comments, Availability |
|---|---|---|---|---|
| OFFL V01.03.02 | TM5 | OMI LER | FRESCO-S | operational 26 Jun 2019 - 29 Nov 2020 |
| OFFL V01.03.02 CAMS | CAMS regional < 3 km<br>CAMS global > 3 km | OMI LER | FRESCO-S | scientific, based on OFFL V01.03.02 |
| PAL V02.03.01 | TM5 | OMI LER | FRESCO-W | operational 4 Nov 2021 - 17 Jul 2022<br>as OFFL V02.03.01<br>reprocessed 1 May 2018 - 14 Nov 2021<br>as PAL V02.03.01 |
| IUP V02.03.01 | TM5 | OMI LER | FRESCO-W | scientific, similar to PAL V02.03.01,<br>a priori assumptions can be changed,<br>campaign period |
| IUP V02.03.01 REG | CAMS regional < 3 km<br>TM5 > 3 km | OMI LER | FRESCO-W | scientific, campaign period |
| IUP V02.03.01 REG TROPOMI LER | CAMS regional < 3 km<br>TM5 > 3 km | TROPOMI LER | FRESCO-W | scientific, campaign period |
| IUP V02.03.01 REG TROPOMI DLER | CAMS regional < 3 km<br>TM5 > 3 km | TROPOMI DLER | FRESCO-W | scientific, campaign period |

### 3.1.5 TROPOMI data set

In the present study, we evaluate the TROPOMI tropospheric $NO_2$ VCD product from 12 September to 18 September 2020 of the two described data products OFFL V01.03.02 and PAL V02.03.01, as well the described scientific data products.

Each TROPOMI pixel has a quality assurance value (qa_value) indicating the quality of the processing and retrieval result. Following the recommendation by Eskes and Eichmann (2022), we only use observations with a qa_value above 0.75 for all used TROPOMI data products. This removes problematic retrievals and observations with cloud radiance fractions of more than 50 %. Since the campaign measurement days were mostly cloud free, the cloud radiance fraction retrieved in the TROPOMI $NO_2$ spectral window, was on average $0.21 \pm 0.10$ with a maximum of 0.48 and thus all data can be used.

Large tropospheric $NO_2$ VCDs are observed in central Europe, e.g., over Paris, London, Milan, and Antwerp, with the largest

values of $1.6 \cdot 10^{16}\,\mathrm{molec\,cm^{-2}}$ in the campaign region in North Rhine-Westphalia (see Fig. 1). The campaign area is clearly distinguished from surrounding rural areas, which have low tropospheric $NO_2$ VCDs below approximately $3 \cdot 10^{15}\,\mathrm{molec\,cm^{-2}}$.

## 3.2 AirMAP

AirMAP, an airborne imaging spectrometer developed by the Institute of Environmental Physics in Bremen (IUP-Bremen), has been used in several campaigns for trace gas measurements and pollution mapping (Schönhardt et al., 2015; Meier et al., 2017; Tack et al., 2019; Merlaud et al., 2020). During the campaign, AirMAP was installed on a Cessna 207-Turbo, operated by the Freie Universität Berlin. AirMAP is a push-broom imaging DOAS instrument with the ability to create spatially continuous and nearly gap-free measurements. The scattered sunlight from below the aircraft is collected with a wide-angle entrance optic

resulting in an across track field of view of around $52°$. This leads to a swath width of approximately 3 km, about the same size as the flight altitude, during the campaign. With a sorted fiber bundle of 35 fibers, vertically stacked at the spectrometer entrance slit, orthogonally oriented to the flight direction, the radiation is coupled into the UV-Vis imaging grating spectrometer. The $400\,\mathrm{g\,mm^{-1}}$ grating, blazed at 400 nm provides measurements in the 429 - 492 nm wavelength range, with a spectral resolution between 0.9 nm and 1.6 nm full width at half maximum. The spectrometer is temperature stabilized at $35°\mathrm{C}$. The along-track

resolution depends on the speed of the aircraft (around $60\,\mathrm{m\,s^{-1}}$) and the exposure time (0.5 s). At a flight altitude of 3300 m, this results in a typical ground scene having a footprint of around 100 m x 30 m. More details about AirMAP can be found in Schönhardt et al. (2015), Meier et al. (2017) and Tack et al. (2019).

### 3.2.1 AirMAP data retrieval

For the $NO_2$ retrieval, the DOAS method is applied to the measured spectra in a fitting window of 438 - 490 nm. The $NO_2$

differential SCDs (dSCDs) are retrieved relative to in-flight-measured reference background spectra, which were measured over a region with small $NO_2$ concentrations during the same flight. The dSCD is converted into a tropospheric SCD ($SCD_{trop}$) by correcting for the amount of $NO_2$ in the reference background measurement ($SCD_{ref}$):

$$SCD_{trop} = dSCD + SCD_{ref} = dSCD + VCD_{trop,ref} \cdot AMF_{trop,ref} \tag{1}$$

For the conversion to the desired tropospheric VCD ($VCD_{trop}$), the $SCD_{trop}$ is divided by the tropospheric airmass factor

($AMF_{trop}$):

$$VCD_{trop} = \frac{SCD_{trop}}{AMF_{trop}} = \frac{dSCD + VCD_{trop,ref} \cdot AMF_{trop,ref}}{AMF_{trop}} \tag{2}$$

Since the AMF of the actual measurement ($AMF_{trop}$) and of the reference background measurement ($AMF_{trop,ref}$) are usually not the same, simply adding the $VCD_{trop,ref}$ would introduce additional uncertainties. To correct for the $NO_2$ in the reference spectrum ($SCD_{ref}$), we assume a tropospheric VCD of $1 \cdot 10^{15}\,\mathrm{molec\,cm^{-2}}$ over the reference background region, which is a

290 typical value during summer in Europe (Popp et al., 2012; Huijnen et al., 2010). This assumption can be supported by the car DOAS measurements, see Sect. 3.3.1. All measurements of the campaign were performed around noon close to the S5P

overpass. The maximum difference between the time of the reference background and the actual measurement is of around 3 h, which is the total measurement time. We assume that the effect of the changing solar zenith angle (SZA) and the diurnal variation of the stratospheric $NO_2$ concentration are small (Schreier et al., 2019), and a stratospheric correction of the data is therefore not necessary.

The AMF calculated using SCIATRAN estimates the relative light path length through the absorbing layer by accounting for the effects of sun and viewing geometry, surface reflectance, aerosols and the $NO_2$ profile assuming cloud free conditions. As only limited information about the $NO_2$ profile is available in the campaign area, and the profile shape is expected to vary strongly within each flight region every day, we assume a typical urban $NO_2$ profile, which is based on an old WRF-chem (Weather Research and Forecasting model coupled with Chemistry) run and scaled to a height of 1 km (see Fig. A2). This assumption is supported by typical boundary layer heights in the measurement area and time of approximately 1 km (ERA5 reanalysis, Hersbach et al. (2018)). Input parameters related to aerosols (single scattering albedo, asymmetry factor and aerosol optical thickness) were extracted from the AERONET station FZJ-JOYCE at the Jülich research center (Löhnert et al., 2015), which is the only known source providing local ground-based aerosol information in the campaign area. During the campaign measurement days, the daily averages of aerosol optical thickness (AOT) at 440 nm measured at FZJ-JOYCE ranged between 0.235 and 0.398 with a mean value of 0.285. This information is spatially constrained, and the situation can differ during the flights in the Duisburg and Cologne area. A sensitivity study using AMFs for a range of AOTs between 0.003 and 0.6 for the AirMAP $NO_2$ VCD retrieval demonstrated that the influence on the AirMAP tropospheric $NO_2$ VCD data set is small (< 1 %, comparing AirMAP tropospheric $NO_2$ VCDs assuming AOTs of 0.003 and 0.6). TROPOMI and AirMAP tropospheric $NO_2$ VCD scatter plots for AOTs of 0.003, 0.3 and 0.6 can be found in the Appendix Fig. A3. Considering the mean AOT of 0.285 from the AERONET station and the results from the sensitivity study, the AirMAP data set was retrieved using an AOT of 0.3 for all measurement days. In following discussions we are also considering the pre-operational TROPOMI AOT product (de Graaf, 2022), which can provide a larger picture of the aerosol situation (see Fig. A4). In general it is showing AOT values in the same range as investigated within the sensitivity study.

Bright surfaces enhance the relative contribution of light reflected from the surface to the signal received by the airborne instrument, increasing the sensitivity to $NO_2$ near the ground. Therefore, areas of high surface reflectance in the fitting window generally show larger dSCDs for the same amount of $NO_2$. Thus, differences in the surface reflectivity must be accounted for in the AMF calculations. As far as we are aware, reflectance data, having a sufficient spatial resolution are not available for the region of our flight campaign. Therefore, we use the individual AirMAP recorded intensities together with a method, based on a reference area with a known surface reflectance taken from the ADAM database (A surface reflectance DAtabase for ESA's earth observation Missions, Prunet et al. (2013)) and a look-up table of AirMAP radiances. Detailed information about the derivation of the surface reflectance and also about the general conversion from dSCDs to tropospheric $NO_2$ VCDs can be found in Meier et al. (2017).

The total uncertainty on the tropospheric $NO_2$ VCD comprises error sources of the dSCD retrieval, the estimation of the $NO_2$ in the reference background spectrum and the AMF calculation. We follow the same approach for error estimation and thus the same assumptions, as made in Meier et al. (2017) and Tack et al. (2019).

The total uncertainty of the AirMAP tropospheric $NO_2$ VCD follows the error propagation of the three error sources given by:

$$\sigma_{VCD_{trop}} = \sqrt{\left(\frac{\sigma_{dSCD}}{AMF_{trop}}\right)^2 + \left(\frac{\sigma_{SCD_{trop,ref}}}{AMF_{trop}}\right)^2 \left(\frac{SCD_{trop}}{AMF_{trop^2}} \cdot \sigma_{AMF_{trop}}\right)^2} \tag{3}$$

The error from the dSCD retrieval is estimated from the fit residual and is a direct output of the DOAS retrieval algorithm. Since no direct measurements of the $NO_2$ column in the reference ground scene exist, we assume a systematic error with an uncertainty of 100 % on the estimated value of $1 \cdot 10^{15}$ molec cm$^{-2}$. The error resulting from the AMF determination depends in large part on the values of the uncertainty attributed to the surface reflectance, the accuracy of the $NO_2$ vertical profile, and the aerosol optical depth as a function of altitude and location. Following Meier et al. (2017), the total error on the AMF is estimated to be smaller than 26 %. Taking the mean dSCD value ($1.2 \cdot 10^{16}$ molec cm$^{-2}$) and the mean dSCD error ($2 \cdot 10^{15}$ molec cm$^{-2}$) as typical values, the total error of the tropospheric $NO_2$ VCD is ~35 %. More details on error contributions can be found in Meier et al. (2017).

### 3.2.2 AirMAP campaign data set

Figure 3 shows a timeseries of tropospheric $NO_2$ VCDs measured by AirMAP for each of the seven flight days of the campaign. The mean over the 35 viewing directions is shown in dark colors and their standard deviation in light colors. The colors red, blue and green represent the respective research flight areas around Cologne, Jülich and Duisburg. The S5P overpass times with respective VZA and the times of the AirMAP reference background measurement are marked by the vertical dashed lines. Two flights were performed in the research flight area around Cologne (red), two flights in the Jülich area (blue) and three flights in the Duisburg area (green). The first two flights, shown in Fig. 3 are weekend days, a Saturday, and a Sunday. The columns show strong variability between the three target areas and from day to day with the highest tropospheric $NO_2$ VCDs being ~ $5 \cdot 10^{16}$ molec cm$^{-2}$ over the Duisburg area on Monday 14 September and Tuesday 15 September 2020 and much lower values for both flights in the Cologne area, having tropospheric $NO_2$ VCDs of up to $2.5 \cdot 10^{16}$ molec cm$^{-2}$. Maps of the tropospheric $NO_2$ VCD for each flight are displayed in Fig. 4.

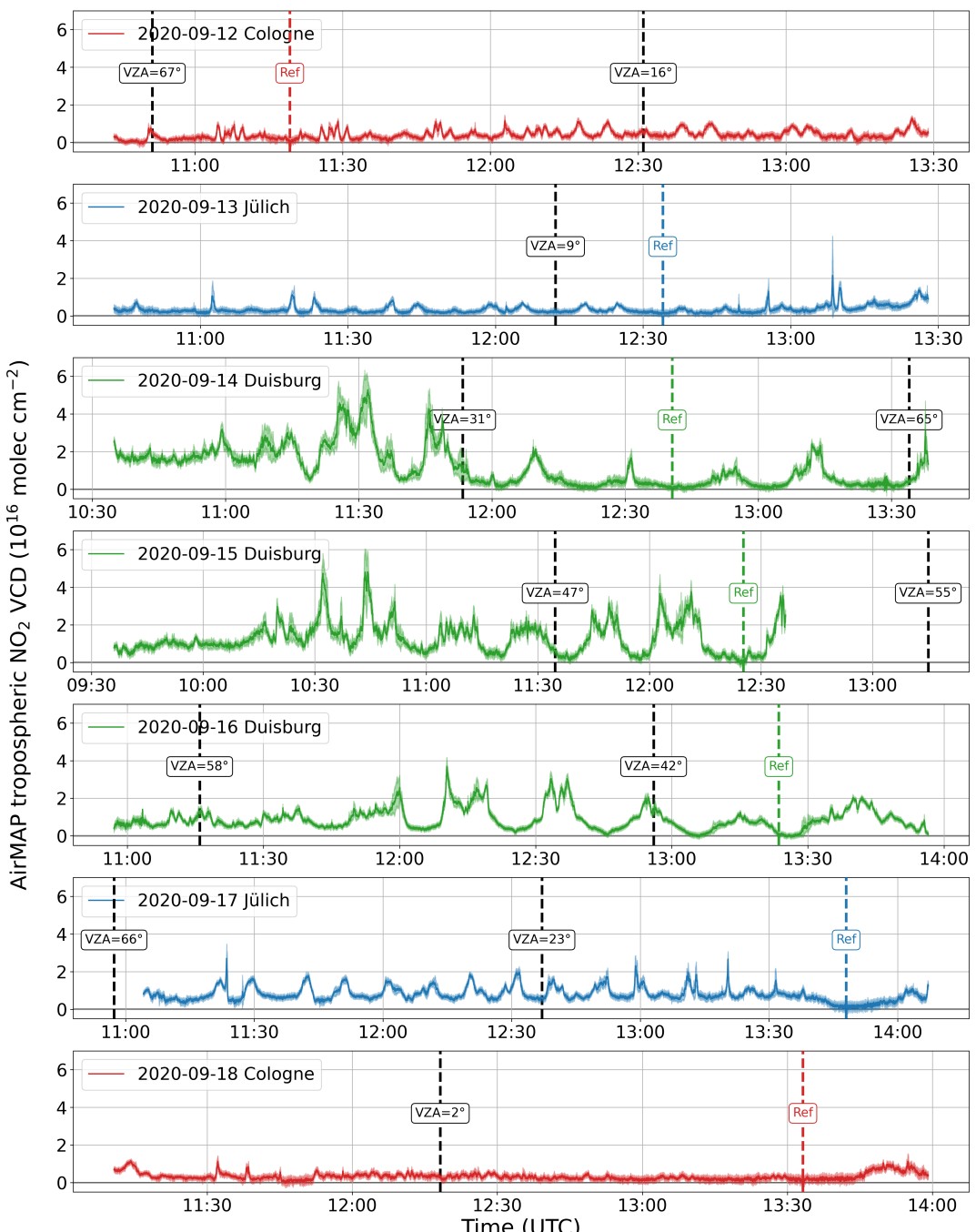

**Figure 3.** Plots of the AirMAP timeseries of tropospheric $NO_2$ VCD (mean over the 35 viewing directions with standard deviation as dark line and bright area, respectively) for the seven flight days from Saturday 12 September 2020 – Friday 18 September 2020. These show strong variability from day to day (weekday vs weekend) and between the three target areas (Cologne, Jülich, Duisburg). The dashed black vertical lines indicate S5P overpass times with their viewing zenith angle. The dashed colored vertical lines indicate the times of the AirMAP reference measurement.

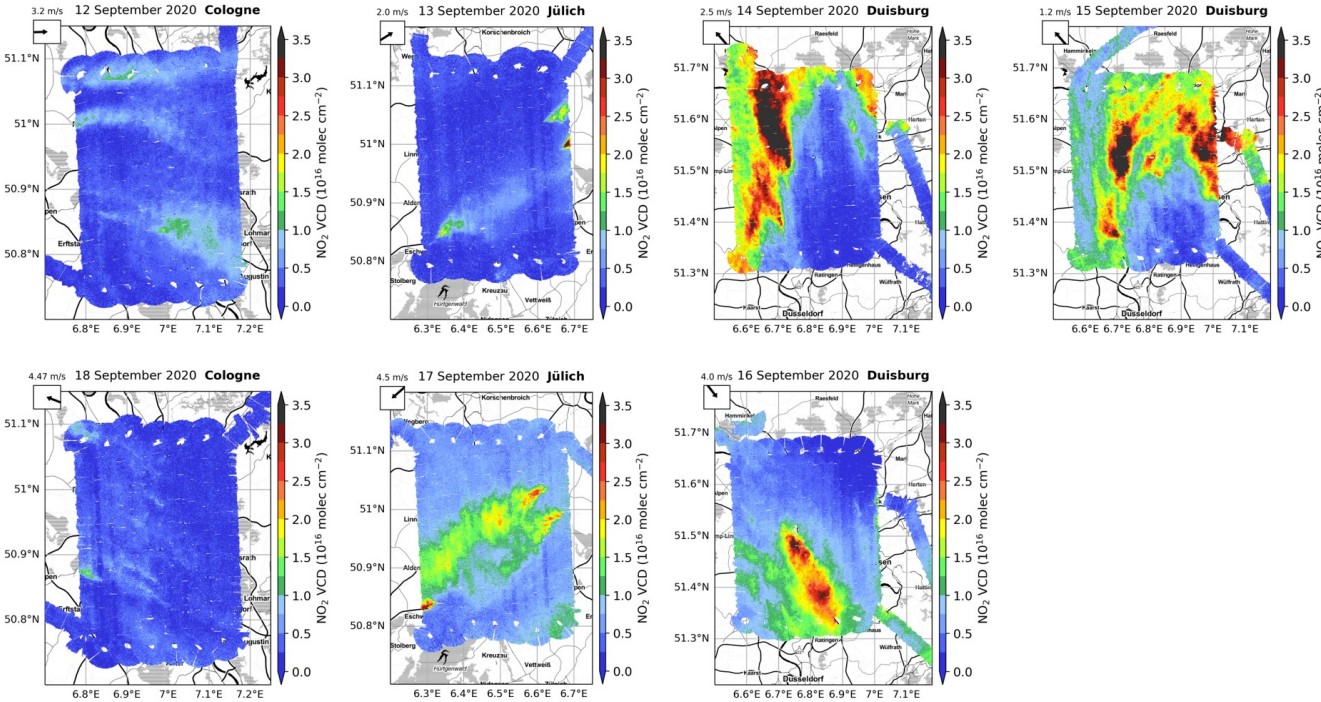

**Figure 4.** Maps of VCD NO$_2$ from AirMAP flights from 12 September to 18 September 2020. Two flights in the research flight area around Cologne (left column), two flights in the flight area around Jülich (second column) and three flights in the flight area around Duisburg (third and fourth column). The mean wind direction and speed in the flight area, determined from ERA5 10 m wind data for the middle of the flight, are given in the top left corner.

Jülich research flight area: The tropospheric NO$_2$ VCD over the Jülich flight area is smaller during the flight on Sunday 13
September than on Thursday 17 September, where several peaks in the NO$_2$ VCD up to $2.5 \cdot 10^{16} \, \mathrm{molec \, cm^{-2}}$ are visible. These peaks are caused by plumes of NO$_2$ coming from three large power plants, located in the Jülich research flight area, which are clearly visible in the maps of the AirMAP NO$_2$ VCD in Fig. 4. Two power plants are located in the Northeast and one in the Southwest of the Jülich flight area. The plumes, which have enhanced tropospheric NO$_2$ VCDs compared to low background VCDs outside of the plume, are blown in the mean wind direction (shown in the top left corner of the maps) determined from
ERA5 10 m wind data (Hersbach et al., 2018) for the flight area and in the middle of the flight time. Differences between the two measurement days over the Jülich flight area are related to wind conditions potentially enhanced by a weekend effect. On Sunday 13 September, there was a weak wind coming from the Southwest blowing the plumes to the Northeast, thus two out of three plumes were mostly outside the flight area and cleaner air from a rural area was prevalent. On Thursday 17 September, a stronger wind coming from the opposite direction, the Northeast, was blowing the plumes to the Southwest.
Duisburg research flight area: The three maps from flights over the Duisburg flight area show the strong NO$_x$ emissions from power plants and the industrial area in Duisburg with plumes oriented depending on wind direction.

Cologne research flight area: The two AirMAP flights in the Cologne area show only slightly enhanced $NO_2$ amounts compared to the background tropospheric $NO_2$ VCD on both days.

## 3.3 Car DOAS instruments

During the S5P-VAL-DE-Ruhr campaign, mobile car DOAS measurements were performed by three institutions, the IUP-Bremen, the Max Planck Institute for Chemistry in Mainz (MPIC) and the Royal Belgian Institute for Space Aeronomy (BIRA). More information about the car DOAS instruments can be found in Schreier et al. (2019), Donner (2016), and Merlaud (2013). The measurement elevation angle was for the majority of measurements in zenith-sky with some off-zenith measurements. The off-zenith measurements are used in the estimation of the $NO_2$ SCD in the reference spectrum and the stratospheric $NO_2$

contribution for the BIRA and MPIC car DOAS measurements. The focus on zenith-sky measurements during driving has the advantage of a stable viewing direction when the direction of travel changes, variations from relative azimuth changes are avoided and measurements cannot be blocked by buildings, which can be a large problem in cities. In addition, the highest horizontal resolution is achieved with this viewing geometry.

### 3.3.1 IUP car DOAS instrument and data retrieval

The IUP car DOAS instrument uses an experimental setup, which comprises an Avantes spectrometer and a light fiber with a fixed viewing direction to the zenith measuring scattered sun light in the UV-Vis range (see also Schreier et al., 2019). Collected spectra are averaged over 10 s, which corresponds to travelled distances of around 80 - 300 m, depending on the driving speed. The DOAS method is applied to the measured spectra in a fitting window of 425 - 490 nm. The tropospheric $NO_2$ VCD from car DOAS zenith-sky measurements is determined in a similar manner to that used for the AirMAP measurements by the

following equation:

$$\mathrm{VCD_{trop}} = \frac{\mathrm{dSCD} + \mathrm{SCD_{ref}} - \mathrm{VCD_{strat}} \cdot \mathrm{AMF_{strat}}}{\mathrm{AMF_{trop}}}$$

$$= \frac{\mathrm{dSCD} + \mathrm{VCD_{trop,\,ref}} \cdot \mathrm{AMF_{trop,\,ref}} + \mathrm{VCD_{strat,\,ref}} \cdot \mathrm{AMF_{strat,\,ref}} - \mathrm{VCD_{strat}} \cdot \mathrm{AMF_{strat}}}{\mathrm{AMF_{trop}}} \tag{4}$$

The dSCD are retrieved relative to reference background spectra, measured in a region with small $NO_2$ concentrations on 13 September around noon. The $\mathrm{SCD_{ref}}$ cannot be measured directly. Similar to the AirMAP VCD determination, the $NO_2$ in the reference background spectrum is corrected for by assuming a tropospheric $NO_2$ VCD of $1 \cdot 10^{15}\,\mathrm{molec\,cm^{-2}}$ over the

reference background region. The other car DOAS instruments do not rely on this value as they use dedicated measurements taken at lower elevation angle to directly estimate the tropospheric column in the reference measurement. Thus, the assumption of a $\mathrm{VCD_{trop,\,ref}}$ of $1 \cdot 10^{15}\,\mathrm{molec\,cm^{-2}}$ can be supported by a comparison of collocated car DOAS measurements of the three instruments, which shows a very good agreement (see Fig. A5). Using a larger $\mathrm{VCD_{trop,\,ref}}$ in the IUP car DOAS retrieval would increase the offset compared to the MPIC and BIRA car DOAS data. Since we used a fixed reference background

measurement for all car DOAS measurement days, a stratospheric correction based on the Bremen 3d chemistry transport

model (B3dCTM, Hilboll et al. (2013b)), providing a daily diurnal cycle of the stratospheric $NO_2$ VCDs, scaled to TROPOMI stratospheric VCDs in the measurement area is applied to the car DOAS data. Stratospheric AMFs are calculated with the radiative transfer model SCIATRAN (Rozanov et al., 2014) as function of the SZA. For the conversion of tropospheric SCDs to tropospheric $NO_2$ VCDs, a constant tropospheric AMF of 1.3 was used. The AMF of 1.3 for an elevation angle of 90° is

closer to the true AMF (derived from radiative transfer simulations) than the geometric approximation for the tropospheric AMF of 1 (Shaiganfar et al., 2011; Merlaud, 2013; Schreier et al., 2019). Merlaud (2013) analyzed the AMF distribution for a large number of simulations, resulting in a mean of $1.33 \pm 0.2$ for measurements in 90° viewing zenith angle. Since we only analyze data close to the AirMAP overpass, which was performing measurements around noon, the SZA is not varying much. Following the mentioned studies we assume an uncertainty of 20 % for the AMF.

### 3.3.2   MPIC car DOAS instrument and data retrieval


The MPIC car DOAS instrument uses an Avantes spectrometer with an active temperature stabilization and takes in addition to the zenith-sky measurements also off-axis measurements at 22° elevation (see also Donner, 2016). During the validation measurement period, only zenith-sky measurements were used to increase spatial and temporal coverage. The integration time was 30 s. Before and after the validation measurements, the elevation angles alternate between 22° elevation and zenith-sky

(90°). The combination of both angles allows the determination of the absorption in the reference spectrum $SCD_{ref}$, as well as the absorption in the stratosphere. The DOAS analysis is performed in a wavelength interval of 400 - 460 nm using a daily fixed reference background at 90° elevation, at low SZA in a region with small $NO_2$ concentrations. $NO_2$ dSCDs retrieved from the DOAS analysis are converted to tropospheric $NO_2$ VCDs by using Eq. 4 (see also Wagner et al., 2010; Ibrahim et al., 2010). Radiative transfer model calculations for $NO_2$ box profiles of 500 m or 1000 m and moderate aerosol loads provide on average

tropospheric AMFs of 3 and 1.3 with an assumed uncertainty of 20 % for the 22° and 90° elevation angle measurements, respectively (Shaiganfar et al., 2011; Merlaud, 2013).

### 3.3.3   BIRA car DOAS instrument and data retrieval

The BIRA car DOAS instrument consists of two Avantes spectrometers measuring simultaneously scattered light in 90° and 30° elevation (see also Merlaud, 2013). Individual spectra are co-added, and the DOAS analysis is performed in a wavelength

interval of 450 - 515 nm on spectra averaged every 30 s using a single pair of time-coincident low SZA zenith reference spectra for all measurement days. The measurements on both channels being simultaneous, the retrieval of tropospheric $NO_2$ VCDs follows the MAX-DOAS principle (see Eq. 5), using the differences in dSCDs and AMFs for two elevation angles. For the AMFs, a sun position-dependent look-up table (LUT) is used. This LUT was calculated using DISORT and provides AMFs of 2.5 and 1.3 for the 30° and 90° elevation angle measurements, respectively (Merlaud, 2013). An additional zenith-DOAS

instrument was operated for $SO_2$ measurements, results are not shown in this study.

### 3.3.4 Car DOAS campaign data set

For the verification of the car measurements, regular collocations of the cars were used at selected meeting points and over-lapping measurement routes. Fig. A5 in the Appendix shows a scatter plot of the collocated car DOAS measurements, demon-strating a good agreement between the three instruments. In general, the car DOAS measurements were planned in a way that each car made measurements during a round trip of a large part of the research flight area. The routes were also chosen to pass by the ground-based measurement stations. The duration of the car measurements was typically around 4 h per day. This enabled measurements to be made during the complete AirMAP flight and the S5P overpass times to gather many and closely collocated measurements. Several round trips, about three to four, were performed, dependent on traffic conditions. In addition to spatial variations of $NO_2$ also temporal changes are observed.

Figure 5 shows maps of car DOAS tropospheric $NO_2$ VCDs for the seven days in the research flight areas around Cologne, Jülich, and Duisburg. Measurements are within $\pm 1$ h of the S5P overpass time given in the map title. As already seen in the

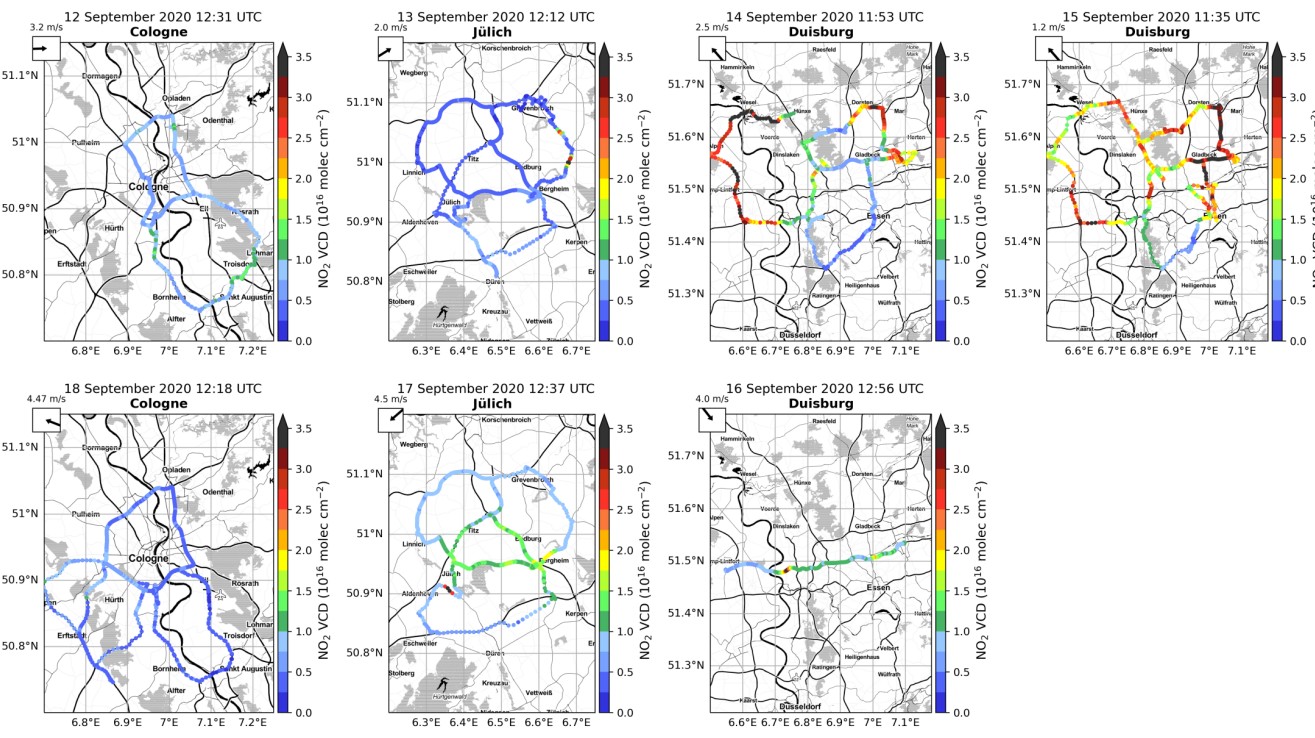

**Figure 5.** Maps of tropospheric $NO_2$ VCDs from car DOAS measurements from 12 September to 18 September 2020 in the research flight areas around Cologne, Jülich, and Duisburg. Measurements are within $\pm 1$ h of the S5P overpass time given in the title.

AirMAP data, strong variability between the three target areas is observed. The highest amounts of $NO_2$ are visible around Duisburg with high spatial variability within the target area. The lowest amounts of $NO_2$ are found in the area around Cologne, which confirms the findings of the AirMAP measurements. The car DOAS measurements in the Jülich area show enhanced

NO$_2$ values where the AirMAP measurements also see the plumes of the two power plants located in the Northeast of the flight area.

### 3.4  Ground-based instruments

During the campaign period, six ground-based instruments, two zenith-sky DOAS, two MAX-DOAS and two Pandora instruments were measuring in the three target areas. The instrument locations are marked in the map of the TROPOMI tropospheric
NO$_2$ VCDs in Fig. 1 and the flight overview map in Fig. 2.

#### 3.4.1  Zenith-sky DOAS

Two zenith-sky DOAS instruments were deployed and operated within the Ruhr area for several months. The instruments use an experimental setup, which comprises an Avantes spectrometer (290 - 550 nm) and a light fiber with a fixed viewing direction to the zenith measuring scattered sun light in the UV-Vis spectral range (similar as in Schreier et al., 2019). One
instrument is located at the Jülich research center next to the Pandora (Zenith-DOAS JUE) and the second at a local residence in Gelsenkirchen (Zenith-DOAS GEL), in the Duisburg research flight area. The tropospheric NO$_2$ VCDs are estimated from the dSCDs resulting from the DOAS fit using Eq. 4. For the reference background spectra in the DOAS fit, we use a fixed spectrum taken in summer on a clean day around noon. The amount of NO$_2$ in the reference background spectrum, SCD$_{ref}$, is determined from the long time series using the lowest measured NO$_2$. For the measurements made by the Zenith-DOAS GEL, this is a
SCD$_{ref}$ of $1.7 \cdot 10^{16}$ molec cm$^{-2}$. For the Zenith-DOAS JUE, the SCD$_{ref}$ is determined as $1.0 \cdot 10^{16}$ molec cm$^{-2}$ using the same approach. The SCDs$_{ref}$ given here include the stratospheric and tropospheric NO$_2$ in the reference background spectrum. Since the reference measurements were taken during summer a relatively large part is stratospheric NO$_2$. An uncertainty of 30 % for the SCD in the reference spectrum is assumed. The VCD$_{strat}$ is estimated from twilight Langley fits (e.g. Constantin et al., 2013) with an uncertainty of $2 \cdot 10^{14}$ molec cm$^{-2}$, and the stratospheric AMFs are obtained from SCIATRAN calculations. For
the tropospheric AMF we use the same value of 1.3 as for the car DOAS. Since we only analyze the measurements close to the AirMAP overpass, i.e. around noon, the SZA does not vary much and the influence on the AMF is small (see Sect. 3.3.1).

#### 3.4.2  MAX-DOAS measurement truck

From 7 September to 19 September 2020, the IUP Bremen measurement truck performed MAX-DOAS measurements in the harbor area of Duisburg close to the Rhine River (MAX-DOAS DUI). This MAX-DOAS instrument uses a UV spectrometer
(282 - 412 nm) with a light fiber connected to a telescope on a pan-tilt head and was scanning in multiple elevation angles. The tropospheric NO$_2$ VCDs are estimated from the dSCD measurements in 30° elevation angle with a sequential zenith sky reference spectrum (interpolated from the zenith sky measurements shortly before and after the off-axis measurement):

$$\text{VCD}_{\text{trop}} = \frac{\text{dSCD}(30°)}{\text{AMF}_{\text{trop}}(30°) - \text{AMF}_{\text{trop}}(90°)} \tag{5}$$

Based on SCIATRAN AMF calculations for a wavelength of 350 nm, adjusted to the ground-based and AirMAP comparison
times around noon regarding SZA and with typical albedo and AOT values found during the campaign measurement days,

AMFs of 2.5 and 1.4 are used for elevation angles of 30° and 90°, respectively. The total uncertainty of the tropospheric $NO_2$ VCD originates from uncertainties in the retrieved dSCD, which results mainly as the error of the DOAS fit, and uncertainties from the AMF for which we assume 20 %.

### 3.4.3 BIRA SkySpec MAX-DOAS

A further MAX-DOAS instrument was setup at the airport Schwarze Heide in Dinslaken (MAX-DOAS AIRPT) from 3 August 2020 to 29 September 2020. The instrument, deployed by BIRA, was an Airyx Compact SkySpec MAX-DOAS, based on an Avantes spectrometer (300 - 463 nm). A scanning prism in elevation direction can rotate 180° enabling elevation scan measurements in two azimuthal directions (Airyx GmbH, 2022; Kreher et al., 2020). At the airport, the instrument was scanning in azimuths of 132° and 312° and in multiple elevation angles. In this study, only measurements in north-westerly direction
(312°) are used for the analysis.

The tropospheric $NO_2$ VCDs are retrieved by applying the Mexican MAX-DOAS Fit (MMF, Friedrich et al. (2019)) inversion algorithm using dSCDs retrieved with the spectral fitting software QDOAS (Danckaert et al., 2017) using the FRM4DOAS settings and setup (Hendrick et al., 2016). The tropospheric $NO_2$ VCD error is calculated from the covariance smoothing error matrix, the covariance measurement noise error matrix and a systematic error as a fixed fraction of the VCD, based on the
systematic uncertainty of the cross section, for $NO_2$ as 3 % (Vandaele et al., 1998).

### 3.4.4 Pandora

The Pandora instrument is a ground-based UV-Vis spectrometer that provides direct Sun total column and sky scan MAX-DOAS tropospheric column observations, comprising an Avantes spectrometer (270 - 520 nm) (e.g. Herman et al., 2009; Kreher et al., 2020; Verhoelst et al., 2021). Two Pandoras are deployed and operated in the campaign area to provide long term
measurements. They were installed in August 2019 and are still in operation in 2022. One Pandora is located at the Jülich research center (Pandora JUE) and a second is located in Cologne, district Deutz (Pandora COL). Locations are marked in Fig. 1. All data are processed as part of the Pandonia Global Network (PGN, https://www.pandonia-global-network.org/, last access: 18 March 2022). Tropospheric $NO_2$ VCDs are retrieved using coincident sky scan MAX-DOAS and direct-sun observations and are calculated based on the Spinei et al. (2014) approach (Cede et al., 2021). $NO_2$ values are given, together with the
respective uncertainty (Cede et al., 2021), as tropospheric $NO_2$ VCD. The analyzed data are labeled with quality flags, which indicate whether the data quality is high, medium or low, whether the data are quality assured and usable or not. Only data with a quality flag accounting for high and medium quality (assured as well as not assured) are used.

## 4 Evaluating airborne tropospheric $NO_2$ VCD with stationary ground-based data

The data set of the stationary ground-based instruments, deployed at different sites in the three selected flight areas, is used
to evaluate the AirMAP tropospheric $NO_2$ VCD. This, together with the mobile measurements, provides a basis for using the AirMAP data for the evaluation of the TROPOMI tropospheric $NO_2$ VCD. During the campaign, AirMAP overflights were

conducted for all ground-based measurement stations.

A scatter plot of all coincident measurements is shown in Fig. 6. Each point is colored according to its instrument type and location. The shown AirMAP tropospheric $NO_2$ VCDs, are averages of the measurements from an area of 500 m x 500 m around the ground-based measurement station. This is then assigned to the selected ground-based stationary measurements, which are averaged in time intervals of 20 min around the AirMAP overpass time. In total 25 coincident measurements were obtained by this procedure. Error bars of Fig. 6 represent the error in the tropospheric $NO_2$ VCD retrieval, averaged within the 500 m x 500 m grid boxes and 20 min time intervals. Fitting of the data was done with orthogonal distance regression, as for all following data shown in the present study. The AirMAP and ground-based tropospheric $NO_2$ VCDs are highly correlated (Pearson correlation coefficient r = 0.88) with a slope and standard deviation of $0.90 \pm 0.09$ and an offset of $1.16 \pm 0.15 \cdot 10^{15}$ molec cm$^{-2}$. Overall, the data show good agreement with a tendency of slightly larger values from the ground-based instruments as compared to the airborne data. Part of the scatter and deviation may result from the different retrieval algorithms with different assumptions on radiative transfer, aerosols and reference background spectra. Additionally, spatiotemporal variability of $NO_2$ is influencing the agreement of the comparison. Figure A6 in the Appendix shows the same as Fig. 6, but error bars represent the 10th and 90th percentile within the 500 m x 500 m grid boxes and 15 min time intervals to illustrate the spatiotemporal variability within the comparison criteria.

## 5 Evaluating airborne tropospheric $NO_2$ VCD with car DOAS data

The mobile car DOAS measurements performed by IUP, MPIC and BIRA were synchronized to the AirMAP measurements. They were measuring during the complete flight in the same area as the AirMAP instrument to gather many closely collocated measurements between the instruments. The data are used, in addition to the stationary ground-based measurements, to evaluate the tropospheric $NO_2$ VCD maps retrieved from AirMAP. Compared to the stationary data, the car measurements have the advantage that they can cover larger and more diverse areas and thus potentially also a wider range of $NO_2$ values. As a result of having more opportunities to make near simultaneous synchronized measurements, consequently, a larger number of collocated measurements can be compared. For the comparison, the car DOAS measurements are averaged in time intervals of 15 min and gridded in areas of 500 m x 500 m. The same grid is applied to the AirMAP measurements and a comparison of measurements in the same grid box and time interval is performed.

A scatter plot of all coincident car DOAS and AirMAP measurements fulfilling a time criterion of $\pm$ 15 min is shown in Fig. 7. Each point is colored by the respective car DOAS instrument. In total, 572 pairs of coincident measurements are considered. Error bars of Fig. 7 represent the error in the tropospheric $NO_2$ VCD retrieval, averaged within the 500 m x 500 m grid boxes and 15 min time intervals. The comparison shows an offset of $-1.29 \pm 0.15 \cdot 10^{15}$ molec cm$^{-2}$. This offset could be adjusted to be closer to zero by increasing the estimated $VCD_{trop, ref}$ in the AirMAP retrieval by more than a factor of 2. However, the offset in the comparison of AirMAP and ground-based stationary data of $1.16 \pm 0.15 \cdot 10^{15}$ molec cm$^{-2}$. is positive instead of negative, and a larger $VCD_{trop, ref}$ in the AirMAP retrieval would further increase this offset. Because of this, and a lack of justification for a large difference between the $VCD_{trop, ref}$ for the car and AirMAP retrieval, we chose to leave the $VCD_{trop, ref}$

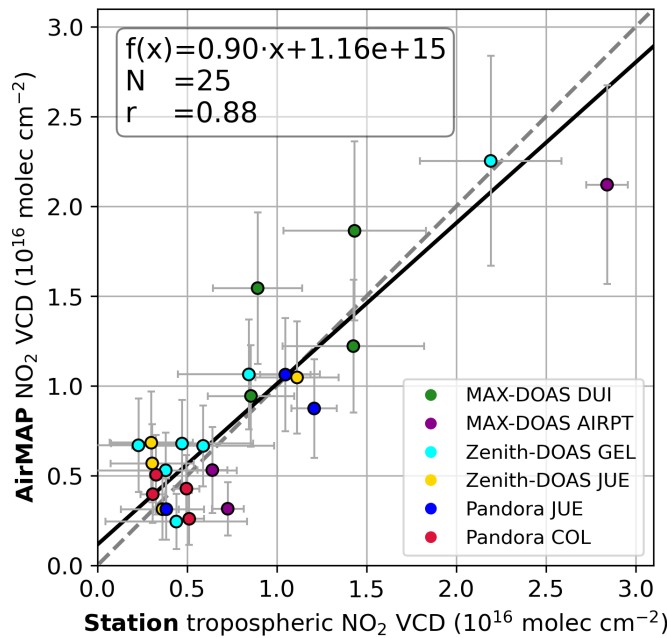

**Figure 6.** Scatter plot of AirMAP data against the stationary ground-based NO$_2$ VCDs averaged over a time interval of 20 min closest to the AirMAP overpass data, which are averaged over a 500 m x 500 m area around the station site. Each point is colored according to its ground-based instrument type and location. Error bars represent the error in the tropospheric NO$_2$ VCD retrieval, averaged within the 500 m x 500 m grid boxes and 20 min time intervals. The 1:1 line is indicated by the grey dashed line. The solid black line represents the orthogonal distance regression.

as it is. Nevertheless, it is clear that the validation of the offset has a large relative uncertainty as there may be offsets in the reference measurements. Besides that Fig. 7 shows a good correlation between the airborne and car DOAS instruments, with a correlation coefficient of r = 0.89. The orthogonal distance regression reveals a slope of $0.89 \pm 0.02$, i.e. close to unity. Considering tropospheric NO$_2$ VCD retrieval errors, that the data retrieved from the different instruments used for this comparison were analyzed independently by the different groups and retrieval methods are only partly harmonized, with

different assumptions about the radiative transfer, aerosols and reference background spectra, the data show good agreement. Coincident measurements that are furthest from the 1:1 line are mostly cases where the time difference was at the outer edge of the time filter criterion and may therefore be caused by the rapid natural variability of NO$_2$ (see right plot in Fig. A7). Figure A7 shows the same as Fig. 7 and an additional plot where points are color coded by time difference, but error bars represent the 10th and 90th percentile within the 500 m x 500 m grid boxes and 15 min time intervals to illustrate the spatiotemporal

variability.

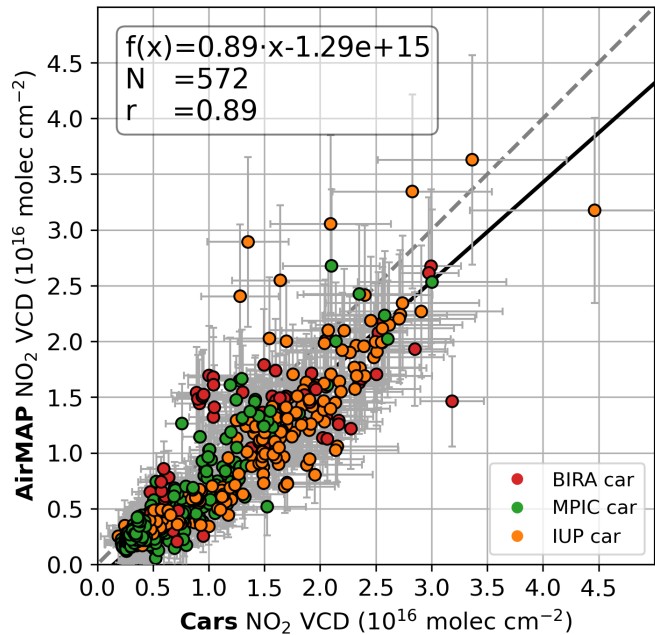

**Figure 7.** Scatter plot between collocated car DOAS ($\pm$ 15 min window from the aircraft overpass) and AirMAP NO$_2$ VCDs using grid boxes of 500 m x 500 m and 15 min time intervals. The data points from BIRA, MPIC and IUP car DOAS instruments are color coded red, green and orange. The 1:1 line is indicated by the grey dashed line. The thick solid black line represents the orthogonal distance regression. Error bars represent the error in the tropospheric NO$_2$ VCD retrieval, averaged within the 500 m x 500 m grid boxes and 15 min time intervals.

## 6  Evaluating TROPOMI tropospheric NO$_2$ VCD with AirMAP tropospheric NO$_2$ VCD data

The good agreement of the ground-based stationary and car DOAS data set with the AirMAP data, gives confidence for using the AirMAP tropospheric NO$_2$ VCD data set to evaluate the TROPOMI products. Airborne observations are valuable for the evaluation of TROPOMI data, as a large number of satellite pixels are mapped in relatively short time. The AirMAP measurement time per flight is in the order of three hours, with measurements over the target area planned to be taken at least $\pm$ 1 h around the S5P overpass with the smallest VZA, c.f. Fig. 3. In the comparison TROPOMI pixels are only considered, when they are at least 75 % mapped by AirMAP pixels. AirMAP data are considered when they match the temporal coincidence criteria of $\pm$ 30 min around the S5P overpass time. These spatial and temporal coincident criteria are following the suggestion by Judd et al. (2020). During the seven flight days (for which TROPOMI data are only available on six days, due to ground-segment anomalies), AirMAP measurements coincide with 117 TROPOMI pixels. For the comparison of the two data sets, the AirMAP measurements are averaged within the TROPOMI pixel. Figure 8 shows the six daily TROPOMI PAL V02.03.01 and AirMAP tropospheric NO$_2$ VCDs maps over the designated flight area as well as the AirMAP measurements scaled to the coincident TROPOMI pixel.

The averaged AirMAP tropospheric NO$_2$ VCDs are compared to the coincident satellite data for three different TROPOMI

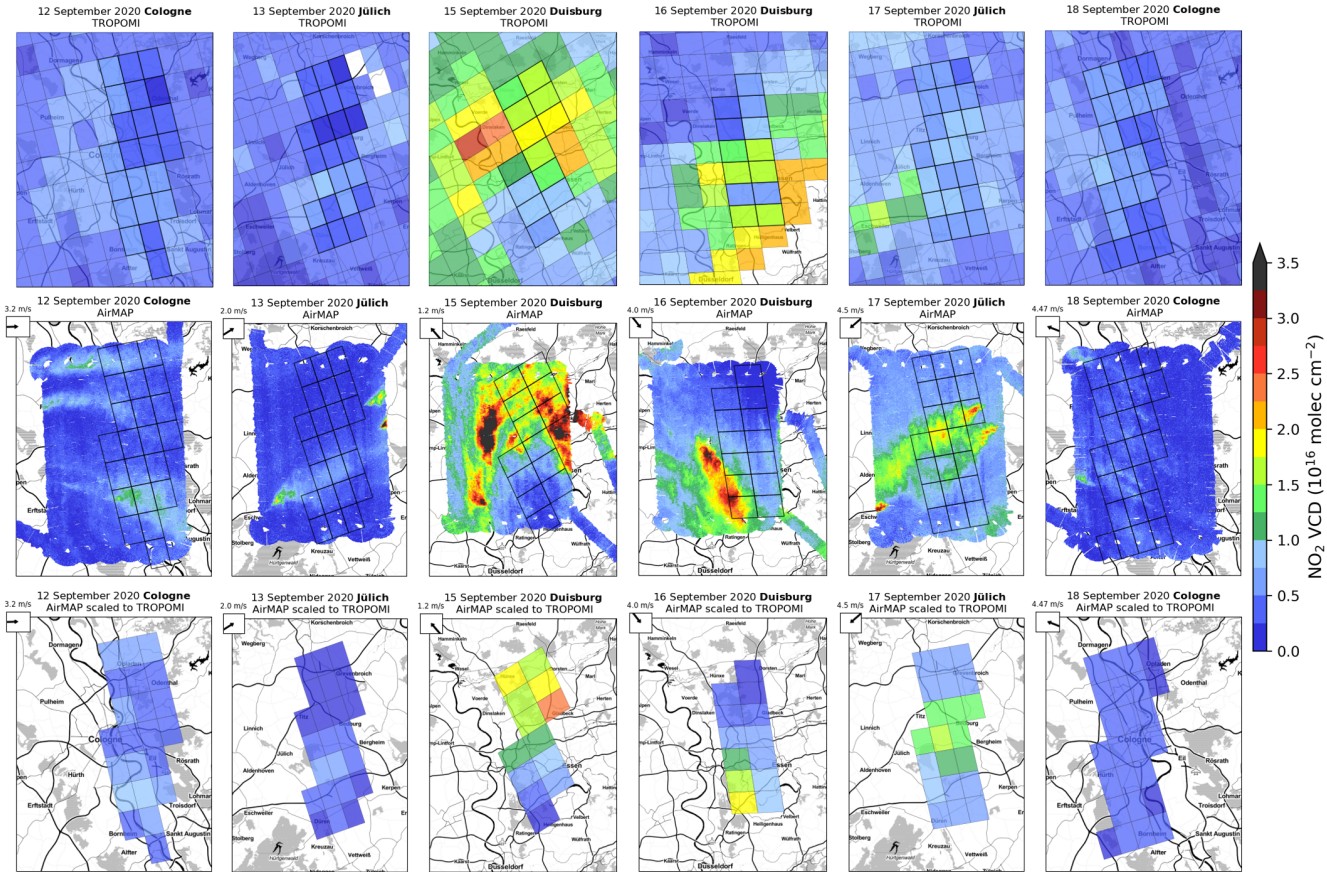

**Figure 8.** Daily maps of tropospheric $NO_2$ VCDs demonstrating how AirMAP data are matched to TROPOMI measurements. (top) TROPOMI PAL V02.03.01 tropospheric $NO_2$ VCDs where qa_value > 0.75. (middle) AirMAP tropospheric $NO_2$ VCDs with overlaid TROPOMI pixel outlines which are fulfilling the collocation criteria of a coverage of at least 75 % and AirMAP measurements $\pm$ 30 min around the S5P overpass. (bottom) AirMAP tropospheric $NO_2$ VCDs scaled to the TROPOMI pixel.

$NO_2$ data versions in Fig. 9. It shows scatter plots with an orthogonal distance regression analysis of the TROPOMI and AirMAP $NO_2$ VCDs for (a) the TROPOMI operational OFFL V01.03.02 data, (b) the adapted scientific TROPOMI V01.03.02 CAMS data using CAMS-based $NO_2$ profiles, and (c) the reprocessed data version PAL V02.03.01. Details on the different data versions are summarized in Table 3.

   The horizontal error bars correspond to the 10th and 90th percentiles of all airborne measurements within the respective
TROPOMI pixel. Vertical error bars represent the reported precision of the TROPOMI tropospheric $NO_2$ VCD. Error bars are shown only for these three examples to illustrate their magnitude and are not shown in the following plots for a better visibility of the data. An investigation of the different available TROPOMI $NO_2$ data versions compared to the AirMAP data with their different behavior (scatter, bias) gives further insight into the influence of different a priori assumptions made within each

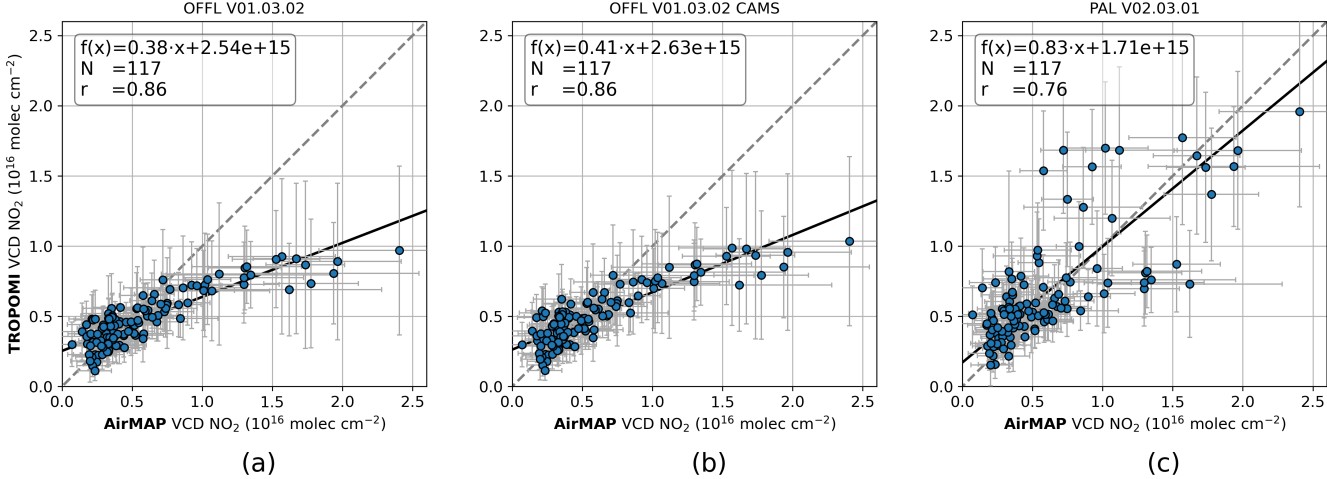

**Figure 9.** Scatter plots of TROPOMI NO$_2$ VCDs versus collocated AirMAP NO$_2$ VCDs for different versions of TROPOMI data: (a) operational OFFL V01.03.02, (b) V01.03.02 based on the CAMS NO$_2$ profiles, (c) PAL V02.03.01. Collocation criteria for AirMAP: ± 30 min around S5P overpass, gridded to the TROPOMI pixels and covering them at least to 75 %. The horizontal error bars represent the 10th and 90th percentiles of airborne measurements within the TROPOMI pixel. Vertical error bars show the reported precision of the TROPOMI tropospheric NO$_2$ VCD. Error bars on the TROPOMI measurements are shown to illustrate their magnitude and are not shown for all further plots for better visibility of the data.

retrieval.

Figure 9a shows coincidences between the TROPOMI operational OFFL V01.03.02 data and the AirMAP data, with a high correlation coefficient of 0.86, a slope of $0.38 \pm 0.02$, an offset of $2.54 \pm 0.15 \cdot 10^{15}\,\mathrm{molec\,cm^{-2}}$ and a median relative difference of -9 % with an interquartile range of -28 % to +16 %. All statistics of the comparisons between the different TROPOMI tropospheric NO$_2$ VCDs data versions and the AirMAP measurements are summarized in Table A1 in the Appendix. Figure A11 shows box-and-whisker plots summarizing the bias and spread of the difference between the TROPOMI versions and

AirMAP tropospheric NO$_2$ VCDs. The regression parameters and their standard errors are calculated for the plotted data points. Taking the uncertainties of the data points into account and considering the parameters of the orthogonal distance regression over the complete range of these uncertainties yields a standard deviation of 0.14 for the slope and $0.39 \cdot 10^{15}\,\mathrm{molec\,cm^{-2}}$ for the offset. The slope of 0.38 is significantly lower than the 0.68 from comparisons of TROPOMI NO$_2$ OFFL V01.03.02 data and aircraft measurements in the New York City/Long Island Sound region reported by Judd et al. (2020) and the 0.82 from

comparisons of TROPOMI and APEX measurements over Brussels and Antwerp reported by Tack et al. (2021).

The scientific TROPOMI data V01.03.02 CAMS based on the OFFL data V01.03.02 has the objective to investigate the influence of the NO$_2$ profile information by replacing the 1° x 1° TM5 NO$_2$ profiles with the spatially higher resolved 0.1° x 0.1° CAMS-based profiles. The scatter plot comparing this TROPOMI data version with the AirMAP data is presented in Fig. 9b and shows a correlation coefficient of 0.86 and a slope of $0.41 \pm 0.02$. The median relative difference improves from -9 % to

-5 %. The correlation has not changed compared to the original data version and the slope increased only slightly demonstrating that the replacement of the $NO_2$ profile has only a small impact on this data set. In general, the replacement of the $NO_2$ profile increases the dynamical range of $NO_2$ VCDs with the largest impact (5 - 30 %) in emission hot spots but is dependent on the location and conditions (Douros et al., 2022). Tack et al. (2021) observed an increasing slope from 0.82 to 0.93 from the original data version to the version using the CAMS regional a priori over Belgium. Thus, the relative difference in slope between the

original V01.03.02 and the V01.03.02 CAMS data is similar with 13 % in Tack et al. (2021) and 8 % found in this study.

Since already several validation activities reported that the $NO_2$ data V01.02 - 01.03 are biased low, a modified TROPOMI $NO_2$ retrieval led to the development of V02.03.01 and a complete mission reprocessing (see Sect. 3.1.3). The comparison of this TROPOMI product PAL V02.03.01 with the AirMAP data in Fig. 9c shows much more scatter with a correlation coefficient which is significantly poorer than for the OFFL V01.03.02 product, changing from 0.86 to 0.76. The slope, however, increased

by more than a factor of 2 from $0.38 \pm 0.02$ to $0.83 \pm 0.06$, demonstrating that the updates in the new TROPOMI $NO_2$ data version have a large impact on the analyzed data set from the Rhine-Ruhr region. Due to the large scatter and driven by the large number of measurements with tropospheric $NO_2$ VCDs of less than about $7 \pm 0.15 \cdot 10^{15}\,\text{molec cm}^{-2}$, the PAL V02.03.01 product has a positive median relative difference of +20 % with an interquartile range of -14 % to +66 % (see Fig. A11). As described in Sect. 3.1.3, the main change from V01.03 to V02.03.01 is the switch to the FRESCO-wide product,

which provides more realistic higher cloud altitudes for measurements with cloud fractions larger than zero. Only 1 out of the 117 TROPOMI pixels used in this study has a cloud fraction of zero. Higher cloud altitudes result in decreased tropospheric AMFs and therefore higher tropospheric $NO_2$ VCDs. With the update many of the 117 data points show increased TROPOMI VCDs and are now closer or even over the 1:1 line and thus increasing the slope and the median relative difference. However, there is a lower branch of data points (with low TROPOMI $NO_2$, but large AirMAP $NO_2$ VCDs) which is not much affected

by the modifications in the new data version and is still matching the pattern of the OFFL V01.03.02 comparison (Fig. 9a). Comparisons of coincidences between the AirMAP and TROPOMI OFFL V01.03.02 and PAL V02.03.01 data, on a basis of single days show different magnitudes of the described impact from the TROPOMI data version change (see Appendix Fig. A8 and Fig. A9). The addressed lower branch visible in the overall comparison of TROPOMI PAL V02.03.01 and AirMAP (Fig. 9c) is dominated by observations from 17 September and is even after the change from FRESCO-S to FRESCO-wide linked

to cloud pressures close to the surface (see Fig. A9, points are color coded in the surface and cloud pressure difference). In the OFFL V01.03.02 product, 110 out of 117 pixels and thus 97 % of the TROPOMI observations were found to have cloud heights very close to the surface (within 50 hPa), which is not realistic and especially not for such a large amount of observations. In the new PAL V02.03.01 product, the cloud retrieval yields for 28 out of 117 pixels a cloud height close to the surface, resulting in a better slope of the regression line. However, since some scenes remain problematic, it results in more scatter. Previous

studies showed that for scenes with low clouds, i.e. close to the surface, a height that is even closer to the surface was retrieved by the original FRESCO implementation. Since the cloud algorithm does not discriminate between clouds and aerosols, this also holds for low aerosol layers. In many cases, FRESCO then retrieves the surface height, which is incorrect (Compernolle et al., 2021; van Geffen et al., 2022b). Observations during the flights and VIIRS images of the campaign measurement days revealed nearly perfect cloud free conditions during the measurements over the target areas. Thus, the high cloud pressures

are suspected to be caused by a higher aerosol load which is identified as cloud. This assumption can be supported by the pre-operational TROPOMI AOT product (de Graaf, 2022). The daily maps depicted in the Appendix Fig. A4 show a quite variable AOT over the region and between the different days, without any obvious correlation with the TROPOMI tropospheric $NO_2$ VCD. The highest AOT is found on 17 September spanning the pixels which are showing much lower tropospheric $NO_2$ VCDs than seen by AirMAP and are causing the lower branch in the scatter plot.

## 6.1 Cloud effects

For TROPOMI the tropospheric $NO_2$ VCDs are corrected for cloud and aerosol effects by the AMFs accounting for cloud-contaminated pixels using a combination of a cloudy tropospheric AMF and a clear-sky tropospheric AMF ($AMF_{trop, clr}$). The determined cloud radiance fraction from the $NO_2$ window is on average $0.21 \pm 0.10$ with a maximum of 0.48. As mentioned before, based on observations during the measurement flights, VIIRS images and the TROPOMI AOT product these clouds detected by the cloud retrieval must be mostly aerosols, which are identified as clouds in the cloud correction. For nearly cloud free observations, the cloud correction is more an aerosol correction (Boersma et al., 2011). To investigate the impact of the cloud correction on the TROPOMI tropospheric $NO_2$ VCDs, we calculated VCDs without this correction, $VCD_{trop, no\ cc}$, by:

$$VCD_{trop, no\ cc} = \frac{VCD_{trop} \cdot AMF_{trop}}{AMF_{trop, clr}} \tag{6}$$

Figure 10b shows the scatter plot between the TROPOMI PAL V02.03.01 tropospheric $NO_2$ VCD without cloud correction and the AirMAP tropospheric $NO_2$ VCD, having a high correlation of 0.85, a slope of $0.73 \pm 0.04$ and a median relative difference of +16 %. For comparison Fig. 10a shows again the original PAL V02.03.01 tropospheric $NO_2$ VCD data with the original cloud correction, having a correlation of 0.76, a slope of $0.83 \pm 0.06$ and a median relative difference of +20 %. The data version without cloud correction does not show the discussed lower branch anymore, and the upper branch is reduced. Hence, the product without cloud correction has a much better correlation and illustrates that the two branches are caused by the cloud correction.

To investigate the effect of TROPOMI observations with cloud pressures close to the surface, we use an additional coincidence criterion separating the TROPOMI data in observations in which clouds respectively aerosols are retrieved close to the surface and for cases in which this is not the case. As in Judd et al. (2020) the criterion is looking for differences between the cloud pressure and the surface pressure ($\Delta CS$), but different from Judd et al. (2020), data with $\Delta CS > 50\,hPa$ are kept and the observations for which low clouds are retrieved are filtered out or replaced. The limit of $50\,hPa$ is chosen, based on the reported uncertainty of the cloud pressure retrieval (van Geffen et al., 2022a). For the 117 coincident pixels from the six measurement days this criterion reduces the number of coincidences in the PAL V02.03.01 to 89. Thus, the cloud retrieval of PAL V02.03.01 yields a cloud height close to the surface for 23 % of the observations. In comparison, this is true for 53 % of the TROPOMI observations of V01.02 used in Judd et al. (2020) and for 97 % in V01.03.02 used in this study. Figure 10c shows the scatter plot of the TROPOMI PAL V02.03.01 versus the collocated AirMAP $NO_2$ VCDs limited to pixels with surface pressure - cloud pressure $\Delta CS > 50\,hPa$. Compared to the unfiltered PAL V02.03.01 product the slope and correlation increased from $0.83 \pm 0.06$ to $0.96 \pm 0.06$, and 0.76 to 0.84. The median relative difference increased from +20 % to +29 %. In a next step we

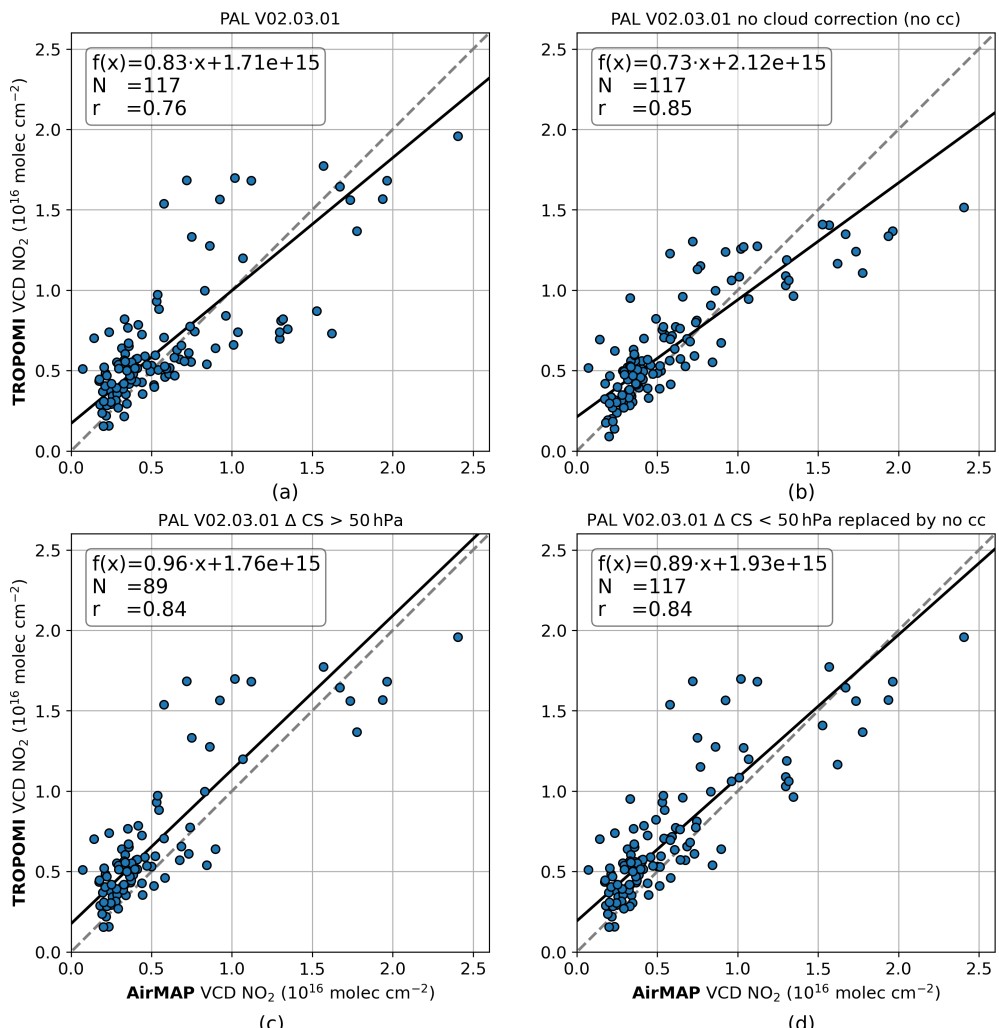

**Figure 10.** Scatter plots of TROPOMI $NO_2$ VCDs versus collocated AirMAP $NO_2$ VCDs for different versions of TROPOMI data: (a) PAL V02.03.01, (b) PAL V02.03.01 without cloud correction, (c) PAL V02.03.01 only pixels with surface pressure - cloud pressure $\Delta CS >$ 50 hPa, (d) PAL V02.03.01 pixels with $\Delta CS < 50$ hPa are replaced by $NO_2$ VCDs without cloud correction.

replaced the 28 observations with cloud pressures close to the surface with the VCDs without cloud correction. In this way, the number of coincidences is maintained. Figure 10d shows the result with a slope of $0.89 \pm 0.05$, a correlation of 0.84 and a median relative difference of +26 %.

The new TROPOMI data V02.03.01 provides a more realistic estimate of the cloud pressure for a large part of measurements as compared to earlier data versions. However, for certain cases with a higher aerosol load, which is treated as a cloud in the cloud retrieval, the cloud pressures remain close to the surface and lead to negative biased TROPOMI tropospheric $NO_2$ VCDs. Whether the cloud correction actually improves the $NO_2$ results in the presence of aerosols depends on the details of the

vertical distributions of aerosols and $NO_2$. In some cases, the results can be better if no cloud correction is made. To investigate this further, additional information about the vertical distributions of aerosols and $NO_2$ in the campaign area are needed.

## 6.2   $NO_2$ profile shape and surface reflectivity effects

To evaluate the influence of the auxiliary data, such as surface reflectivity or a priori $NO_2$ vertical profiles on the TROPOMI $NO_2$ data we developed a custom TROPOMI $NO_2$ product based on the retrieval of the PAL V02.03.01 product, named IUP
V02.03.01, with the possibility to change auxiliary data used within the retrieval.

    Figure 11a shows the comparison between the IUP V02.03.01 tropospheric $NO_2$ VCD and the AirMAP VCDs in dark blue. The PAL V02.03.01 data are shown in light blue (for details and regression statistics see Fig. 9c). The correlation is 0.76, as in the PAL data comparison. The slope of $0.88 \pm 0.06$ is slightly higher than the $0.83 \pm 0.06$, and within the uncertainties. Since the agreement between the PAL V02.03.01 and the IUP V02.03.01 version is fairly good, we assume that the effects of
changing auxiliary data would be similar for the PAL V02.03.01 product.

    To demonstrate the impact of higher resolved a priori $NO_2$ vertical profiles on the PAL V02.03.01 data, we recalculated AMFs and the tropospheric $NO_2$ VCDs using a priori tropospheric profiles from the regional $0.1° \times 0.1°$ CAMS-Europe analyses for altitudes between the surface and 3 km as described in Sect. 3.1.4. These IUP V02.03.01 REG tropospheric $NO_2$ VCDs are compared to the AirMAP data in Fig. 11b. Using the spatially higher resolved $NO_2$ profiles in the IUP V02.03.01 retrieval
increases the slope from $0.88 \pm 0.06$ (IUP V02.03.01) to $1.00 \pm 0.07$ (IUP V02.03.01 REG), while maintaining nearly the same correlation of 0.75 as compared to 0.76. With a relative difference in slope of 14 %, the change is showing a slightly larger impact than the 8 % we found for changing the a priori $NO_2$ profile information from TM5 to CAMS-Europe for the OFFL V01.03.02 data set. Using the spatially higher resolved profile information has the effect that the profile shape over source regions is improved in the sense that there is more $NO_2$ near the ground which decreases the AMF and thus increases the
tropospheric $NO_2$ VCD and is compensating the reduced sensitivity of TROPOMI for trace gases close to the surface. This has a larger effect in the case of the more realistic lower cloud pressures of the PAL V02.03.01. Observations for which the cloud pressure is still determined to be close to the surface, which are represented by the lower branch of points, are less affected by the change to the higher resolved profiles. In combination with the improved cloud treatment, however, the improved $NO_2$ profiles reveal their positive impact.

Recalculating AMFs with the regional CAMS $NO_2$ profiles and the TROPOMI LER result in the IUP V02.03.01 REG LER product. Figure 11c compares the IUP V02.03.01 REG TROPOMI LER and AirMAP tropospheric $NO_2$ VCD, showing a slope of $1.02 \pm 0.07$ and a correlation of 0.74. Compared to the IUP V02.03.01 REG data (Fig. 11b) the slope increased slightly from $1.00 \pm 0.07$ to $1.02 \pm 0.07$ and the correlation hardly changed from 0.75 to 0.74. The median relative difference changed from +31 % to +24 %. This comparison shows that replacing the OMI LER with the TROPOMI LER data only has a small impact
on the TROPOMI $NO_2$ VCD retrieval for our data set. Differences between the OMI LER and TROPOMI LER are rather small in the campaign region and in the $NO_2$ fit window but can be larger in other regions and a change would thus have a greater impact there.

    Recalculating AMFs with the regional CAMS $NO_2$ profiles and the TROPOMI DLER result in the IUP V02.03.01 REG DLER

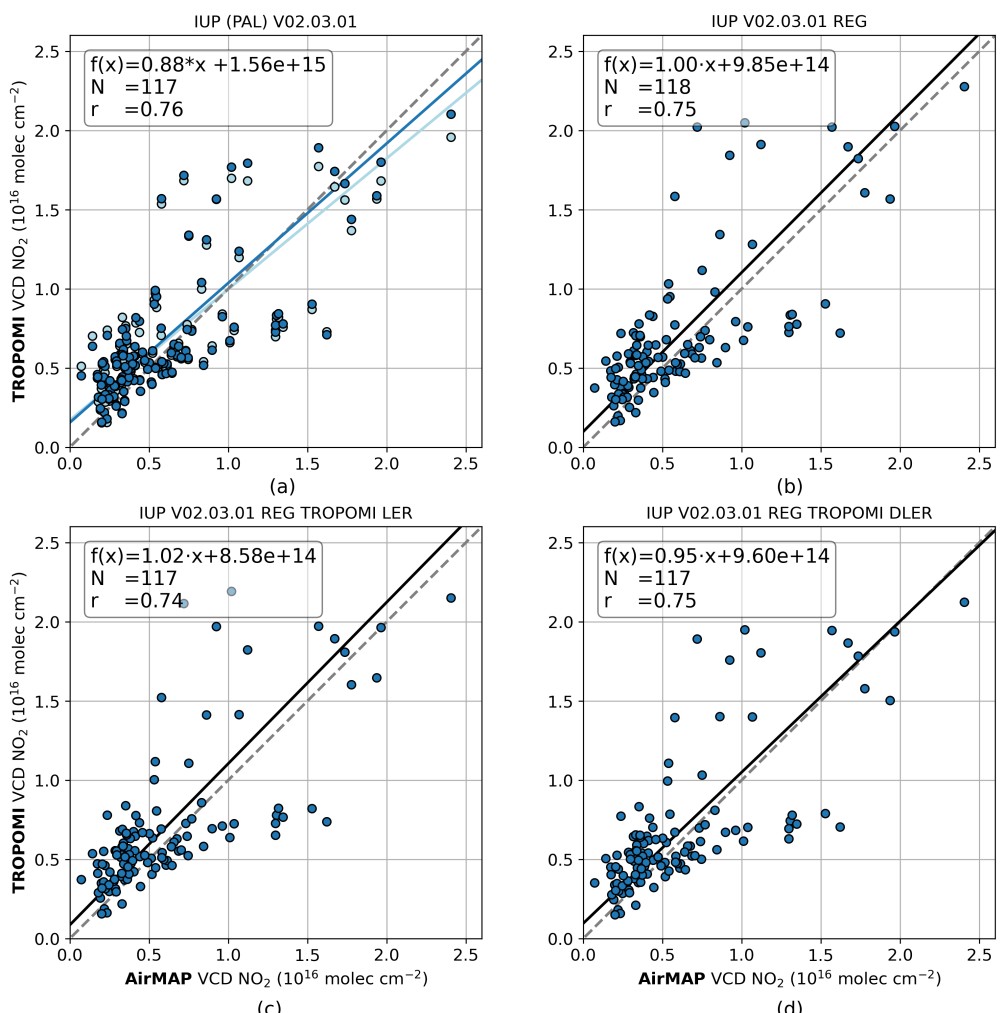

**Figure 11.** Scatter plots of TROPOMI NO$_2$ VCDs versus collocated AirMAP NO$_2$ VCDs for different versions of TROPOMI data: (a) the IUP V02.03.01 in dark blue and the PAL V02.03.01 in light blue, regression information are given for IUP V02.03.01, (b) the IUP V02.03.01 with regional CAMS profiles replacing the TM5 profile information, (c) the IUP V02.03.01 with regional CAMS profiles and TROPOMI LER replacing the OMI LER, (d) the IUP V02.03.01 with regional CAMS profiles and TROPOMI DLER.

product which is compared to the AirMAP data in Fig. 11d. The implementation of the DLER product leads to decreased TROPOMI NO$_2$ VCDs as compared to the products using OMI LER (b) or TROPOMI LER (c) and results in a slope of $0.95 \pm 0.07$ and a median relative difference of $+21\,\%$ with a correlation of 0.75. Thus, the directional aspect of the surface reflectivity only plays a small role in the tropospheric NO$_2$ retrieval in the campaign region with nearly cloud free conditions (mean cloud radiance fraction $= 0.21 \pm 0.10$) during the measurement days. As for the comparison between OMI LER and TROPOMI LER, it should be pointed out that this result is specific to the area, month and also cloud conditions, as the

reflectivity is influencing the cloud height retrieval and thus also the AMF. Larger differences could for example be expected for snow-covered surfaces with high reflectivity. Figure A10 in the Appendix shows scatter plots of the TROPOMI tropospheric $NO_2$ VCD retrieved with TROPOMI LER and TROPOMI DLER for the 117 TROPOMI pixels used throughout the study but also for larger areas up to one full orbit. All comparisons show only minor influences by the directional component. Since only TROPOMI observations made in September are compared, no larger snow-covered areas are expected and a more detailed

analysis including a different period and area would be needed to investigate possible larger differences.

All statistics of the comparisons between the different TROPOMI tropospheric $NO_2$ VCDs data versions and the AirMAP measurements are summarized in Table A1 and the box-and-whisker plots in Fig. A11 in the Appendix.

## 7 Conclusions

The presented comparisons have shown that the airborne imaging DOAS measurements performed by the AirMAP instrument

are specifically well suited for validating the TROPOMI tropospheric $NO_2$ VCDs. The airborne data set provides independently measured tropospheric $NO_2$ VCDs from seven mapping flights during the S5P-VAL-DE-Ruhr campaign in North-Rhine-Westphalia from 12 to 18 September 2020 covering in total 117 TROPOMI ground pixels on six of the days. These flights were accompanied by ground-based stationary and mobile car DOAS instruments. The important advantage of airborne imaging DOAS measurements is the mapping of the $NO_2$ variability within a satellite footprint, quantifying the expected dif-

ferences (representative errors) between satellite and surface measurements at a fixed location.

The ground-based stationary measurements conducted by different types of DOAS instruments (2 zenith-sky DOAS, 2 MAX-DOAS, 2 Pandora) deployed at different locations in the flight area provide independent, high precision and well-established data for the evaluation of the AirMAP retrievals. The AirMAP tropospheric $NO_2$ VCDs are highly correlated (r = 0.88) with the stationary ground-based VCDs with a slope of $0.90 \pm 0.09$. Due to limited overflight possibilities, the comparison is limited

to in total 25 coincident measurements.

The car DOAS measurements have the advantage that they are mobile, can cover larger and more diverse areas, and can be better synchronized to the AirMAP measurements. They have a high temporal resolution and are coordinated in the AirMAP flight area to gather many collocated measurements. For the evaluation of the AirMAP $NO_2$ VCD, 572 coincident measurements are considered which are highly correlated (r = 0.89) with a slope of $0.89 \pm 0.02$.

The combination of the two independent data sets to assess the AirMAP data gives confidence for using the AirMAP tropospheric $NO_2$ VCD data set to evaluate the TROPOMI products. Despite the fairly good spatial resolution of the TROPOMI measurements, the spatial variability within TROPOMI pixels can be large and cannot be fully captured by ground-based instruments. The AirMAP data, having a resolution of about $100\,m \times 30\,m$, create a link between the ground-based and the TROPOMI measurements with a nadir resolution of $3.5\,km \times 5.5\,km$. Airborne measurements are more representative of the

satellite measurements than point measurements as a large number of TROPOMI pixels can be fully mapped in a relatively short time.

For the comparison of TROPOMI and AirMAP tropospheric $NO_2$ VCDs, only TROPOMI pixels that are at least 75 % mapped

by AirMAP are used and measurements that are less than $\pm$ 30 min separated in time. This results in 117 TROPOMI pixels coinciding with AirMAP measurements during the six flights. Due to nearly cloud free conditions during the measurement days, the cloud radiance fraction retrieved in the TROPOMI $NO_2$ spectral window was on average $0.21 \pm 0.10$ with a maximum of 0.48 and thus for all measurements below the recommended filter criterion of 0.5.

We evaluate the TROPOMI tropospheric $NO_2$ VCD data from 12 September to 18 September 2020, using the two data products OFFL V01.03.02 and PAL V02.03.01 as well as scientific data versions. One scientific version is based on the OFFL V01.03.02 with a replacement of the a priori $NO_2$ profiles from the TM5 model by the CAMS-Europe and CAMS-global product, and one scientific product reproduces the PAL V02.03.01 in which different a priori assumptions are replaced and their effects investigated.

The different TROPOMI and AirMAP data sets are correlated with correlation coefficients between 0.74 and 0.86, slopes of $0.38 \pm 0.02$ to $1.02 \pm 0.07$ and relative mean differences between -9 % and 31 %. The operational OFFL V01.03.02 and the scientific V01.03.02 CAMS product show a clear underestimation of TROPOMI compared to the AirMAP tropospheric $NO_2$ VCDs with a slope of $0.38 \pm 0.02$ respectively $0.41 \pm 0.02$ and median relative differences of -9 % and -5 %. Both products show a high correlation with a correlation coefficient of 0.86.

The updates implemented in the TROPOMI PAL V02.03.01 product increase the slope from $0.38 \pm 0.02$ to $0.83 \pm 0.06$ but result in much more scatter and reduce the correlation from 0.86 to only 0.76, demonstrating the large impact of the modifications on the analyzed data set. Due to the large scatter and driven by the large number of measurements with tropospheric $NO_2$ VCDs of less than about $7 \pm 0.15 \cdot 10^{15}\,\mathrm{molec\,cm^{-2}}$, the PAL V02.03.01 product has a median relative difference of +20 % with an interquartile range of -14 % to +66 %. The main change influencing the tropospheric $NO_2$ VCD is the switch from the FRESCO-S to the FRESCO-wide product which results in more realistic higher cloud altitudes, therefore decreased tropospheric AMFs and higher tropospheric $NO_2$ VCDs. In the analyzed TROPOMI data set many of the data points are effected by the modifications and thus closer or even over the 1:1 line and are increasing the slope and the median relative difference. However, there is a lower branch with low TROPOMI $NO_2$ VCDs, but large AirMAP $NO_2$ VCDs that still shows cloud pressures close to the surface. The clearly decreased correlation is mainly caused by this separation of the data into two branches; one branch around the 1:1 line and a second branch with low biased TROPOMI observations close to the distribution seen in OFFL V01.03.02.

We found that the TROPOMI observations on the lower branch are dominated by the observations from one day and are linked to cloud pressures which are still close to the surface as in the OFFL V01.03.02 product, i.e. they are not much affected by the modifications. Due to nearly cloud free conditions during the measurement flights, the high cloud pressures are suspected to be caused by a higher aerosol load which is identified as cloud and are not accounted for adequately in the cloud correction. This assumption is supported by the TROPOMI AOT product which is showing a high AOT for the pixels which are showing much lower tropospheric $NO_2$ VCDs than seen by AirMAP and are causing the lower branch in the scatter plot.

Comparing TROPOMI PAL V02.03.01 VCDs without cloud correction with the AirMAP VCDs decreases the slope from $0.83 \pm 0.06$ to $0.73 \pm 0.04$ and the median relative difference from +20 % to +16 % but brings the two branches together which improves the correlation from 0.76 to 0.84. This illustrates that the two branches are caused by the cloud correction. We intro-

duced an additional criterion that filters TROPOMI observations with surface to cloud pressure differences of less than 50 hPa, i.e. clouds close to the surface, and either excluded these pixels or replaced them with the $NO_2$ VCDs without cloud correction.

This increases the slope from $0.83 \pm 0.06$ to $0.96 \pm 0.06$, respectively $0.89 \pm 0.05$, and improves the correlation from 0.76 to 0.84. Thus, we saw that the PAL V02.03.01 $NO_2$ product provides a more realistic estimate of the cloud pressure for a large part of measurements as compared to earlier data versions but for certain cases with a higher aerosol load cloud pressures remain close to the surface and lead to negative biased TROPOMI tropospheric $NO_2$ VCDs and a larger scatter. Therefore, in some cases, the results can be better if no cloud correction is made.

We developed a custom TROPOMI $NO_2$ product based on the retrieval of the PAL V02.03.01 but replacing the TM5 a priori $NO_2$ profiles with the spatially higher resolved CAMS-Europe product for altitudes up to 3 km. This modification increases the slope from $0.88 \pm 0.06$ to $1.00 \pm 0.07$ with consistent correlation.

Replacing, in addition, the OMI LER data with the higher resolved TROPOMI LER or DLER data in the $NO_2$ fit window, respectively, only has a small impact on the TROPOMI $NO_2$ VCDs of our data set and the comparison to the AirMAP data. The

slope increases from $1.00 \pm 0.07$ to $1.02 \pm 0.07$ using the TROPOMI LER and decreases to $0.95 \pm 0.07$ using the TROPOMI DLER. The influence of the surface reflectivity on the VCD retrieval is rather small in the campaign region but can be larger in other regions, seasons, especially with snow-covered surfaces and under different cloud conditions, as the reflectivity is influencing the cloud height retrieval and thus also the AMF. A larger impact is expected when applying the TROPOMI DLER in the NIR-FRESCO cloud retrieval, effecting the $NO_2$ retrieval through adjusted cloud parameters.

In summary, a validation of the TROPOMI tropospheric $NO_2$ retrievals based on airborne mapping flights, supported by ground-based stationary and car DOAS measurements, has been presented. We found that the modifications in the cloud pressure retrieval in the TROPOMI PAL V02.03.01 data product leads to more realistic lower cloud pressures and thus larger tropospheric $NO_2$ VCDs for a large part of the analyzed observations compared to the OFFL V01.03.02 product. While this improves the slope, it significantly increases the scatter. The results can be improved, if for cases with high aerosol load and

retrieved cloud pressures close to the surface no cloud correction is made. Spatially higher resolved a priori $NO_2$ profile information can further increase the tropospheric $NO_2$ VCDs, while the application of the TROPOMI LER and DLER had only small effects. Further validation activities on the TROPOMI PAL V02.03.01 data product using larger data sets in more regions with different pollution levels, surface reflectance, aerosol and cloud conditions would help to evaluate the performance of the TROPOMI $NO_2$ product under different conditions and confirm the results found in this data set. After reprocessing of the

new V02.04 $NO_2$ retrieval, which has a consistent implementation of the TROPOMI DLER climatology in the $NO_2$ fit window and the NIR band for the cloud retrieval, comparisons to the campaign data set can investigate the impact of this modification. The presented validation strategy can be assigned to future validation activities for upcoming satellite missions such as GEMS, TEMPO, Sentinel-4, and Sentinel-5.

*Data availability.* TROPOMI data from July 2018 onwards are freely available via https://s5phub.copernicus.eu/ (S5P Data Hub, 2022).

The reprocessed PAL V02.03.01 data product is freely available via https://data-portal.s5p-pal.com (S5P PAL Data Portal, 2022). The TROPOMI pre-operational AOT product is freely available via https://data-portal.s5p-pal.com (S5P PAL Data Portal, 2022). The data of both Pandora instruments are freely available from the PGN data archive (https://pandonia-global-network.org/, last access: 21 March 2022). The TROPOMI DLER database is freely available via https://www.temis.nl/surface/albedo/tropomi_ler.php. The ERA5 reanalysis data are freely available from the Copernicus Climate Change (C3S) climate data store (CDS).

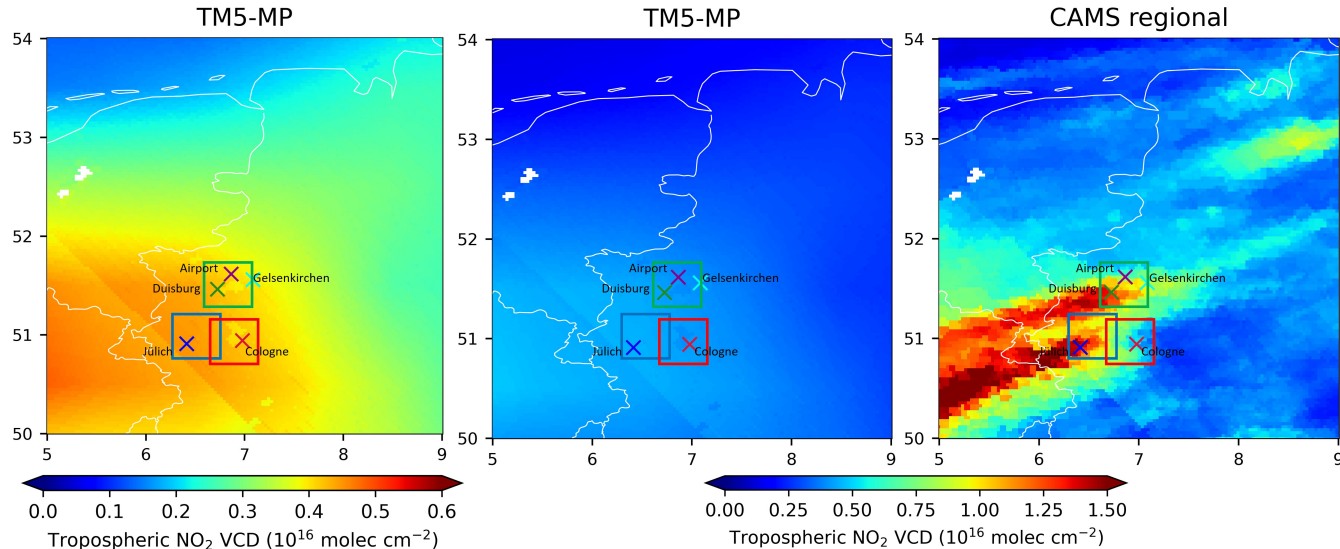

**Figure A1.** Tropospheric $NO_2$ VCD of the TM5-MP (1° x 1°) and the CAMS regional (0.1° x 0.1°) analysis for the campaign region on 17 September 2020, interpolated to TROPOMI pixels and oversampled to a 0.03° x 0.03° resolution.

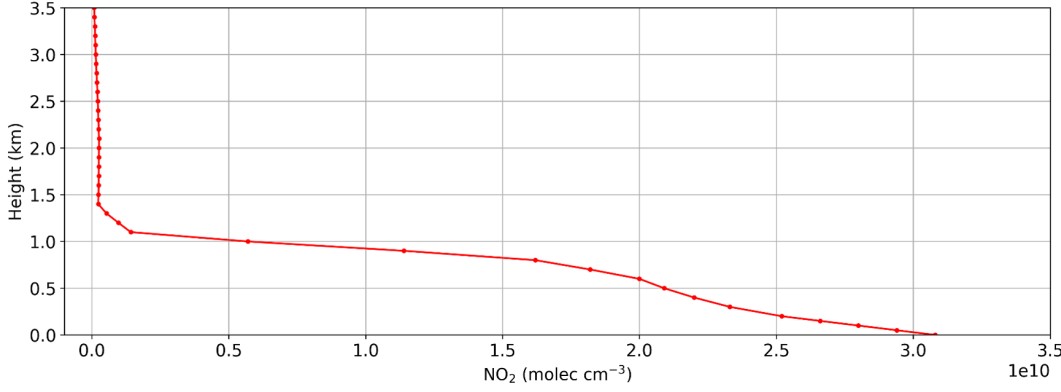

**Figure A2.** $NO_2$ profile used in the SCIATRAN tropospheric AMF calculations. The profile is based on old WRF-Chem model runs and scaled to the typical boundary layer height during the measurement days around noon.

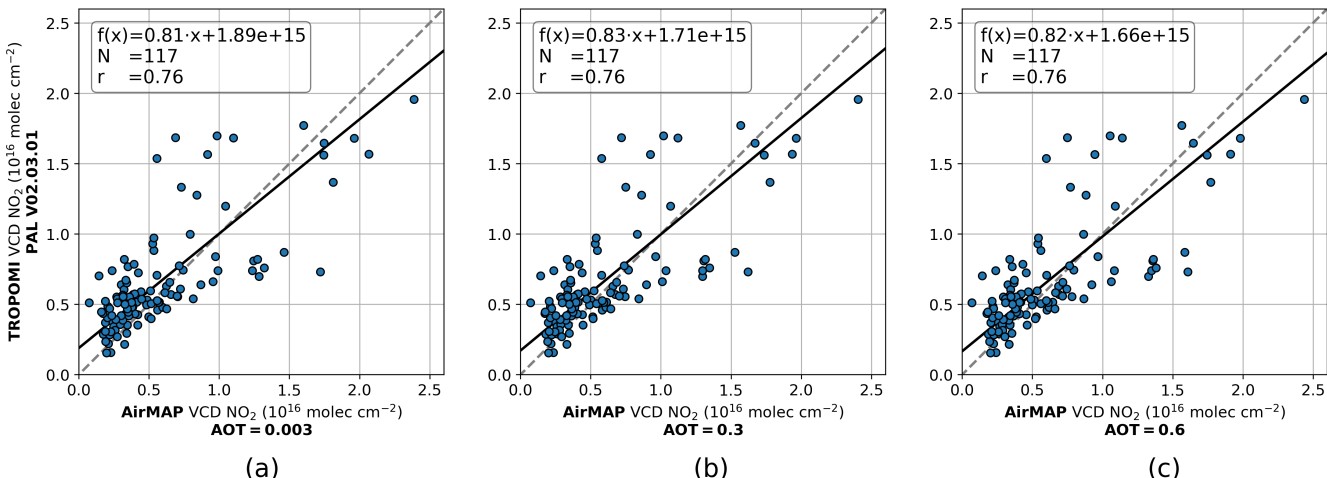

**Figure A3.** Scatter plots of TROPOMI PAL V02.03.01 tropospheric $NO_2$ VCDs versus collocated AirMAP tropospheric $NO_2$ VCDs with (a) AOT of 0.003, (b) AOT of 0.3 and (c) AOT of 0.6. Collocation criteria for AirMAP: $\pm 30$ min around S5P overpass, gridded to the TROPOMI pixels and covering them at least to 75 %.

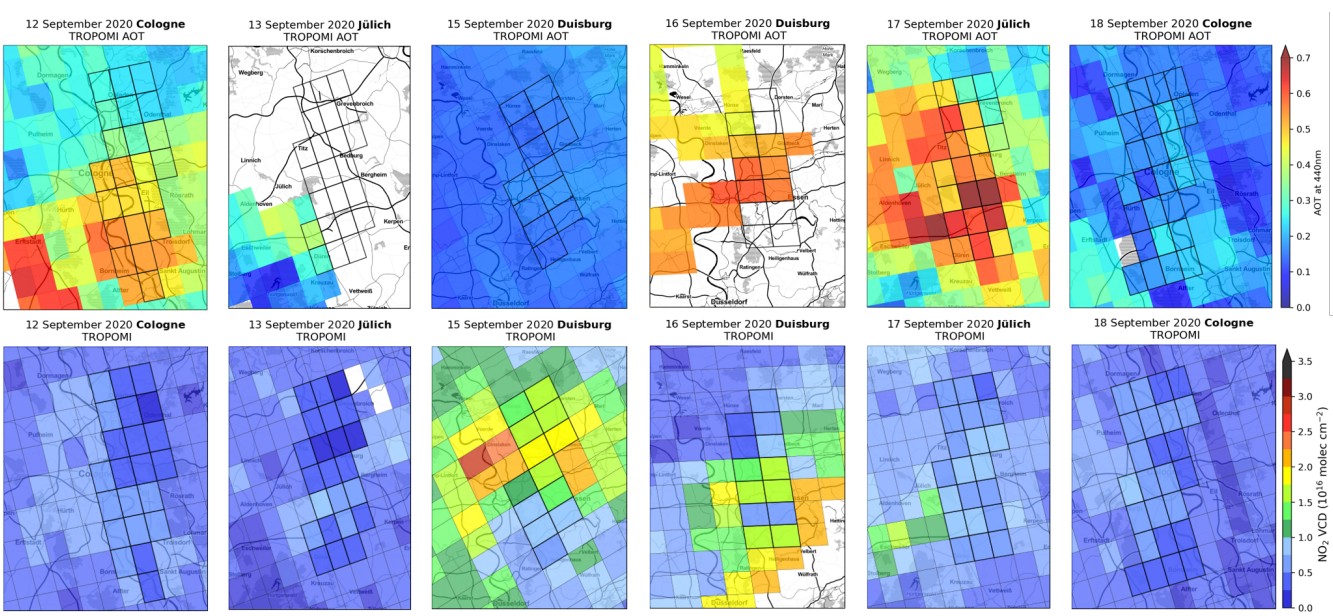

**Figure A4.** Daily maps of (top) TROPOMI AOT at 440 nm where qa_value > 0.5 and (bottom) TROPOMI PAL V02.03.01 tropospheric $NO_2$ VCDs where qa_value > 0.75. Black boxes are representing TROPOMI pixel outlines which are fulfilling the collocation criteria of a AirMAP coverage of at least 75 % and AirMAP measurements performed $\pm 30$ min around the S5P overpass (see also Fig. 8).

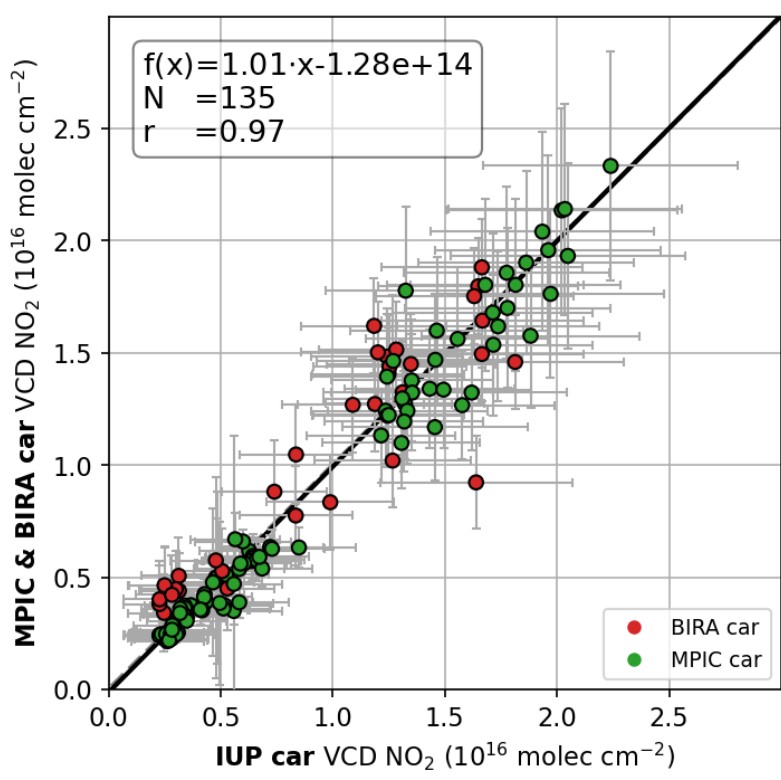

**Figure A5.** Scatter plot between collocated car DOAS measurements ($\pm 5$ min time window) of MPIC and BIRA car DOAS data versus IUP car DOAS tropospheric $NO_2$ VCDs averaged within 200 m x 200 m grid boxes and 5 min time intervals. The data points from the BIRA and MPIC car DOAS instrument are color coded in red and green. The thick solid black line represents the orthogonal distance regression. Error bars represent the error in the tropospheric $NO_2$ VCD retrieval, averaged within the 200 m x 200 m grid boxes and 5 min time intervals.

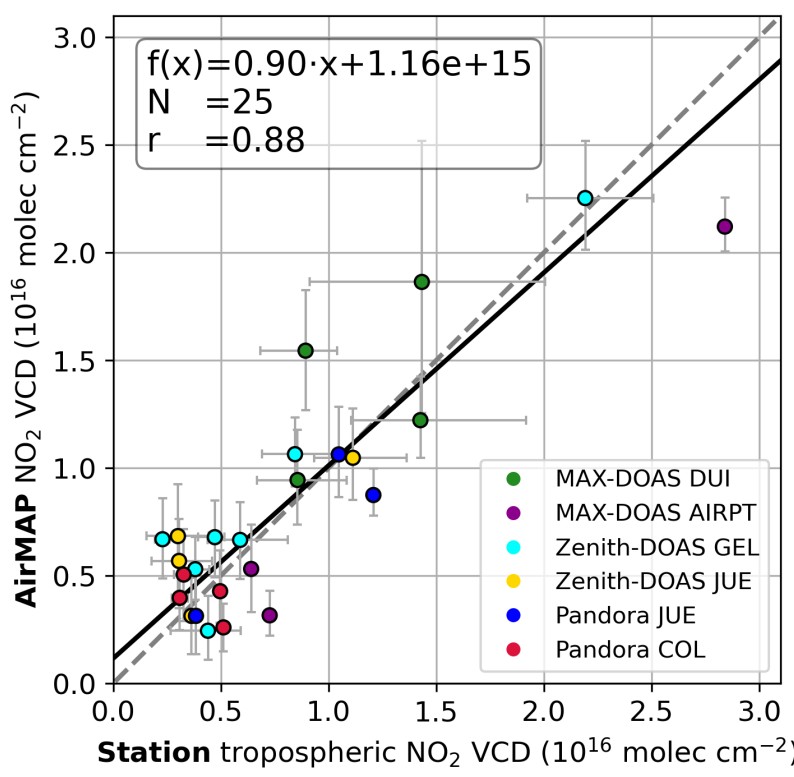

**Figure A6.** Same as Fig. 6 with different error bars. Scatter plot showing the stationary ground-based NO₂ VCDs averaged in a time interval of 20 min closest to the AirMAP overpass data which are averaged over a 500 m x 500 m box around the station site. Error bars represent the 10th and 90th percentile within the 500 m x 500 m grid boxes and 20 min time intervals.

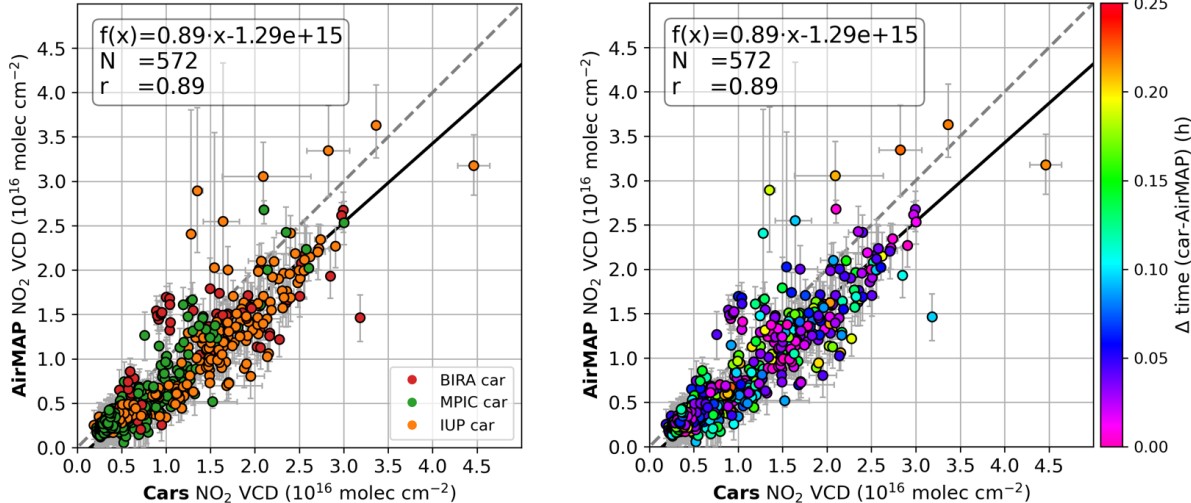

**Figure A7.** Similar to Fig. 7 with different error bars. Scatter plots showing collocated car DOAS ($\pm$ 15 min window from the aircraft overpass) and AirMAP NO$_2$ VCDs using grid boxes of 500 m x 500 m and 15 min time intervals. The data points from BIRA, MPIC and IUP car DOAS instruments are color coded red, green and orange (left). The color coding in the right plot shows the time difference between the AirMAP and car DOAS measurements. Error bars represent the 10th and 90th percentile within the 500 m x 500 m grid boxes and 15 min time intervals.

**Table A1.** Statistics of the comparisons between the different TROPOMI tropospheric NO$_2$ VCDs data versions and AirMAP measurements. Slope and offset $\pm$ standard deviation (SD) of the orthogonal distance regression, median relative difference and Pearson correlation coefficient.

| TROPOMI NO$_2$ data version | Slope $\pm$ SD | Median difference (%) | Offset $\pm$ SD ($\cdot 10^{15}$ molec cm$^{-2}$) | Correlation coefficient |
|---|---|---|---|---|
| OFFL V01.03.02 | 0.38 $\pm$ 0.02 | -9 | 2.54 $\pm$ 0.15 | 0.86 |
| OFFL V01.03.02 CAMS | 0.41 $\pm$ 0.02 | -5 | 2.63 $\pm$ 0.16 | 0.86 |
| PAL V02.03.01 | 0.83 $\pm$ 0.06 | 20 | 1.71 $\pm$ 0.42 | 0.76 |
| PAL V02.03.01, AirMAP AOT=0.003 | 0.81 $\pm$ 0.06 | 24 | 1.89 $\pm$ 0.41 | 0.76 |
| PAL V02.03.01, AirMAP AOT=0.6 | 0.82 $\pm$ 0.06 | 17 | 1.66 $\pm$ 0.43 | 0.76 |
| PAL V02.03.01 no cloud correction (no cc) | 0.73 $\pm$ 0.04 | 16 | 2.12 $\pm$ 0.29 | 0.85 |
| PAL V02.03.01 $\Delta$CS > 50 hPa | 0.96 $\pm$ 0.06 | 29 | 1.76 $\pm$ 0.41 | 0.84 |
| PAL V02.03.01 $\Delta$CS > 50 hPa replaced with no cc | 0.89 $\pm$ 0.05 | 26 | 1.93 $\pm$ 0.37 | 0.84 |
| IUP V02.03.01 | 0.88 $\pm$ 0.06 | 26 | 1.56 $\pm$ 0.45 | 0.76 |
| IUP V02.03.01 REG | 1.00 $\pm$ 0.07 | 31 | 0.99 $\pm$ 0.51 | 0.75 |
| IUP V02.03.01 REG TROPOMI LER | 1.02 $\pm$ 0.07 | 24 | 0.86 $\pm$ 0.54 | 0.74 |
| IUP V02.03.01 REG TROPOMI DLER | 0.95 $\pm$ 0.07 | 21 | 0.96 $\pm$ 0.50 | 0.75 |

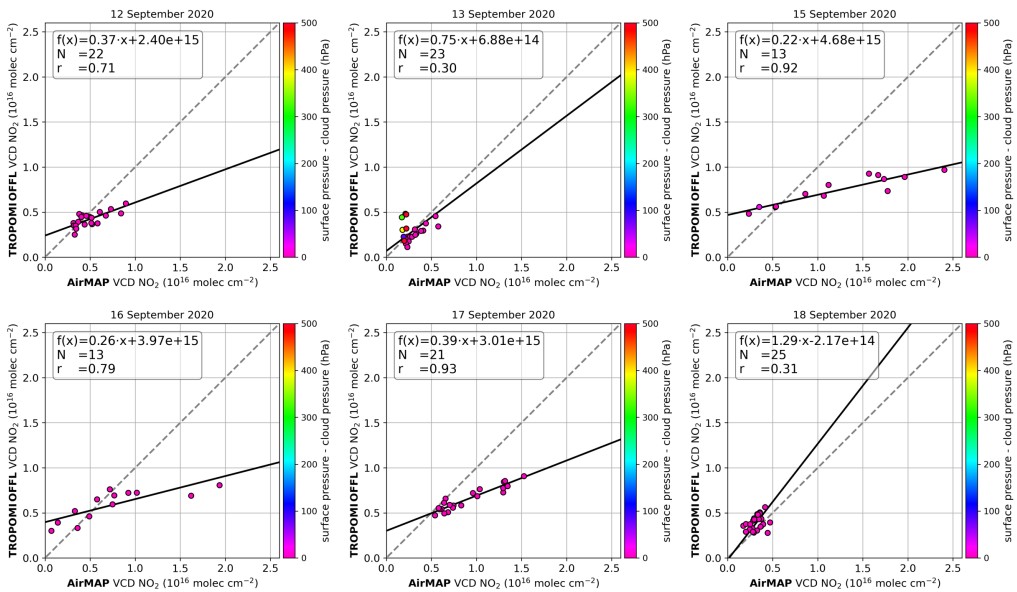

**Figure A8.** Daily scatter plots of TROPOMI operational OFFL V01.03.02 tropospheric $NO_2$ VCDs versus collocated AirMAP tropospheric $NO_2$ VCDs for the six measurement days. Points are color coded in the surface and cloud pressure difference. Collocation criteria for AirMAP: $\pm 30$ min around S5P overpass, gridded to the TROPOMI pixels and covering them at least to 75 %.

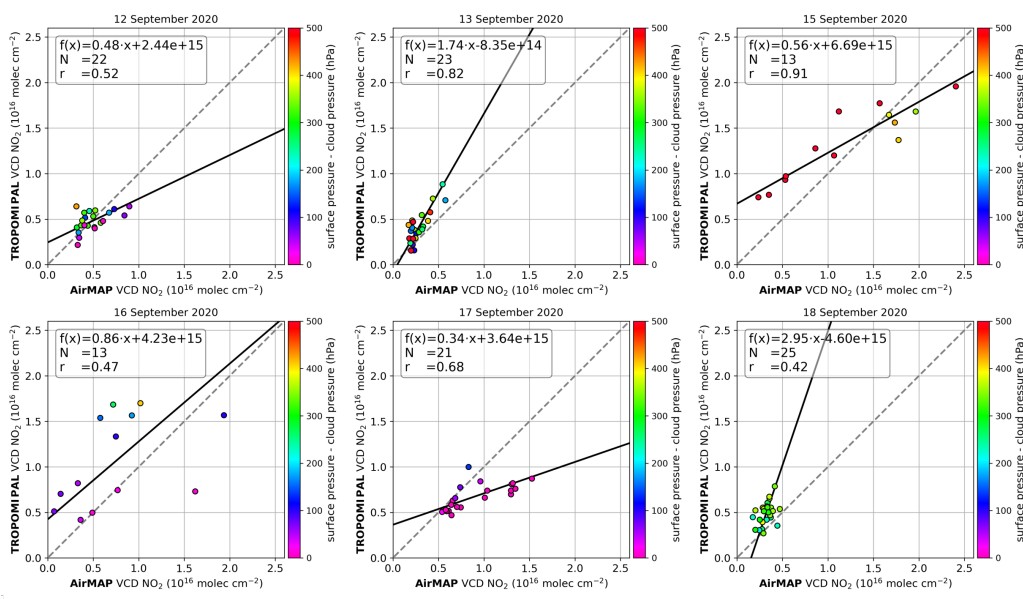

**Figure A9.** Same as Fig. A8 but for TROPOMI PAL V02.03.01 tropospheric $NO_2$ VCDs versus collocated AirMAP tropospheric $NO_2$ VCDs for the six measurement days.

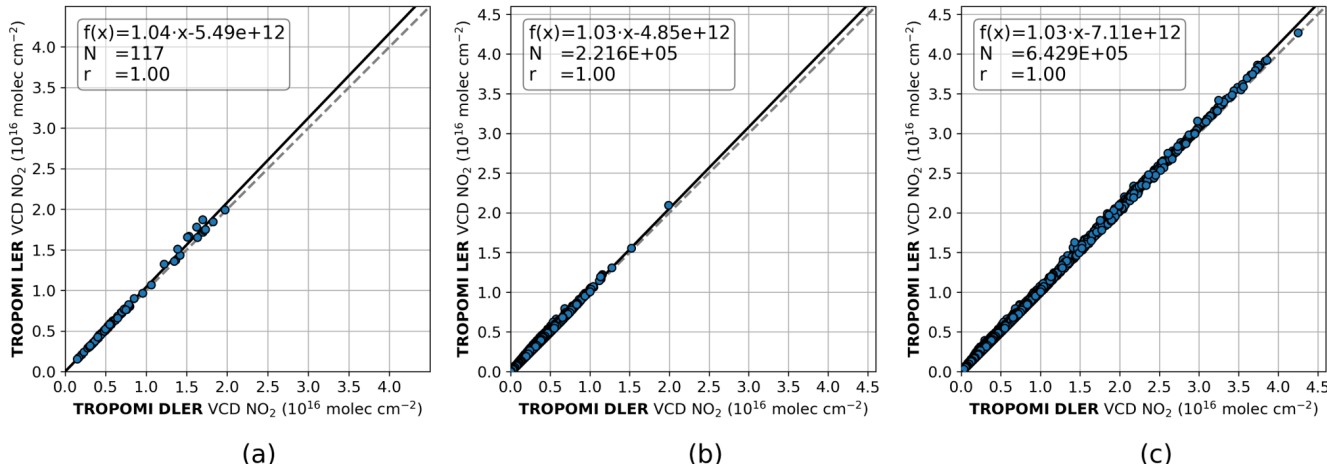

**Figure A10.** Scatter plots of TROPOMI IUP V02.03.01 tropospheric NO$_2$ VCDs with TROPOMI LER respectively TROPOMI DLER for: (a) the 117 TROPOMI pixels coinciding with the AirMAP measurements used throughout the study, (b) a larger orbit segment over Western Europe on 13 September 2020 and (c) one full orbit including the campaign area on 13 September 2020. All data are quality and cloud filtered using the qa_value of 0.75.

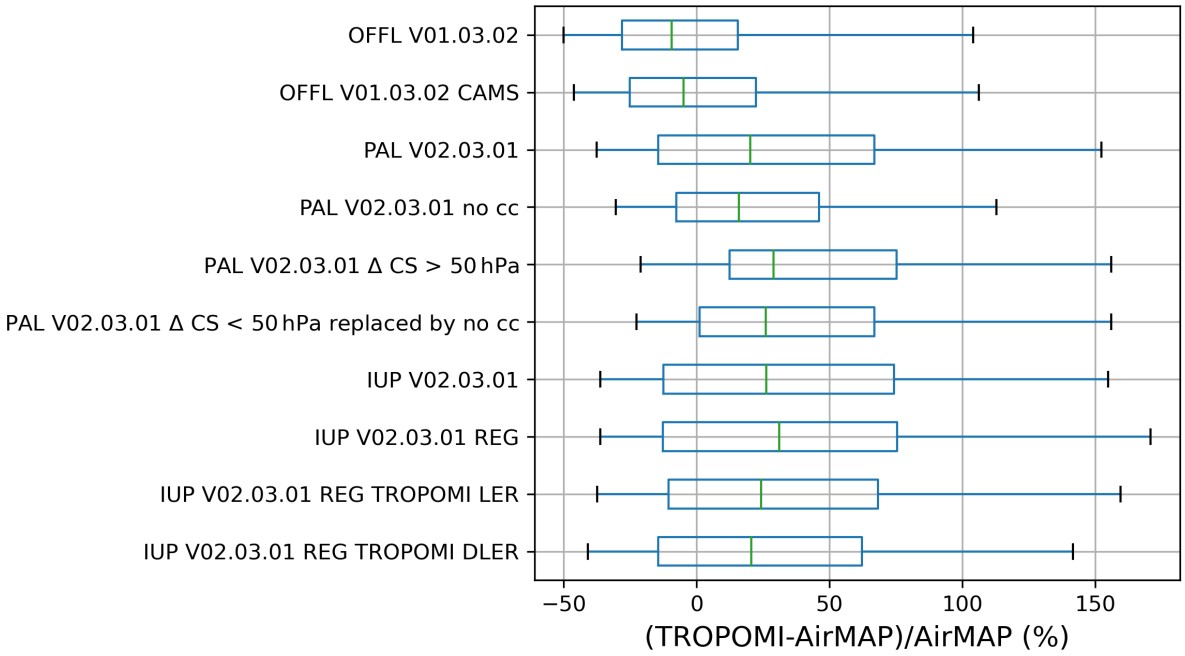

**Figure A11.** Box-and-whisker plots summarizing the bias and spread of the difference between the different TROPOMI versions and AirMAP tropospheric NO$_2$ VCDs. The green line inside the box represents the median relative difference. Box bounds mark the 25 and 75 percentiles while whiskers represent the 5 and 95 percentiles.

*Author contributions.* All co-authors contributed to the campaign either as participants and instrument operators and/or during campaign preparation, follow-up data analysis and providing their data of the individual instruments. KL and ACM performed the final data analysis. KL, AR, AS, ACM, TB, ASE and JPB interpreted the results of the study and wrote the paper with feedback and contributions from all other co-authors.

*Competing interests.* The authors declare that they have no conflict of interest. Andreas Richter and Thomas Wagner are executive editors at AMT.

*Acknowledgements.* The European Space Agency (ESA; contract 4000128426/19/NL/FF/ab; QA4EO Atmospheric Composition Uncertainty Field Studies Project) is gratefully acknowledged for funding the Ruhr campaign. The Deutsches Zentrum für Luft- und Raumfahrt (grant no. 50 EE 1709A) is gratefully acknowledged for financial support. Copernicus Sentinel-5P level-2 $NO_2$ data are used in this study.

Sentinel-5 Precursor is a European Space Agency (ESA) mission on behalf of the European Commission (EC). The TROPOMI payload is a joint development by ESA and the Netherlands Space Office (NSO). The Sentinel-5 Precursor ground-segment development has been funded by the ESA and with national contributions from the Netherlands, Germany, Belgium, and UK. We acknowledge the free use of the TROPOMI surface DLER database provided through the Sentinel-5p+ Innovation project of the European Space Agency (ESA). The TROPOMI surface DLER database was created by the Royal Netherlands Meteorological Institute (KNMI). Authors acknowledge

AERONET-Europe for providing calibration service. AERONET-Europe is part of ACTRIS-IMP project that received funding from the European Union (H2020-INFRADEV-2018-2020) under Grant Agreement No 871115. We would like to acknowledge the Umwelt- und Verbraucherschutzamt Stadt Köln for providing location and support for the Pandora Cologne measurement site and Ulrich Quass for providing location and support for the zenith-sky DOAS instrument. We thank the pilot of the aircraft, Jeremy Gordon, for his calm and professional flights as well as his guidance in all matters related to the aircraft and the weather conditions.

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
