# Peer review of "Validation of Sentinel-5P TROPOMI tropospheric $NO_2$ products by comparison with $NO_2$ measurements from airborne imaging, ground-based stationary, and mobile car DOAS measurements during the S5P-VAL-DE-Ruhr campaign"

_Atmospheric Measurement Techniques, 2022_

## Referee Comment (RC1)

**General suggestions**

Sect.3: Why are the fitting windows so variable from instrument to instrument (DOAS instruments)? Please comment on this in the instrument description. Looking at the comparison it doesn't seem to make much of a difference, but I would suggest commenting on this.

Sect. 3, make sure to include uncertainty estimates of all instruments and include references to existing validation papers. The sections 3.3.4, and 3.4.1 to 3.4.4 lack references.

l. 550-555: I would suggest to check the TROPOMI AOT in the area for that day, available from https://data-portal.s5p-pal.com/; e.g. you could include maps of this in the appendix or at the very least state the average (and std) for the flights. Alternatively, VIIRS AOD or MODIS MAIAC AOD are very good AOD products (but this might be a little too much effort to include here, maybe just something to keep in mind for future studies).

l. 619: just a thought: there was no snow during the campaign; there could be a larger difference between the DLER and LER TROPOMI product for snow covered surfaces with high reflectivity. It would be nice to include a little comparison of the DLER and LER product for snow covered surfaces. Nothing extensive, just a sentence (near l. 619) and a scatter plot (TROPOMI DLER vs TROPOMI LER) in the appendix (if time permits).

**Technical/minor suggestions**

l. 1: suggesting to change to "Airborne, ground-based stationary and car imaging differential optical absorption spectroscopy (DOAS) measurements…"

l. 4: emitters -> sources

l. 5: "The DOAS measurements…"

l. 7… suggest using "observations" instead of measurements throughout the text (technically measurements are in situ measurements), and remote-sensing are observations

l.13: data create

l. 13-31: This paragraph can be shortened, I think the most important points are: 1) The PAL version improves the bias significantly, 2) cloud height and NO2 profile have a major impact, 3) surface reflectivity has a minor impact (in this region and time of year – this is likely different when there is snow on the ground). These points get a little lost in the lengthy paragraph, maybe include the correlations and biases inside the sentence in brackets rather than writing whole sentences about it (a little repetitive).

l.36: change to "…combustion processes, such as power plants and engines, as well as anthropogenic biomass burning." (How much does anthropogenic biomass burning impact the Ruhr area? Consider removing the last half of the sentence.)

l. 37: "NOx is primarily emitted as NO, the reaction…"

l.37: "The NOx sources are …" (remove the characteristics)

l. 38: chemically active -> reactive and short lived

l. 38/39: "…, there is a high spatial and temporal variability of NO2 near emission sources." (there is not much variation in background areas)

l. 39: remove "on"

l.44: "is remotely observed from different platforms" -> " can be observed remotely on a variety of platforms"

l.46f: is identified -> can be identified; are -> can be

l. 53: remove "on board the European…satellite."

l. 55 TEMPO is planned for launch in March 2023

l. 56-58: *consider re-wording this sentence, maybe change it to 2 shorter sentences*

l. 86: include a sentence about the new TROPOMI version and what changed/improved in comparison to the previous version. Maybe include studies that validated this new version, I know of Zhao et al. (2022) see reference list, there might be others too. E.g. Riess et al (2022) also talks about the improvements of the new TROPOMI version.

l. 89: industrial estates -> industrial facilities, arterial highways -> busy highways (or large highways)

l. 89: "Back-ground areas with low pollution, as well as moderately polluted regions are also observed…"

l. 96: remove "In the following,"

l. 107: 5 million inhabitants -> has a population of 5 million

l. 108f: "The region, including nearby metropolitan centres along the Rhine and populated surroundings is called Metro… and is comprised of a population of over 10 m, large power plants [*can you include a number here*], … industrial facilities and several large highways."

l. 110: above the campaign location -> in the MRR

l.120:"… dominated by the emissions of three lignite fire power plants in the area (see European Pollutant Release and Transfer Register (E-PRTR)[*include reference or url here*]).

l. 121: "…around Cologne and Duisburg…" , remove latter part of the sentence "and the flight area around Duisburg has a similar character to that of the Cologne area with a mixture of urban and industrial emitters but includes the central metropolitan Ruhr area, which has a large variety of pollution sources. "

l. 137: here and any other occurrences: don't shorten Table to Tab. as per AMT guidelines, "are given in Table 1"

l. 143: "…comparison of the aircraft and TROPOMI NO2…"

l. 143: remove  "prior to the dedicated evaluation … satellite pixel area", I'm not sure what you mean here, but I think it's not necessary

l. 144f: remove: "In this manner", " on the one hand", "on the other hand"

l. 145: local -> ground-based , with restricted -> that have restricted, with satellite -> to satellite

l. 151: Table 2

l. 157: of aerosol -> aerosol

l. 160: thus by far -> currently

l. 167: AMFs. The AMFs are generated using the OMI…

l. 167: , and cloud fraction

l. 171: (e.g. Verholst et al, 2021).  *(there are many others)*

l. 173f: very long sentence, consider numbering the reasons and removing unnecessary details: " other factors that could contribute are: (1)…"

l. 186: recipe provided-> approach as detailed

l. 231: Tab.->Table

l. 255: remove "used"

l. 261: *438-490nm; is this the same or different to TROPOMI?, and other DOAS fits used in this study*

l. 265ff: *it's unusual to trop as a superscript, subscripts would be more common, e.g. subscript of* $_{trop,ref}$ *could be used*

l. 271: during -> near

l.272:change to: " There is a maximum difference of 3h between the time of the reference background and the actual measurements."

l. 276: *how small, do you have a reference that it is negligible?*

l. 279: (see ERA5 reanalysis; Hersbach et al. (2018). *You can include the data  source in the data availability section, and data source reference*

l. 298: comprises of

l. 300: remove "further details can be found therein."

l. 352: *why is the spectral window different to the AirMAP window?*

l. 371/379: *again why is the spectral window different?*

l. 397: "different target areas" -> "three target areas"

l. 411: do you have a reference for this? This paragraph in general could benefit from a couple of reference.

l. 413: The same AMF (1.3) is used to convert to a VCD, wouldn't it depend on the SZA? How much of an impact does the SZA have?

l. 423: insert reference here

l. 440-450: what is the uncertainty of the PANDORA observatons. Also please include references in this paragraph.

l. 465: is it really +- 10-90 percentile, I think it is just the 10th and 90th percentile.

l. 470: data is -> data are

l. 473: As a result of having more opportunities to make near simultaneous synchronized measurements, -> Consequently,

l. 487:  is it really +- 10-90 percentile, I think it is just  the 10th and 90th percentile.

Fig. 7: I would suggest moving one of these (I suggest Fig. 7b) to the appendix, as they essentially show the same.

l. 637: areas -> areas,

l.642: scatter -> scatter,

l.642: measurements -> measurements,

l. 687: dataset -> data set (either or but be consistent, I found both dataset and data set used in the study, please fix this)

l. 704: Sentinel-4 -> Sentinel-4,

References

Zhao, X.; Fioletov, V.; Alwarda, R.; Su, Y.; Griffin, D.; Weaver, D.; Strong, K.; Cede, A.; Hanisco, T.; Tiefengraber, M.; McLinden, C.; Eskes, H.; Davies, J.; Ogyu, A.; Sit, R.; Abboud, I.; Lee, S.C. Tropospheric and Surface Nitrogen Dioxide Changes in the Greater Toronto Area during the First Two Years of the COVID-19 Pandemic. *Remote Sens.* **2022**, *14*, 1625. https://doi.org/10.3390/rs14071625

Riess, T. C. V. W., Boersma, K. F., van Vliet, J., Peters, W., Sneep, M., Eskes, H., and van Geffen, J.: Improved monitoring of shipping NO2 with TROPOMI: decreasing NOx emissions in European seas during the COVID-19 pandemic, Atmos. Meas. Tech., 15, 1415–1438, https://doi.org/10.5194/amt-15-1415-2022, 2022.

---

## Author Response (AR1)

**Reply on Reviewer Comments**

We would like to thank all three reviewers for their helpful comments. We hope that we could address all questions and unclear points satisfactorily.

In the course of the revision, we have made the following important changes to the analysis:

Based on a suggestion from Referee#2 we have looked into the TROPOMI AOT product. We added daily maps of the TROPOMI AOT in the Appendix of the manuscript. The lower branch visible in the TROPOMI PAL versus AirMAP comparison is mainly caused by data from 17 September (Fig. A9) and was discussed to be likely caused by a higher aerosol load which is identified as cloud in the retrieval and not treated adequately in the cloud correction, ending up with too high cloud pressures. This discussion can now be supported by the TROPOMI AOT data, which is showing a high AOT over a large area on 17 September.

During the corrections in the review process we found that the tropospheric $NO_2$ VCD retrieval for the IUP car DOAS used an incorrect AMF of 1.5 instead of 1.3. This was corrected and Fig. 6 was updated. The correlation between the AirMAP and car DOAS measurements remains unchanged at 0.89, but the slope decreased from 0.98 to 0.89.

Referee#3 questioned the use of a $NO_2$ box profile for the AMF calculations for the AirMAP flights, which we have stated in the text. This was an outdated information which we overlooked during the correction phase. The SCIATRAN tropospheric AMF calculations used in the AirMAP tropospheric $NO_2$ VCD retrieval shown in the manuscript are not based on a 1 km box profile but are using a $NO_2$ profile based on an old WRF-chem model run following a more typical urban profile, scaled to the ERA5 boundary layer height, which reached typical values of 1 km around noon. The $NO_2$ profile is added to the Appendix.

Please find below the author's response to the three reviews.

**Comments from anonymous Referee #1:**

We would like to thank the reviewer for his/her helpful comments. We hope that we could address all questions and unclear points satisfactorily.

In the course of the revision, we have made the following important changes:

Based on a suggestion from Referee#2 we have looked into the TROPOMI AOT product. We added daily maps of the TROPOMI AOT in the Appendix. The lower branch visible in the TROPOMI PAL versus AirMAP comparison is mainly caused by data from 17 September (Fig. A8) and was discussed to be probably caused by a higher aerosol load treated as an effective cloud in the retrieval and not treated adequately in the cloud correction, ending up with too high cloud pressures. This discussion can now be supported by the TROPOMI AOT data, which is showing a high AOT over a large area on the 17 September.

During corrections in the review process it was found that in the tropospheric $NO_2$ VCD retrieval for the IUP car DOAS an incorrect AMF of 1.5 was used instead of 1.3, this was corrected and Figure 7 was updated. The correlation between the AirMAP and car DOAS measurements remains unchanged at 0.89, but the slope decreased from 0.98 to 0.89.

Referee#3 questioned the use of a $NO_2$ box profile for the AMF calculations for the AirMAP flights, which we have stated in the text. This was an out dated information and was overseen by us during

correction phase. The SCIATRAN tropospheric AMF calculations used in the AirMAP VCD retrieval shown in the manuscript are not based on a 1 km box profile but are using a NO2 profile based on an old WRF-chem model run following a more typical urban profile, scaled to the ERA5 boundary layer height, which reached typical values of 1 km around noon.

Legend: Referee comments in black, author comments in blue

This manuscript provides an evaluation of TROPOMI NO2 vertical tropospheric columns (VTCs) against airborne NO2 observations from the AirMAP imaging spectrometer, which itself is compared with ground-based stationary and mobile DOAS instruments.

The study is excellent. The manuscript is very well written and clear and the topic fits well within the scope of AMT. I can only commend the authors for this thorough study and strongly recommend publication.

Excellence:

The study goes well beyond previous evaluation studies in several respects. The validation experiment combining ground-based and airborne remote sensing is very well thoughtout: The ground-based measurements provide a high-quality reference and the airborne observations, which effectively sampled the area covered by individual TROPOMI pixels over regions with strong NO2 gradients, provide the link to the satellite data. This setup allows for a quantitative (almost 1:1) comparison in contrast to more indirect/qualitative comparisons previous studies. Furthermore, the extensive aircraft observations over three different regions with varying NO2 levels allowed covering many individual TROPOMI pixels necessary, which is necessary to compute robust statistics. Finally, the study does not stop at presenting the comparisons but goes a long way towards explaining the reasons for errors and biases in different versions of TROPOMI data. By replicating the TROPOMI retrieval algorithm, the authors were able to analyze the influence of key input parameters such as a priori NO2 profiles and surface reflectance on the data. The main source of error was found to be the FRESCO cloud retrieval, which tended to place the cloud tops at too low elevation probably due the inability of the algorithm to properly account for the effect of aerosols. This finding is essential to guide further developments of the retrieval algorithms and improvements of the operational TROPOMI NO2 product in the future.

The study is very well written with almost no typos or grammatical errors, logically organized, well balanced in terms of conciseness and detail, the figures and tables are of high quality, and the Appendices add valuable information.

We would like to thank the reviewer for his/her nice comments.

I have only two small points to consider:

The differences in the NO2 VTCs between the former operational offline algorithm (OFFL V01.03.02) and the improved new algorithm PAL V02.03.01 are very large. The authors mention that the main change was a switch from the FRESCO-S to the FRESCOwide cloud retrieval algorithm, but there is little information on what else changed what the (potential) influence of these changes were. It would be useful to get some more insight into the changes.

Thank you for the comment. We added more details about the changes from the old OFFL V01.03.02 to the new PAL V02.03.01 version with reference to the van Geffen et al. (2022) paper in the "TROPOMI NO2 PAL V02.03.01 product version" section.

"The main change compared to the OFFL V01.03.02 impacting the tropospheric NO2 VCD data is the use of the FRESCO-wide algorithm instead of the FRESCO-S algorithm, which was already introduced in V01.04 and was operational from 29 November 2020 to 1 July 2021. The FRESCO-wide algorithm provides lower and therefore more realistic cloud pressures (i.e. clouds are at higher altitudes), especially for scenes when cloud fractions are low. This change results in decreased tropospheric AMFs, which leads to higher tropospheric NO2 VCDs (van Geffen et al., 2022b). Another update that can have a significant impact is the correction of the surface albedo over cloud free scenes by using the observed reflectance. This increases the tropospheric NO2 VCDs by about 15% over polluted regions in case the retrieved cloud fraction is zero (van Geffen et al., 2022b). For this study the effect is negligible since only 1 out of the here analyzed 117 TROPOMI 210 pixels is observed as cloud free. van Geffen et al. (2022b) also describes the following other modifications, which have only a small or no impact on the tropospheric NO2 VCD data. Level-1b v2.0 (ir)radiance spectra are updated in the new version, and are increasing the NO2 SCD of about 3 %, from which most of it ends up in a slightly increased stratospheric VCD. The improved level-1b v2.0 also leads to a small increase of completely cloud-free pixels and to slightly lower cloud pressures for pixels with a small cloud fraction, resulting in tropospheric NO2 VCDs being about 5% higher for these ground pixels. An introduced outlier removal is increasing the amount of good quality retrievals over the South Atlantic Anomaly and over bright clouds where saturation can occur. The change to new spatially higher resolved snow and ice information is increasing the amount of valid retrievals at high latitudes."

The comparisons between AirMAP and the ground-based mobile and stationary suggest that the ground-based measurements (separately analyzed by the different groups) provide a consistent set of reference measurements. Nevertheless the question arises whether there has been no direct comparison between the ground-based instruments, e.g. when a car DOAS passed by a the location of a stationary instrument or when several car instrument were placed at the same location. If such intercomparisons have been made, it would be good to learn about them and add the results e.g. in an Appendix.

Thank you for the suggestion. We added a comparison plot of the three car DOAS instruments in the Appendix (see Fig. 1).

[Figure]

Figure 1: Scatter plot between collocated car DOAS measurements (± 5 min time window) of MPIC and BIRA car DOAS data versus IUP car DOAS tropospheric NO2 VCDs averaged within 200 m x 200 m grid boxes and 5 min time intervals. The data points from the MPIC and BIRA car DOAS instrument are color coded in green, respectively red. The thick solid black line represents the orthogonal distance regression. Error bars represent the error in the tropospheric NO2 VCD retrieval, averaged within the 200 m x 200m grid boxes and 5 min time intervals.

Small corrections:

Page 7, Line 148: Change "are retrieved" to "were retrieved"

done

Page 8, Line 179: I think the acronym DLER has not been introduced before.

Yes, this is right, we added the explanation here and deleted it later in the text.

Page 12, Equation 12: Why is the VDC_ref,trop not simply added to dSCD/AMF_trop? Why do we need to multiply VCD_ref,trop with AMF_ref,trop / AMF_trop? Please explain.

The dSCDs are slant colums retrieved relative to the reference measurement and the AMF of the actual measurement $AMF_{trop}$ and the reference measurement $AMF_{ref,trop}$ are not the same, so adding simply the $VCD_{ref,trop}$ to dSCD/ $AMF_{trop}$ would introduce an additional uncertainty. We added a comment in the manuscript.

"Since the AMF of the actual measurement ($AMF_{trop}$) and of the reference background measurement ($AMF_{trop, ref}$) are usually not the same, simply adding the $VCD_{trop}$, ref would introduce additional uncertainties."

Page 29, line 661: I think the recommended filter criterion of 0.5 applies to the cloud radiance fraction, not to cloud fraction.

Yes, right, this was a mistake. In the beginning we accidentally checked the cloud fraction instead of cloud radiance fraction to calculate the mean value. We thought we changed it everywhere in the text but have overseen it here.

On page 5 line 127 we have already written "The cloud radiance fraction retrieved in the TROPOMI NO2 spectral window (cloud_radiance_fraction_nitrogendioxide_window_crb) for S5P overpass times, was on average 0.21 ± 0.10 with a maximum of 0.48 and thus for all measurements below the recommended filter criterion of 0.5."

We have changed the text in line 661 accordingly.

Reference:

van Geffen, J., Eskes, H., Compernolle, S., Pinardi, G., Verhoelst, T., Lambert, J.-C., Sneep, M., Linden, M., Ludewig, A., Boersma, K. F., and Veefkind, J. P.: Sentinel-5P TROPOMI NO2 retrieval: impact of version v2.2 improvements and comparisons with OMI and ground-based data, Atmospheric Measurement Techniques, 15, 2037–2060, https://doi.org/10.5194/amt-15-2037-2022, 2022.

**Comments from anonymous Referee #2:**

We would like to thank the reviewer for his/her helpful comments. We hope that we could address all questions and unclear points satisfactorily.

In the course of the revision, we have made the following important changes:

Based on a suggestion from Referee#2 we have looked into the TROPOMI AOT product. We added daily maps of the TROPOMI AOT in the Appendix of the manuscript. The lower branch visible in the TROPOMI PAL versus AirMAP comparison is mainly caused by data from 17 September (Fig. A9) and was discussed to be likely caused by a higher aerosol load which is identified as cloud in the retrieval and not treated adequately in the cloud correction, ending up with too high cloud pressures. This discussion can now be supported by the TROPOMI AOT data, which is showing a high AOT over a large area on 17 September.

During the corrections in the review process we found that the tropospheric $NO_2$ VCD retrieval for the IUP car DOAS used an incorrect AMF of 1.5 instead of 1.3. This was corrected and Fig. 6 was updated. The correlation between the AirMAP and car DOAS measurements remains unchanged at 0.89, but the slope decreased from 0.98 to 0.89.

Referee#3 questioned the use of a $NO_2$ box profile for the AMF calculations for the AirMAP flights, which we have stated in the text. This was an outdated information which we overlooked during the correction phase. The SCIATRAN tropospheric AMF calculations used in the AirMAP tropospheric $NO_2$ VCD retrieval shown in the manuscript are not based on a 1 km box profile but are using a $NO_2$ profile based on an old WRF-chem model run following a more typical urban profile, scaled to the ERA5 boundary layer height, which reached typical values of 1 km around noon. The $NO_2$ profile is added to the Appendix.

Legend: Referee comments in black, author comments in blue

The manuscript by Lange et al. discusses the S5P-VAL-DE-Ruhr validation campaign. It includes a very extensive and well-presented validation of the TROPOMI dataset with aircraft and ground-based measurements, including a good measurement campaign overview. It further includes a comparison of different TROPOMI $NO_2$ datasets and shows the significant improvements of the latest product version PAL over OFFL. It is of interest to readers of AMT. I would recommend publication after addressing some suggestions (see supplement).

General suggestions:

Sect.3: Why are the fitting windows so variable from instrument to instrument (DOAS instruments)? Please comment on this in the instrument description. Looking at the comparison it doesn't seem to make much of a difference, but I would suggest commenting on this.

Thank you for the suggestion, we now included a comment on this in the instrument description. The fitting window is restricted by the spectrometer's wavelength ranges, which are different for the different instruments. The exact fitting window was either chosen to be close to the CINDI-2 fitting window (425-490 nm) or from the groups experience with their instruments. Differences using the different fitting windows were found to be small. Differences using the different fitting windows were found to be small.

Furthermore, we added additional columns with spectrometer wavelength range, fitting window, and information about the VCD calculation and AMF used to Table 2.

Sect. 3, make sure to include uncertainty estimates of all instruments and include references to existing validation papers. The sections 3.3.4, and 3.4.1 to 3.4.4 lack references.

Uncertainty estimates are included for all instruments in the respective figures.

We added references to studies/campaigns in which the instruments participated and some studies with more details about the data retrieval, as far as available.

l. 550-555: I would suggest to check the TROPOMI AOT in the area for that day, available from https://data-portal.s5p-pal.com/; e.g. you could include maps of this in the appendix or at the very least state the average (and std) for the flights. Alternatively, VIIRS AOD or MODIS MAIAC AOD are very good AOD products (but this might be a little too much effort to include here, maybe just something to keep in mind for future studies).

Thank you for your suggestion. We checked the TROPOMI AOT data during the campaign period for the different flight areas. We added daily maps of the TROPOMI AOTs in the Appendix (see Fig.1).

[Figure]

*Figure 1: Daily maps of TROPOMI AOT with qa_value > 0.5 (top row) and TROPOMI PAL V02.03.01 tropospheric NO2 VCDs with qa_value > 0.75.*

The observed AOT is quite variable, but it is good to see that there is no correlation between the TROPOMI AOT and NO2 data product. The lower branch visible in the TROPOMI PAL vs AirMAP comparison is mainly caused by data from 17 September and was discussed to be likely caused by a higher aerosol load which is treated as clouds in the retrieval and not accounted for adequately in the cloud correction, ending up with too high cloud pressures. This discussion can now be supported by the TROPOMI AOT data, which is showing a high AOT over a large area on the 17 September.

l. 619: just a thought: there was no snow during the campaign; there could be a larger difference between the DLER and LER TROPOMI product for snow covered surfaces with high reflectivity. It would be nice to include a little comparison of the DLER and LER product for snow covered surfaces. Nothing extensive, just a sentence (near l. 619) and a scatter plot (TROPOMI DLER vs TROPOMI LER) in the appendix (if time permits).

We added a sentence to clarify that differences between the DLER and LER TROPOMI product can be larger about snow-covered surfaces and that the found differences are only valid for the campaign area and time. We only have the data available for the campaign period and region. Nevertheless, we did some additional comparisons with the available data. Figure 2 shows the TROPOMI tropospheric NO2 VCD retrieved with TROPOMI LER respectively TROPOMI DLER for (a) the 117 TROPOMI pixels

used throughout the study in comparison with AirMAP, showing a slope of 1.04, with slightly higher NO2 VCDs for the TROPOMI NO2 product using the TROPOMI LER in the AMF calculations instead of the DLER. Figure 2b shows the same comparison for a larger orbit segment over western Europe on 13 September 2020 and Fig. 2c the complete orbit, both showing a correlation coefficient of 1 and a slope of 1.03. All data are quality and cloud filtered using a qa_value of 0.75. Since the observations are made in September, no larger snow covered areas are expected and a more detailed analysis including a larger period and area would be needed. We included Figure 2 in the Appendix of the manuscript.

[Figure]

Figure 2: Scatter plot of TROPOMI IUP V02.03.01 tropospheric NO2 VCDs with TROPOMI LER respectively TROPOMI DLER for: (a) the 117 TROPOMI pixels coinciding with the AirMAP measurements used throughout the study, (b) a larger orbit segment over western Europe on 13 September 2020 and (c) one full orbit including the campaign area on 13 September 2020.

Comment added in the manuscript: "Larger differences could for example be expected for snow-covered surfaces with high reflectivity. Figure A10 in the Appendix shows scatter plots of the TROPOMI tropospheric NO2 VCD retrieved with TROPOMI LER and TROPOMI DLER for the 117 TROPOMI pixels used throughout the study but also for larger areas up to one full orbit. All comparisons show only minor influences by the directional component. Since only TROPOMI observations made in September are compared, no larger snow-covered areas are expected and a more detailed analysis including a different period and area would be needed to investigate possible larger differences."

Technical/minor suggestions:

l. 1: suggesting to change to "Airborne, ground-based stationary and car imaging differential optical absorption spectroscopy (DOAS) measurements…"

Since not all mentioned measurements are imaging, we did not implement this proposed change.

l. 4: emitters -> sources

Changed

l. 5: "The DOAS measurements…"

Changed

l. 7… suggest using "observations" instead of measurements throughout the text (technically measurements are in situ measurements), and remote-sensing are observations

Thank you for the comment. Since it is common in the community to use the term measurements also for the here described cases we stayed with this usage but tried to change it to observations in some more specific descriptions.

l.13: data create

Changed

l. 13-31: This paragraph can be shortened, I think the most important points are: 1) The PAL version improves the bias significantly, 2) cloud height and NO2 profile have a major impact, 3) surface reflectivity has a minor impact (in this region and time of year – this is likely different when there is snow on the ground). These points get a little lost in the lengthy paragraph, maybe include the correlations and biases inside the sentence in brackets rather than writing whole sentences about it (a little repetitive).

Thank you for the suggestion. We slightly shortened this paragraph to highlight the major findings.

l.36: change to "…combustion processes, such as power plants and engines, as well as anthropogenic biomass burning." (How much does anthropogenic biomass burning impact the Ruhr area? Consider removing the last half of the sentence.)

It is right that anthropogenic biomass burning has not a large impact in the Ruhr area, but since in this section of the introduction the sources of NOx are described more in general, we left it like this.

l. 37: "NOx is primarily emitted as NO, the reaction…"

Changed to "NOx is primarily emitted as NO, which is reacting with ozone (O$_3$) and is rapidly forming NO$_2$."

l.37: "The NOx sources are …" (remove the characteristics)

Done

l. 38: chemically active -> reactive and short lived

Changed

l. 38/39: "…, there is a high spatial and temporal variability of NO2 near emission sources." (there is not much variation in background areas)

Changed to "As a result, the spatial and temporal variability of NO2 is large, especially in regions characterized by a variety of NOx emission sources."

l. 39: remove "on"

Done

l.44: "is remotely observed from different platforms" -> " can be observed remotely on a variety of platforms"

Changed to "is remotely observed on a variety of platforms"

l.46f: is identified -> can be identified; are -> can be

Done

l. 53: remove "on board the European…satellite."

Removed

l. 55 TEMPO is planned for launch in March 2023

Changed it to 2023.

l. 56-58: consider re-wording this sentence, maybe change it to 2 shorter sentences

Rewritten into two sentences.

l. 86: include a sentence about the new TROPOMI version and what changed/improved in comparison to the previous version. Maybe include studies that validated this new version, I know of Zhao et al. (2022) see reference list, there might be others too. E.g. Riess et al (2022) also talks about the improvements of the new TROPOMI version.

We added a few sentences about the new TROPOMI version and included the study of van Geffen et al. (2022) which is also comparing the new tropospheric NO2 VCD version with MAX-DOAS data. We have not included the suggested studies since Zhao et al. (2022) is doing validation only with total columns and Riess et al. (2022) is using the new version for monitoring shipping NOx emissions but is not including validation.

l. 89: industrial estates -> industrial facilities, arterial highways -> busy highways (or large highways)

Included the first suggestion, second left like it is.

l. 89: "Back-ground areas with low pollution, as well as moderately polluted regions are also observed…"

Changed

l. 96: remove "In the following,"

Done

l. 107: 5 million inhabitants -> has a population of 5 million

"The Ruhr area itself has over 5 million inhabitants." changed to "The Ruhr area itself has a population of 5 million."

l. 108f: "The region, including nearby metropolitan centres along the Rhine and populated surroundings is called Metro… and is comprised of a population of over 10 m, large power plants [can you include a number here], … industrial facilities and several large highways."

Thank you for your suggestions. We changed it to:

"Together with the populated surroundings and metropolitan centers along the Rhine, the region is called Metropolitan area Rhine-Ruhr (MRR). It comprises a population of more than 10 million inhabitants, large power plants, energy intensive industrial facilities and several large highways."

l. 110: above the campaign location -> in the MRR

Changed

l.120:"… dominated by the emissions of three lignite fire power plants in the area (see European Pollutant Release and Transfer Register (E-PRTR)[include reference or url here]).

Changed to "The research flight area around Jülich is expected to be dominated by the emissions of three large lignite fired power plants located in the area (see European Pollutant Release and Transfer Register, https://industry.eea.europa.eu/, last access: 18 November 2022)"

l. 121: "…around Cologne and Duisburg…" , remove latter part of the sentence "and the flight area around Duisburg has a similar character to that of the Cologne area with a mixture of urban and industrial emitters but includes the central metropolitan Ruhr area, which has a large variety of pollution sources. "

Shortened but kept the information about the difference between the two areas.

l. 137: here and any other occurrences: don't shorten Table to Tab. as per AMT guidelines, "are given in Table 1"

Done

l. 143: "…comparison of the aircraft and TROPOMI NO2…"
l. 143: remove "prior to the dedicated evaluation … satellite pixel area", I'm not sure what you mean here, but I think it's not necessary

Rewritten this passage. We hope it is clearer now.

l. 144f: remove: "In this manner", " on the one hand", "on the other hand"

Removed

l. 145: local -> ground-based , with restricted -> that have restricted, with satellite -> to satellite

Changed

l. 151: Table 2

Changed

l. 157: of aerosol -> aerosol

Changed

l. 160: thus by far -> currently

Changed

l. 167: AMFs. The AMFs are generated using the OMI…

Not changed, as we think it would change the meaning in a wrong way.

l. 167: , and cloud fraction

Changed

l. 171: (e.g. Verholst et al, 2021). (there are many others)

Changed

l. 173f: very long sentence, consider numbering the reasons and removing unnecessary details: " other factors that could contribute are: (1)…"

Thank you for your suggestion we restructured a bit and numbered the reasons.

l. 186: recipe provided-> approach as detailed

Changed to "approach described".

l. 231: Tab.->Table

Done

l. 255: remove "used"

Done

l. 261: 438-490nm; is this the same or different to TROPOMI?, and other DOAS fits used in this study

The AirMAP fitting window is different from the TROPOMI fitting window. Due to different spectrometer wavelength ranges it is not possible to use the same fitting window.

TROPOMI: 310 - 500 nm wavelength range, fitting window: 405 - 465 nm

AirMAP: 429 - 492 nm wavelength range, fitting window: 438 - 490 nm

We included a comment on the different fitting windows in the instrument description. See comment in general suggestions.

l. 265ff: it's unusual to trop as a superscript, subscripts would be more common, e.g. subscript of trop,ref could be used

Changed

l. 271: during -> near

Changed to "close to".

l.272:change to: " There is a maximum difference of 3h between the time of the reference background and the actual measurements."

We changed the sentence.

l. 276: how small, do you have a reference that it is negligible?

We have written "We assume that the effect of the changing solar zenith angle (SZA) and the diurnal variation of the stratospheric NO2 concentration are small, and a stratospheric correction of the measurement data is therefore not necessary." We added that this is only true during the measurement time around noon and not during twilight and included a reference to Schreier et al. 2019, where this was analyzed for car DOAS measurements in Vienna (see Fig. 3).

Schreier et al. 2019: "The uncertainty of the diurnal variation is large at twilight but small during the day as changes in stratospheric NO2 are small when compared to tropospheric NO2 columns in polluted regions, such as the urban area of Vienna. As a rough estimate, the uncertainty of the stratospheric correction is assumed to be less than 10% or typically $1\times10^{15}$ molec/cm$^2$."

[Figure]

Figure 3: Figure 5 from Schreier et al. (2019): Stratospheric NO2 above Vienna on 19 October 2014 (red line) as obtained from the Bremen 3d chemistry transport model (B3dCTM). The green rectangle indicates the time period of car DOAS measurements performed on that day.

l. 279: (see ERA5 reanalysis; Hersbach et al. (2018). You can include the data source in the data availability section, and data source reference

Changed

l. 298: comprises of

Not included

l. 300: remove "further details can be found therein."

We removed this part.

l. 352: why is the spectral window different to the AirMAP window?

We included a comment on the different fitting windows in the instrument description. See comment in general suggestions.

l. 371/379: again why is the spectral window different?

We included a comment on the different fitting windows in the instrument description. See comment in general suggestions.

l. 397: "different target areas" -> "three target areas"

Changed

l. 411: do you have a reference for this? This paragraph in general could benefit from a couple of reference.

We wrote: "An uncertainty of 30% for the SCD in the reference spectrum is assumed". Unfortunately we do not have a reference for the given uncertainty.

We inserted a reference for the instrument setup and the Langley-plot method.

l. 413: The same AMF (1.3) is used to convert to a VCD, wouldn't it depend on the SZA? How much of an impact does the SZA have?

We agree that the AMF depends on the SZA. Since we only used data close to the AirMAP overpass, which was performing measurements around noon, the SZA is not varying much. In the paper we have referred to the work of Merlaud (2013), who analyzed the AMF distribution for a large number of simulations, resulting in a mean of 1.33 ± 0.2 and 2.52 ± 0.32 for the AMF for measurements in 90° respectively 30° viewing zenith angle (see Fig. 4 and Table 1).

[Figure]

*Figure 4: From Fig. 7.3 from Merlaud (2013): Distribution of air mass factors calculated with the parameters of table 7.1(left) and of the resulting differential air mass factor (right).*

*Table 1: Table 7.1 from Merlaud (2013): Parameters and ranges used in the air mass factors calculations.*

| Wavelength(nm) | 460 |
|---|---|
| Direction | Zenith,30° |
| Surface visibility(km) | 5, 10, 15, 20, 25, 30 |
| $NO_2$ mixing height(m) | 100, 300, 500, 700, 900 |
| Relative azimuth(°) | 0, 30, 60, 90, 120, 150, 180 |
| Solar zenith angle(°) | 20,30, 40, 50, 60, 70, 80, 90 |
| Albedo | 0.01, 0.03, 0.05, 0.07, 0.09, 0.11, 0.13, 0.15 |

l. 423: insert reference here

We included a reference.

l. 440-450: what is the uncertainty of the PANDORA observations. Also please include references in this paragraph.

We included additional references to the Pandora instrument and data product in this paragraph. The uncertainty provided with the tropospheric NO2 VCD is described in Cede et al. (2021), we added this reference to the text.

l. 465: is it really +- 10-90 percentile, I think it is just the 10th and 90th percentile.

Thank you for pointing this out, we have changed it to "the error bars are representing the 10th and 90th percentile" throughout the text.

l. 470: data is -> data are

Changed

l. 473: As a result of having more opportunities to make near simultaneous synchronized measurements, -> Consequently,

Included

l. 487: is it really +- 10-90 percentile, I think it is just the 10th and 90th percentile.

Thank you for pointing this out, we have changed it to "the error bars are representing the 10th and 90th percentile" throughout the text.

Fig. 7: I would suggest moving one of these (I suggest Fig. 7b) to the appendix, as they essentially show the same.

Thank you for the suggestion. We removed Fig. 7b completely since we only used it here to point out the time difference between the AirMAP and car DOAS measurements as a reason for the outlier. We added a reference to Fig. A 7 in the Appendix, in which we kept both plots.

l. 637: areas -> areas,

Included

l.642: scatter -> scatter,

Included

l.642: measurements -> measurements,

Included

l. 687: dataset -> data set (either or but be consistent, I found both dataset and data set used in the study, please fix this)

Changed "dataset" to "data set" throughout the text.

l. 704: Sentinel-4 -> Sentinel-4,

Included

References:

Zhao, X.; Fioletov, V.; Alwarda, R.; Su, Y.; Griffin, D.; Weaver, D.; Strong, K.; Cede, A.; Hanisco, T.; Tiefengraber, M.; McLinden, C.; Eskes, H.; Davies, J.; Ogyu, A.; Sit, R.; Abboud, I.; Lee, S.C. Tropospheric and Surface Nitrogen Dioxide Changes in the Greater Toronto Area during the First Two Years of the COVID-19 Pandemic. Remote Sens. 2022, 14, 1625. https://doi.org/10.3390/rs14071625

Riess, T. C. V. W., Boersma, K. F., van Vliet, J., Peters, W., Sneep, M., Eskes, H., and van Geffen, J.: Improved monitoring of shipping NO2 with TROPOMI: decreasing NOx emissions in European seas during the COVID-19 pandemic, Atmos. Meas. Tech., 15, 1415–1438, https://doi.org/10.5194/amt-15-1415-2022, 2022.

Cede, A., Tiefengraber, M., Gebetsberger, M., and Spinei Lind, E.: Pandonia Global NetworkData Products Readme Document, Tech. rep., PGN-DataProducts-Readme, version 1.8-5, 31 December 2021, last access: 2 December 2022, 2021.

Merlaud, A.: Development and use of compact instruments for tropospheric investigations based on optical spectroscopy from mobile platforms, Presses univ. de Louvain, 2013.

Schreier, S. F., Richter, A., and Burrows, J. P.: Near-surface and path-averaged mixing ratios of NO2 derived from car DOAS zenith-sky and tower DOAS off-axis measurements in Vienna: a case study, Atmospheric Chemistry and Physics, 19, 5853–5879, https://doi.org/10.5194/acp-19-5853-2019, https://acp.copernicus.org/articles/19/5853/2019/, 2019.

**Comments from anonymous Referee #3:**

We would like to thank the reviewer for his/her helpful comments. We hope that we could address all questions and unclear points satisfactorily.

In the course of the revision, we have made the following important changes:

Based on a suggestion from Referee#2 we have looked into the TROPOMI AOT product. We added daily maps of the TROPOMI AOT in the Appendix of the manuscript. The lower branch visible in the TROPOMI PAL versus AirMAP comparison is mainly caused by data from 17 September (Fig. A9) and was discussed to be likely caused by a higher aerosol load which is identified as cloud in the retrieval and not treated adequately in the cloud correction, ending up with too high cloud pressures. This discussion can now be supported by the TROPOMI AOT data, which is showing a high AOT over a large area on 17 September.

During the corrections in the review process we found that the tropospheric NO2 VCD retrieval for the IUP car DOAS used an incorrect AMF of 1.5 instead of 1.3. This was corrected and Fig. 6 was updated. The correlation between the AirMAP and car DOAS measurements remains unchanged at 0.89, but the slope decreased from 0.98 to 0.89.

Referee#3 questioned the use of a NO2 box profile for the AMF calculations for the AirMAP flights, which we have stated in the text. This was an outdated information which we overlooked during the correction phase. The SCIATRAN tropospheric AMF calculations used in the AirMAP tropospheric NO2 VCD retrieval shown in the manuscript are not based on a 1 km box profile but are using a NO2 profile based on an old WRF-chem model run following a more typical urban profile, scaled to the ERA5 boundary layer height, which reached typical values of 1 km around noon. The NO2 profile is added to the Appendix.

Legend: Referee comments in black, author comments in blue

This manuscript uses DOAS data collected in September 2020 in a polluted region in western Germany from airborne, ground, and car-based instrumentation to validate the set of TROPOMI L2 NO2 products (both research and operational). The airborne datasets are first validated by the ground and car-based systems which then justifies the airborne use for validating TROPOMI. This paper fits the scope of AMT and will be valuable as a validation dataset for the TROPOMI NO2 product. However, before publishing, this manuscript requires some minor technical corrections/clarifications as detailed below but more reflection toward conclusions drawn about improvements in the S5P PAL product and the impact of clouds. Detailed comments below.

More significant comments:

Most of the results in this work are too heavily based on the slope of the regression, which is not representing the complete behavior of the validation activity. Table A1 has at least median difference in % which actually in some cases contradicts the results of the slope (e.g., having a 21% higher column from TROPOMI as a median from S5P PAL V02.03.01.). Consider more in-depth analysis based on statistics other than slope for all intercomparisons.

Thank you for the comment. We have included the median differences given in Table A1 with some additional comments in the text analyzing the comparisons and added a Figure with Box-and-whisker plots summarizing the bias and spread of the difference between the different TROPOMI versions and AirMAP tropospheric NO2 VCDs in the Appendix (see Fig. 1).

[Figure]

*Figure 1: Box-and-whisker plots summarizing the bias and spread of the difference between the different TROPOMI versions and AirMAP tropospheric NO2 VCDs. The green line inside the box represents the median difference. Box bounds mark the 25 and 75 percentiles while whiskers represent the 5 and 95 percentiles.*

Some conclusions drawn in section 6 are either overgeneralized or not quite technically correct. These comments do not specify lines in the text but more so in general comments that need to be kept in mind when adding to and editing the analysis based on the suggestions below:

With the data presented, conclusions about the S5P PAL product are only stated as an improvement. This is an overgeneralized conclusion, and the authors should do some more detailed analysis from other statistics. Some of this is already done with discussion of the lower lobe results but it is missing discussion on the higher lobe. Additionally, with the loss in precision, some users may find this result more detrimental than having a predictable low slope and this is not commented upon in the results, abstract, or conclusions.

Thank you for pointing this out. We included comments in the abstract, results and conclusion section highlighting that while the slope improved with the PAL product, the correlation of the data has decreased.

"With the modifications in the NO2 retrieval implemented in the PAL V02.03.01 product the slope and median relative difference increased to 0.83±0.06 and +20 %. However, the modifications resulted in larger scatter and the correlation decreased significantly to r = 0.72."

"The comparison of this TROPOMI product PAL V02.03.01 with the AirMAP data in Fig. 9c shows much more scatter with a correlation coefficient which is significantly poorer than for the OFFL V01.03.02 product, changing from 0.86 to 0.76. The slope, however, increased by more than a factor of 2 from 0.38±0.02 to 0.83±0.06, demonstrating that the updates in the new TROPOMI NO2 data version have a large impact on the analyzed data set from the Rhine-Ruhr region. Due to the large scatter and driven by the large amount of measurements with tropospheric NO2 VCDs of about less than $7 \pm 0.15 \cdot 10^{15}$ molec cm$^{-2}$ the PAL V02.03.01 product has a positive median relative difference of 20% with an interquartile range of -14% to 66% (see Fig. A11)."

In this context we also changed the often used "improved retrieval" to a more neutral form like "modifications/updates in the retrieval". The higher lobe is not as much discussed as the lower lobe since we could not identify completely what is causing this higher lobe, except that it is reduced for the TROPOMI data version without cloud correction (see Sect. 6.1). Nevertheless, we added additional comments highlighting this in the results and conclusion section.

The main reason concluded about the improved PAL product is due to the cloud correction. It is stated that all these changes are due to more 'realistic' cloud pressures or more 'realistic' cloud corrections but this more 'realistic' outcome is not demonstrated in this region. Therefore, these conclusions cannot be stated unless they are proven with the data available in that specific region (e.g., could look at imagery from satellites or other creative sources and reflect on what it should be in reality). In fact, removing the cloud correction all together (Figure 9b) shows that the massive improvement in slope is something else removed from the cloud correction as this is the best result in terms of conserving precision (correlation) and a higher slope.

Previous studies showed that for scenes with low clouds, i.e. close to the surface, a height that is even closer to the surface was retrieved by the original FRESCO implementation. Since the algorithm does not discriminate between clouds and aerosols, this also holds for low aerosol layers. In many cases, FRESCO then retrieves the surface height, which is incorrect (Compernolle et al., 2021, van Geffen et al., 2022).  In the old OFFL V01.03.02 product, 110 out of 117 pixels and thus 97 % of the TROPOMI observations were found to have cloud heights very close to the surface (within 50 hPa), which is not realistic and especially not for such a large amount of observations. In the new PAL product, the cloud retrieval yields only for 23% of the observations (28 out of 117 pixels) a cloud height close to the surface, which can be considered more physically realistic, resulting in a better slope of the regression line. However, since some scenes remain problematic, more scatter results. See also Figure A8 and A9 in the manuscript Appendix which show daily TROPOMI versus AirMAP scatter plots with points color coded by the difference of the surface and cloud pressure.

VIIRS images of the campaign measurement days support the on flight observations of nearly cloud free conditions during the measurement flights over the target areas. "Clouds" detected in the cloud retrieval must therefore be aerosols, which are treated as clouds in the cloud correction. For nearly cloud free observations, the cloud correction is more an aerosol correction. Whether the cloud correction actually improves the NO2 results in the presence of aerosols depends on the details of the vertical distributions of aerosols and NO2. Therefore, in some cases, the results can be better if no cloud correction is made see Fig. 9 (now Fig. 10) in the manuscript.

We have added/highlighted the mentioned points in the results and conclusion of the manuscript. We hope it is now more comprehensible.

Conclusions drawn about the cloud pressure in some cases seems to not be interpreted correctly as written. For example, discussions from line 555-563 talk about the low lobe. (1) It is stated that cloud pressures are too low, but looking at imagery online there seems to be zero clouds seen by VIIRS on this afternoon, so cloud pressures shouldn't be low to start with. (2) Aerosols are also pointed at as a potential cause, but the sensitivity results in Figure A2 show that the impact of aerosols would not be large enough to create this bias in this lobe.

Yes, the campaign days have been nearly cloud free, but since the cloud algorithm does not discriminate between clouds and aerosols, aerosols are treated as clouds in the retrieval and we have to discuss the retrieved „cloud" pressures also for these cases.

Based on a suggestion from Referee#2 we have looked into the TROPOMI AOT product. We added daily maps of the TROPOMI AOT in the Appendix. The lower branch visible in the TROPOMI PAL versus AirMAP comparison is mainly caused by data from 17 September (see Fig. A9) and was discussed to be likely caused by a higher aerosol load which is treated as clouds in the retrieval and not corrected for adequately by the cloud correction, ending up with too high cloud pressures. This discussion can now be supported by the TROPOMI AOT data, which is showing a high AOT over a large area on the 17 September.

Figure A2 (now Fig. A3) shows the impact of aerosols on the AirMAP retrieval, not on the TROPOMI retrieval. Due to different observation heights, the TROPOMI retrieval is expected to be more sensitive to aerosols than the AirMAP retrieval. Also, the effect discussed here is introduced by the TROPOMI cloud correction algorithm which is not applied to the AirMAP data.

We have added/highlighted the mentioned points in the results and conclusion of the manuscript. We hope it is now more comprehensible.

**Technical comments in relation to the AirMAP retrieval that need more justification or clarification.**

It is said that the reference VCD in the troposphere for AirMAP is 1e15. One of the MAX-DOAS retrievals has a different value of 1.5E15 but they are referred to as similar. It is different by 50% rather than similar. Please clarify these difference or explain them.
Can the reference value be justified with any other data from this work? (i.e., What does the CAMS model say the tropospheric amount is?)
Could this reference assumption be the cause for a low offset between the car DOAS systems and the airborne dataset?

Thank you for pointing this out. Since there is no reason for using different reference values, we have decided to use the value of $1 \times 10^{15}$ molec/cm$^2$ for both, the AirMAP and IUP car DOAS tropospheric VCD retrieval. The other car DOAS instruments do not rely on this value as they use dedicated measurements taken at lower elevation angle to directly estimate the tropospheric column in the reference measurement.

The influence of the mentioned difference of $0.5 \times 10^{15}$ molec/cm$^2$ is not very large and cannot explain the offset between the car DOAS and AirMAP dataset, respectively only a very small part of it. Figure 2 shows scatter plots of AirMAP versus car DOAS comparisons. The AirMAP tropospheric NO2 VCDs are retrieved with a VCD$_{trop, ref}$ of $1 \times 10^{15}$ molec/cm$^2$ for both plots. The IUP car DOAS data are retrieved with (a) $1 \times 10^{15}$ molec/cm$^2$ and (b) $1.5 \times 10^{15}$ molec/cm$^2$.

[Figure]

(a)                                    (b)

*Figure 2: Scatter plots of AirMAP versus car DOAS tropospheric NO2 VCDs. The AirMAP tropospheric NO2 VCDs are retrieved with a VCD$_{ref}$ of 1 x 10$^{15}$ molec/cm$^2$ for both plots. The IUP car DOAS data are retrieved with (a) 1 x 10$^{15}$ molec/cm$^2$ and (b) 1.5 x 10$^{15}$ molec/cm$^2$.*

Due to larger differences between the CAMS model and TROPOMI respectively AirMAP tropospheric NO2 VCD in distribution and amount (see paper Fig. 8 and Fig. A1) we have decided not to use the CAMS model data for the determination of the VCD$_{ref}$. Instead we checked the TROPOMI tropospheric NO2 VCD closest in time and space to the AirMAP reference measurement. Figure 3 shows the daily

maps of TROPOMI and AirMAP tropospheric NO2 VCDs. The red cross marks the location over which AirMAP took the reference measurement. The pink cross marks the TROPOMI pixel covering this reference area. Using these TROPOMI observations would yield in a $VCD_{ref}$ of $4.1 \pm 1.5 \times 10^{15}$ molec/cm$^2$. Due to the time difference between the AirMAP reference measurement and the TROPOMI observation, variations are expected and often pixel with lower values can be found close to selected pixel.

[Figure]

*Figure 3: Daily maps of TROPOMI and AirMAP tropospheric NO2 VCDs. Red crosses mark the location over which AirMAP took the reference measurement. Pink crosses mark the TROPOMI pixel covering this reference location.*

Since the TROPOMI data are indicating a higher value for the $VCD_{ref}$, we recalculated the IUP car DOAS data with a $VCD_{ref}$ of $3.13 \times 10^{15}$ molec/cm$^2$. Figure 3 shows scatter plots of collocated car DOAS measurements with IUP car VCDs retrieved with (a) $VCD_{ref} = 1.0 \times 10^{15}$ molec/cm$^2$ and (b) $VCD_{ref} = 3.13 \times 10^{15}$ molec/cm$^2$. The MPIC and BIRA car DOAS tropospheric NO2 VCDs are determined independently with their additional off-axis measurements in 22° respectively 30° as described in the corresponding instrument sections. As illustrated in Fig. 4, the IUP car DOAS VCDs calculated with the larger $VCD_{ref}$ of $3.13 \times 10^{15}$ molec/cm$^2$ are causing a significantly larger offset of $-2.27 \times 10^{15}$ molec/cm$^2$ than with the $VCD_{ref}$ of $1 \times 10^{15}$ molec/cm$^2$. Based on this comparison the IUP car DOAS and AirMAP tropospheric NO2 VCD calculations are based on the $VCD_{ref}$ of $1 \times 10^{15}$ molec/cm$^2$.

[Figure]

*Figure 4: Scatter plot between collocated car DOAS measurements (5 min time window) of MPIC and BIRA car DOAS data against IUP car DOAS tropospheric NO2 VCDs averaged within 200 m x 200 m grid boxes and 5 min time intervals. IUP car DOAS VCDs are retrieved with (a) $VCD_{ref} = 1.0 \times 10^{15}$ molec/cm$^2$ and (b) $VCD_{ref} = 3.13 \times 10^{15}$ molec/cm$^2$.*

We added a comment in the manuscript, discussing the remaining offsets in the comparisons of AirMAP versus car and stationary data:

"The comparison shows an offset of $-1.29 \pm 0.15 \cdot 10^{15}$ molec cm$^{-2}$. This offset could be adjusted to be closer to zero by increasing the estimated VCD$_{trop, ref}$ in the AirMAP retrieval by more than a factor of 2. However, the offset in the comparison of AirMAP and ground-based stationary data of $1.16 \pm 0.15 \cdot 10^{15}$ molec cm$^{-2}$. is positive instead of negative, and a larger VCD$_{trop, ref}$ in the AirMAP retrieval would further increase this offset. Because of this, and a lack of justification for a large difference between the VCD$_{trop, ref}$ for the car and AirMAP retrieval, we chose to leave the VCD$_{trop, ref}$ as it is."

Can the authors justify why a 1km box profile used if CAMS analysis is available for these flights to provide a profile shape and what that assumption impact may be in the results? A 1 km box profile assumes that NO2 is well mixed through that 1km boundary layer which has been demonstrated as not the case with in situ measurements from aircraft near strong sources (which is the case here in many of these flights). (e.g., https://doi.org/10.1002/2015JD024203 and https://doi.org/10.1525/elementa.2020.00163 ). This paper also shows the impact of AMFs based on assuming a 1km box vs an urban profile atmos-meastech.net/3/475/2010/

Thank you for pointing out this mistake. This was an outdated information and was missed by us during correction phase. The SCIATRAN tropospheric AMF calculations used in the AirMAP VCD retrieval are not based on a 1 km box profile but are using a NO2 profile based on an old WRF-chem model run scaled to the ERA5 boundary layer height, which reached typical values of 1 km around noon (see Fig. 5). This assumed profile is following very well the modeled and in-situ aircraft profiles from the DISCOVER-AQ campaign 2011 (Zhang et al., 2016) and the averaged urban NO2 profile from CHIMERE model runs shown in Leitao et al. (2010). We have changed the text accordingly.

[Figure]

*Figure 5: NO2 profile used in the SCIATRAN AMF calculations for the AirMAP measurement flights. The profile is based on WRF-Chem model runs and scaled to the typical boundary layer height during the measurement days around noon.*

Line 291: 'Surfaces with different brightness introduce artefacts in the maps of NO2'. The impact isn't necessarily an artifact at the SCD stage. This is caused by the brighter surface increasing sensitivity in the lower parts of the atmosphere meaning a higher slant column if NO2 is present (if there is not any or minimal NO2 then this spatial pattern will not show up in the slant column). It only becomes an artifact if the surface reflectivity assumption in the AMF calculation doesn't account for this accurately.

Thank you for pointing this out, we have rewritten it and hope that it is clearer now.

"Bright surfaces enhance the relative contribution of light reflected from the surface to the signal received by the airborne instrument, increasing the sensitivity to NO2 near the ground. Therefore, areas of high surface reflectance in the fitting window generally show larger dSCDs for the same amount of NO2. Thus, differences in the surface reflectivity must be accounted for in the AMF calculations."

Minor comments:

When referring to the spatial resolution of TROPOMI as 3.5 km x 5.5 km, please specify that this is at nadir.

Thank you for the comment, we included that the resolution is given for nadir observations.

Line 74. Mention what version Verhoelst et al. validated to be consistent with this analysis and the other mentioned publications.

Done.

Line 94: the conclusion of 'low bias' is prematurely stated (before showing any results). Recommend just removing 'low' from the sentence.

Done.

Figure 2 is mentioned before Figure 1. Consider reordering figures to reflect this or consider combining Figures 1 and 2 for a more helpful side-by-side comparison.

We moved Fig. 2 before Fig. 1.

Line 159: capitalized Ozone Monitoring Instrument

Done.

Line 179-181: The sentence about V02.04.01 should either clearly state that this analysis does not include this product or should be removed.

We added a statement that this version is not included and discussed in this study since it is not yet reprocessed and thus not available for the campaign period.

Lines 173-177: The following sentence needs references: 'Other factors that could contribute to the underestimation are the low spatial resolution of the used a priori NO2 profiles from the TM5-MP global chemistry transport model, the use of the OMI LER climatology given on a grid of 0.5° x 0.5° for the AMF and cloud fraction retrieval in the NO2 fit window and the GOME-2 LER climatology used for the NIR-FRESCO cloud retrieval given on a grid of 0.25° x 0.25° measured at mid-morning.'

We added references to this paragraph.

Line 189: add the spatial resolution of the CAMS global analysis

Added the CAMS global resolution of 0.4° x 0.4° to the text.

Line 198-199: The sentence referring to 15% increases needs a reference.

Added the reference to van Geffen et al. (2022).

Line 308: define quantitatively what polluted means for this statistic.

We added the mean dSCD and mean dSCD error value in the text.

Equation 5 seems to be the same as equation 4. Is it needed?

Yes, this is right, we have deleted Eq.5 and are now referring to Eq. 4.

Consider making a table of all the various information of the retrievals for the AirMAP, car, and stationary DOAS retrievals as the sections get repetitive and there are small differences in places that are hard to keep straight.

Thank you for the suggestion, we added additional columns with spectrometer wavelength range, fitting window, and information about the VCD calculation and used AMF to Table 2, hopefully giving a better overview of all instruments and retrievals.

Are there references for all the individual car or ground-based systems? If so, please add in the sections that describe them.

We added references for the individual car or ground-based instruments or at least to a very similar setup as far as available.

The MAX-DOAS measurement truck is different from the rest in that it measures in the UV rather than the visible wavelengths of the other retrievals. Is it realistic for their AMFs to be the same as the other systems?

Thank you for pointing this out. We did some radiative transfer calculations using the following parameters, which are adjusted to the ground-based and AirMAP comparison times around noon regarding SZA and typical albedo and AOT values found during the campaign measurement days. Based on these calculations the dAMF in the UV is closer to 1.1 instead of the assumed 1.2 (see Fig. 6). Thus, we recalculated the MAX-DOAS measurement truck VCDs, updated the AirMAP versus stations scatter plot and the AMF information in the manuscript.

*Table 1: Parameters and ranges used in the AMF calculations for ground-based measurements.*

| | |
|---|---|
| Wavelength (nm) | 350, respectively 460 |
| Viewing zenith angle (°) | 90, respectively 30 |
| Relative azimuth angle (°) | 0, 45, 90, 135, 180 |
| SZA (°) | 40, 50 |
| Albedo | 0.01, 0.02, 0.03, 0.04, 0.05 |
| AOT | 0.0015, 0.16, 0.31, 0.47, 0.62 |
| $NO_2$ profile | typical urban profile scaled to 1 km boundary layer height (see Fig. A3) |
| Aerosol profile | 1.5 km box profile |

[Figure]

*Figure 6: Distribution of AMFs calculated with the parameters of Table 1 for a wavelength of 350 nm (a) and 460 nm (b) for 90° VZA (dark color) and 30° VZA (light color).*

Line 415-416. The SCD of the reference for this DOAS instrument seems quite large considering the statements that the AMFs for a zenith DOAS retrieval are about 1.3. Is this off by an order of magnitude or are the measurements just in a densely polluted area for the reference?

The $SCD_{ref}$ given here includes the stratospheric and tropospheric NO2 in the reference spectrum. The different AMFs (tropospheric and stratospheric) and the stratospheric and tropospheric columns contributions must be considered. In order to estimate the tropospheric VCD from this value, it must be taken into account that the reference was taken during summer and therefore a relatively large part is stratospheric NO2.

Line 449-451. Is there a reference for the tropospheric NO2 product from Pandora that can be added to this section? This is the first publication I have seen use that product.

To our knowledge, there are no publications yet using the Pandora tropospheric NO2 product. We have added a reference to the Pandora readme document (Cede et al., 2021), which to our knowledge is the only document with further information on the relatively new tropospheric NO2 product.

Line 501: Is it +/- 1 hour or 30 minutes? The rest of the paper seems to reflect 30 minutes.

For the comparisons only data +/- 30 min around the S5P overpass are used. The 1 hour was given as an optimal measurement time over the target area around the S5P overpass, providing the option to adjust and investigate the effect of the temporal collocation criteria. We have restructured this paragraph to make this clearer.

Line 576: Before this line, it says that the criterion for comparison is the same as Judd et al. 2020 but at this location the authors should specify that this criterion (filtering for delta CS less/greater than 50 hpa) is the opposite of the filter applied by Judd et al. to avoid confusion. Bonus suggestion: it could be nice to have a comparison of what the results look like for the points with delta CS less than 50 hPa?

Thank you for pointing out the possible misunderstanding. We changed the text to:

"As in Judd et al. (2020) the criterion is looking for differences between the cloud pressure and the surface pressure (delta CS), but different from Judd et al. (2020), data with delta CS > 50hPa are kept and the observations where low clouds are retrieved are filtered out or replaced."

Line 604-606: 'This behavior is different from the small impact that we observed for changing the a priori NO2 profile information from TM5 to CAMS for the OFFL V01.03.02 dataset'. The change seems to be on the same order of magnitude rather than different.

Thank you for pointing this out, we changed it to: "With a relative difference of 14%, the change is showing a slightly larger impact than the 8% we found for changing the a priori NO2 profile information from TM5 to CAMS for the OFFL V01.03.02 data set."

Line 660. Saying cloud fractions are always lower than 0.14 contradicts from other examples in the text. (e.g., saying it was on average 0.21 in line 128).

Yes, right, this was a mistake. In the beginning we accidentally checked the cloud fraction instead of cloud radiance fraction to calculate the mean value. We thought we changed it everywhere in the text but have overseen it here. We changed it to the "on average 0.21 ± 0.10" as mentioned in line 128

Line 667: it is stated that on average TROPOMI is lower than air map but there are no averages reported in the manuscript.

Thank you for pointing out this formulation, we have changed it.

References:

van Geffen, J., Eskes, H., Compernolle, S., Pinardi, G., Verhoelst, T., Lambert, J.-C., Sneep, M., ter Linden, M., Ludewig, A., Boersma, K. F., and Veefkind, J. P.: Sentinel-5P TROPOMI NO2 retrieval: impact of version v2.2 improvements and comparisons with OMI and ground-based data, Atmospheric Measurement Techniques, 15, 2037–2060, https://doi.org/10.5194/amt-15-2037-2022, 2022.

Cede, A., Tiefengraber, M., Gebetsberger, M., and Spinei Lind, E.: Pandonia Global NetworkData Products Readme Document, Tech. rep., PGN-DataProducts-Readme, version 1.8-5, 31 December 2021, available at: https://www.pandonia-global-network.org/home/documents/reports/, last access: 2 December 2022, 2021.